

# Modelling the gas–particle partitioning and water uptake of isoprene-derived secondary organic aerosol at high and low relative humidity

Dalrin Ampritta Amaladhasan[1], Claudia Heyn[2], Christopher R. Hoyle[2,3], Imad El Haddad[2], Miriam Elser[2,5], Simone M. Pieber[2,6], Jay G. Slowik[2], Antonio Amorim[7], Jonathan Duplissy[8,9], Sebastian Ehrhart[4,10], Vladimir Makhmutov[11], Ugo Molteni[2], Matti Rissanen[12], Yuri Stozhkov[11], Robert Wagner[8], Armin Hansel[13], Jasper Kirkby[4,14], Neil M. Donahue[15], Rainer Volkamer[16], Urs Baltensperger[2], Martin Gysel-Beer[2], and Andreas Zuend[1]

[1]Department of Atmospheric and Oceanic Sciences, McGill University, Montreal, Quebec, H3A 0B9, Canada
[2]Laboratory of Atmospheric Chemistry, Paul Scherrer Institute, 5232 Villigen PSI, Switzerland
[3]Institute for Atmospheric and Climate Science, ETH Zurich, 8092 Zurich, Switzerland
[4]CERN, 1211 Geneva, Switzerland
[5]Swiss Federal Laboratories for Materials Science and Technology, Automotive Powertrain Technologies, Dübendorf, Switzerland
[6]Empa, Laboratory for Air Pollution / Environmental Technology, Ueberlandstrasse 129, CH-8600 Dübendorf, Switzerland
[7]CENTRA and Faculdade de Ciencias, University of Lisbon, 1749-016 Lisbon, Portugal
[8]Institute for Atmospheric and Earth System Research (INAR) / Physics, University of Helsinki, 00014 Helsinki, Finland
[9]Helsinki Institute of Physics, University of Helsinki, 00014 Helsinki, Finland
[10]Finnish Environment Institute (SYKE), Marine Research Centre, 00790, Helsinki, Finland
[11]P.N. Lebedev Physical Institute of the Russian Academy of Sciences, 119991 Moscow, Russian Federation
[12]Aerosol Physics Laboratory, Physics Unit, Tampere University, Tampere, Finland
[13]Department of Ion Physics and Applied Physics, University of Innsbruck, 6020 Innsbruck, Austria
[14]Institute for Atmospheric and Environmental Sciences, Goethe University Frankfurt, 60438 Frankfurt am Main, Germany
[15]Center for Atmospheric Particle Studies, Carnegie Mellon University, Pittsburgh, PA 15213, USA
[16]Department of Chemistry & CIRES, University of Colorado at Boulder, Boulder, CO 80305 USA.

*Correspondence to*:
Dalrin Ampritta Amaladhasan (dalrin.amaladhasan@mail.mcgill.ca) or Andreas Zuend (andreas.zuend@mcgill.ca)

**Abstract.** This study presents a characterization of the hygroscopic growth behaviour and effects of different inorganic seed
particles on the formation of secondary organic aerosols (SOA) from the dark ozone-initiated oxidation of isoprene at low $NO_x$
conditions. We performed simulations of isoprene oxidation using a gas-phase chemical reaction mechanism based on the
Master Chemical Mechanism (MCM) in combination with an equilibrium gas–particle partitioning model to predict the SOA
concentration. The equilibrium model accounts for non-ideal mixing in liquid phases, including liquid–liquid phase separation
(LLPS), and is based on the AIOMFAC model for mixture non-ideality and the EVAPORATION model for pure compound
vapour pressures. Measurements from the Cosmics Leaving Outdoor Droplets (CLOUD) chamber experiments conducted at
the European Organization for Nuclear Research (CERN) for isoprene ozonolysis cases, were used to aid in parameterizing
the SOA yields at different atmospherically relevant temperatures, relative humidity (RH) and reacted isoprene concentrations.
To represent the isoprene ozonolysis-derived SOA, a selection of organic surrogate species is introduced in the coupled





modelling system. The model predicts a single, homogeneously mixed particle phase at all relative humidity levels for SOA formation in the absence of any inorganic seed particles. In the presence of aqueous sulfuric acid or ammonium bisulfate seed particles, the model predicts LLPS to occur below ~80 % RH, where the particles consist of an inorganic-rich liquid phase and an organic-rich liquid phase; however, with significant amounts of bisulfate and water partitioned to the organic-rich phase.

The measurements show an enhancement in the SOA amounts at 85 % RH compared to 35 % RH for both the seed-free and seeded cases. The model predictions of RH-dependent SOA yield enhancements at 85 % RH vs. 35 % RH are 1.80 for a seed-free case, 1.52 for the case with ammonium bisulfate seed and 1.06 for the case with sulfuric acid seed. Predicted SOA yields are enhanced in the presence of an aqueous inorganic seed, regardless of the seed type (ammonium sulfate, ammonium bisulfate or sulfuric acid) in comparison with seed-free conditions at the same RH level. We discuss the comparison of model-predicted

SOA yields with a selection of other laboratory studies on isoprene SOA formation conducted at different temperatures and for a variety of reacted isoprene concentrations.

## 1 Introduction

Atmospheric aerosols, in particular the sub-micrometer sized fraction of particles, have a significant impact on air quality, visibility, cloud formation, and the radiative balance of the Earth's climate system (Kanakidou et al., 2005; Lohmann and Feichter, 2005). Organic matter typically amounts to a substantial fraction of the total aerosol mass in the troposphere. On average 20 – 60 % of the aerosol mass concentration in the continental mid-latitudes (Yu et al., 2007; Zhang et al., 2007;

Docherty et al., 2008) and up to 90 % of the aerosol mass concentration in the tropical atmosphere are due to primary emissions and secondary formation of organic aerosol (Artaxo et al., 2013; Pöhlker et al., 2016). Secondary organic aerosol (SOA) generated by the chemical conversion and partitioning of biogenic and anthropogenic precursor emissions account for a major portion of the total organic aerosol fraction. Therefore, understanding the sources, composition and properties of SOA is crucial to account for the physiochemical processes of SOA formation in air quality and global chemistry–climate models. Field

observations, laboratory chamber measurements and modelling studies have been conducted to estimate the contribution of SOA and its various sub-classifications to the global aerosol budget.

Gas–particle partitioning is a key process responsible for the formation and evolution of SOA when semi-volatile organic compounds (SVOCs) and low-volatility organic compounds (LVOCs) partition between the gas and particle phases governed by thermodynamic equilibrium (Kroll and Seinfeld, 2008). Thermodynamic models have been developed for the computation

of the gas–particle partitioning of organic–inorganic aerosol systems, often for process-level studies (Erdakos and Pankow, 2004; Chang and Pankow, 2010). Such thermodynamic models, run in a quasi-instantaneous equilibration mode or coupled to dynamic mass transfer models, may be implemented in atmospheric 3-dimensional chemical transport models for improvements in the accuracy of the aerosol mass concentration prediction. This serves assessments and operational



forecasting of chemical and physical aerosol processes affecting air quality and aerosol–cloud–climate interactions (Johnson et al., 2006; Cappa et al., 2008; Hallquist et al., 2009). Laboratory experiments and improved model parameterizations addressing the dependence of SOA formation on relative humidity, liquid mixture non-ideality, temperature and acidic/neutral inorganic seed particles are essential to deal with the dynamic nature of precursor-specific systems.

Isoprene (2-methyl-1,3-butadiene, $C_5H_8$) is among the most abundant non-methane biogenic volatile organic compounds present in tropospheric air (Guenther et al., 2006). The two carbon–carbon double bonds of isoprene make it highly susceptible for oxidation by OH radicals, $NO_3$ radicals and ozone ($O_3$) in the atmosphere (Kleindienst et al., 2007; Carlton et al., 2009; Paulot et al., 2009; Perring et al., 2009). Studies about isoprene oxidation in the 1990's suggested that this species does not contribute significantly to the atmospheric SOA budget (Pandis et al., 1991) because early-generation oxidation species from

isoprene are highly volatile. Subsequent laboratory chamber experiments and field observations over the past 20 years have shown that multi-generation oxidation of isoprene and its oxidation products contribute considerably to organic aerosol mass in the atmosphere by the formation of SVOCs, LVOCs and extremely low-volatility organic compounds (ELVOCs) (Claeys et al., 2004b; Dommen et al., 2009; Edney et al., 2005; Kroll et al., 2005; Kleindienst et al., 2006; Kroll et al., 2006; Kleindienst et al., 2007; Matsunaga et al., 2005; Ng et al., 2008; Nguyen et al., 2010; Surratt et al., 2010; Lin et al., 2011; Mao et al., 2013;

Hu et al., 2015; Jokinen et al., 2015; Krechmer et al., 2015; Song et al., 2015; Xu et al., 2015; Xiong et al., 2015; Kourtchev et al., 2016; Lopez-Hilfiker et al., 2016; Rattanavaraha et al., 2016; Riva et al., 2016). Several chemical pathways have been put forward to describe the formation of semi-volatile and low-volatility higher-generation oxidation products via the oxidation of early-generation compounds (Bates et al., 2014; Kameel et al., 2013; Kroll et al., 2006; Mao et al., 2013; Surratt et al., 2010; Worton et al., 2013). Reactive uptake of epoxydiols has been determined as a significant pathway for SOA formation (Kramer

et al., 2016; Riedel et al., 2016; Riva et al., 2016).

Field, laboratory, and modelling studies have been conducted to study the hygroscopic growth of SOA at varying RH levels in the atmosphere and the subsequent contribution to overall aerosol mass as a result of gas–particle partitioning of SOA from isoprene (Carlton and Turpin, 2013; Ervens et al., 2011; Hennigan et al., 2009; Huang et al., 2011; Lim et al., 2010; Marais et al., 2016). Among the first generation products of isoprene oxidation, the role of methyl vinyl ketone (MVK) and methacrolein

(MACR) in reacting with ozone to form Criegee intermediates, which further oxidize to form products of higher molecular mass (SVOCs and LVOCs), has been determined via analysis of experimental rate constants by Neeb et al. (1998). Laboratory experiments quantifying the heterogeneous, aqueous-phase ozonolysis of MVK and MACR formed from isoprene by Chen et al. (2008) also indicate that they contribute to a significant amount to particulate matter by forming higher generation products of lower volatility compared to the parent species. Additionally, studies conducted by Carlton et al. (2009), Surratt et al. (2010),

Kroll et al. (2006), Nguyen et al. (2010), and Couvidat and Seigneur (2011) suggest that the oxidation of MVK and MACR leads to substantial SOA formation from isoprene via further gas-phase oxidation and subsequent gas–particle partitioning of semivolatiles.



Guenther et al. (2006) suggest that the global biogenic emissions of isoprene (~600 Tg yr$^{-1}$) are sufficiently large that the amount of derived SOA formed, even when in small mass yield relative to the isoprene concentration, leads to considerable production of atmospheric particulate matter, thus having a substantial impact on air quality and biogenic aerosol–radiation–climate effects. The formation of semi-volatile and low-volatility compounds during isoprene oxidation has been assessed to

be one of the largest sources of atmospheric organic aerosol mass concentration contributed by a single parent VOC (Claeys et al., 2004a). Hence it is essential to account for isoprene emissions and resulting SOA formation in large-scale models (Couvidat and Seigneur, 2011). Photooxidation of isoprene by hydroxyl radicals (OH) in the gas phase during daytime is considered to be the dominant pathway for the formation of isoprene-derived SOA (Wolfe et al., 2016). Ensuing studies, to re-examine the contributions to SOA by the ozone-initiated oxidation of isoprene, indicate that contributions to aerosol mass by

the ozonolysis reaction pathway, and by combined ozone + OH radical oxidation, can contribute to SOA mass at atmospheric conditions, albeit to a smaller extent than the daytime OH pathway (Kleindienst et al., 2007). Studies focusing on the contribution of semi-volatile and intermediate-volatility species to water-soluble organic compounds in the aqueous phase of aerosols, suggest that heterogeneous reactions are responsible for the increased water-uptake by aerosols (Aumont et al., 2000; Matsunaga et al., 2003, 2004). Additionally, modelling studies (Ervens et al., 2004; Ervens et al., 2008; Lim et al., 2005)

predict that hygroscopic SOA formed by aqueous-phase reactions of isoprene-derived oxidation products influence the global SOA budget significantly. Hence, the water-uptake and hygroscopic growth of SOA from the ozone-initiated oxidation of isoprene is of interest and motivates a realistic representation of its hygroscopicity and gas–particle partitioning in process models, as well as in parameterizations for use in large-scale models.

Laboratory studies and thermodynamic model predictions suggest that the non-ideality of mixtures in liquid particle phases

influences the gas–particle partitioning process of semi-volatile species, including water, which in turn affects the thermodynamic state of the condensed phase, potentially leading to liquid–liquid phase separation (LLPS) (Pankow, 2003; Erdakos and Pankow, 2004; Zuend et al., 2010; Bertram et al., 2011; Smith et al., 2011; Song et al., 2012; Zuend and Seinfeld, 2012). Fine aerosol particles that exhibit LLPS up to high RH can show modified cloud condensation nucleus (CCN) properties compared to single-phase assumptions (Hodas et al., 2016; Renbaum-Wolff et al., 2016; Lin et al., 2017; Ovadnevaite et al.,

2017; Rastak et al., 2017; Song et al., 2017; Lei et al., 2018). Thermodynamic models taking into account the non-ideal liquid-phase interactions of SOA components formed from the photooxidation of isoprene by OH radicals have been developed by Couvidat and Seigneur (2011) and Beardsley and Jang (2016).

Experimental studies of isoprene-derived SOA (Lambe et al., 2015; Zhang et al., 2011) observed an increase in SOA formed in the presence of acidified sulfate seed aerosols relative to neutral seed, as a result of acid-catalysed heterogeneous reactions

in the former seed system (Czoschke et al., 2003). Second-generation epoxydiols of isoprene (IEPOX), formed via the oxidation of isoprene by OH in the presence of acidic seed aerosols, have been shown to play a major role in the enhanced SOA formation under low-NO$_x$ conditions (Paulot et al., 2009; Surratt et al., 2010). Laboratory environmental chamber studies



of isoprene ozonolysis in the presence of acidified seed particles suggest that SOA yields may be underestimated by current air quality and chemistry–climate models (Riva et al., 2016; Nakayama et al., 2018).

This modelling study focuses on a better quantitative understanding of the hygroscopic growth and gas–liquid partitioning behaviour of SOA surrogate systems representing, in a simplified manner, the SOA formed from ozone-initiated isoprene oxidation. SOA formed by the ozone-initiated oxidation of isoprene in the dark (no OH radical scavenger used) is modelled using the equilibrium gas–particle partitioning framework developed by Zuend et al. (2010) with successive improvements by Zuend and Seinfeld (2012). We explore the impacts of relative humidity and therefore particle water content, temperature (5 ºC, 10 ºC, 25 ºC), and the effects of inorganic seed aerosol on the particle-phase mixing behaviour with SOA, while the gas-phase chemical mechanism is kept the same. The model predictions are compared to isoprene ozonolysis chamber experiments, conducted at the Cosmics Leaving Outdoor Droplets (CLOUD) chamber at the European Organization for Nuclear Research (CERN), as well as to additional published data on isoprene-derived SOA under different conditions. Selected data sets from CLOUD experiments are also used to assess and tune adjustable model parameters, as described in Sect. 2. However, in this work the emphasis is placed on the modeling approach and predicted seed and RH-dependences rather than the details of the chamber experiments.

## 2 Methods and Data

### 2.1 Experimental data

Environmental chamber experiments were conducted in a continuous flow mode using the stainless-steel, 26.1 m$^3$ CLOUD chamber at CERN to study the dark ozonolysis of isoprene with and without an inorganic seed at varying RH levels and isoprene to NO$_x$ ratios. A detailed description of the instrumentation, experimental setup and vapour wall-loss corrections used to obtain the determined SOA mass yields is given elsewhere (Fuchs, 2017). One of the goals of the experiments conducted by Fuchs and co-workers was to investigate secondary organic mass yields under different thermodynamic conditions, i.e. distinct temperatures and RH levels, including the intermittent effects of sporadic chamber operation at water vapour supersaturation with in-chamber cloud formation followed by episodes of RH below 95 %. Two measurement campaigns were conducted with the CLOUD chamber operated under low NO$_x$ conditions: (1) During the CLOUD 9 campaign, isoprene ozonolysis experiments were carried out for cases with either near-neutral ammonium sulfate seed or acidic sulfuric acid seed particles at high (~85 %) RH conditions and temperatures of +10 ºC as well as -10 ºC; (2) During the CLOUD 10 campaign in 2015, isoprene ozonolysis experiments in the absence of any seed particles were conducted for relatively low (~35 %) and high (~85 %) RH conditions at 5 ºC.

Mass concentrations of reacted isoprene and early generation products were modelled by the Master Chemical Mechanism (MCM) using the mixing ratios of isoprene and ozone measured in the chamber during the CLOUD 9 and CLOUD 10 campaigns. During those campaigns, the isoprene concentrations were measured with a specially designed proton transfer reaction time-of-flight mass spectrometer (PTR-ToF-MS) instrument described in Bernhammer et al. (2017).The aerosol





particle number size distributions were measured with scanning mobility particle sizer (SMPS) systems (Wiedensohler et al., 2012) consisting of a differential mobility sizer (DMA) and a condensation particle counter (CPC). The average elemental chemical composition of the aerosol particles was measured with an Aerodyne high-resolution time-of-flight aerosol mass spectrometer (HR-ToF-AMS) (DeCarlo et al., 2006). The total organic mass concentration, resulting from the gas-phase and

aqueous-phase oxidation chemistry under dynamic gas–particle partitioning, was derived from the SMPS measurements of the particle number–size distribution, assuming an average aerosol mass density of 1.3 g cm$^{-3}$ (Fuchs, 2017). The average aerosol mass density was calculated based on the HR-ToF-AMS-determined mass fractions by using a parameterization by Kuwata et al. (2011), which is based on the elemental oxygen-to-carbon ratio (O:C) and the hydrogen-to-carbon ratio (H:C). The SOA mass yields were then determined from the organic mass concentration relative to the amounts of reacted isoprene as a function

of time. For this modelling study, experimental data on SOA mass formed under the different conditions is compared to predicted particle composition under thermodynamic and chemical reaction conditions comparable to those of the experiments.

**2.2 Equilibrium gas–particle partitioning framework**

A combination of three models is used in this study to cover distinct aspects of the chemistry and the gas–particle partitioning

thermodynamics of SOA formed from isoprene oxidation. Briefly, liquid-state pure component vapour pressures of SOA surrogate compounds were computed using the Estimation of VApour Pressure of ORganics, Accounting for Temperature, Intramolecular, and Non-additivity effects (EVAPORATION) method by Compernolle et al. (2011), described further below. The Aerosol Inorganic–Organic Mixtures Functional groups Activity Coefficients (AIOMFAC) model by Zuend et al. (2008; 2011) was employed to account for non-ideal mixing behaviour in the condensed aqueous organic–inorganic solution phases.

The MCM (version 3.3.1) (Jenkin et al., 2015) was used to describe the gas phase chemistry of isoprene oxidation in the absence of an OH-radical scavenger (sometimes considered in other studies). The equilibrium gas–particle partitioning framework by Zuend and Seinfeld (2012), a further developed variant of which is employed here, includes the consideration of LLPS and the solid–liquid equilibrium (SLE) of selected inorganic salts (e.g. ammonium sulfate) up to the high ionic strengths potentially occurring in aqueous organic–inorganic mixtures as the relative humidity and liquid-phase compositions

vary. The significance of the equilibrium partitioning model lies in its ability to perform computationally demanding calculations for coupled gas–particle and liquid–liquid partitioning of isoprene-derived aerosols at RH values comparable to those used in experiments, as well as at various other thermodynamic conditions (RH, temperature, seed type) beyond the experimentally accessible range.

While the gas phase is considered to be an ideal gas mixture, the condensed matter is considered to be a non-ideal liquid

mixture of organic and inorganic species (Zuend and Seinfeld, 2012). Therefore, the degree of non-ideality in liquid aerosol phases is taken into account via the calculation of component- and mixture-specific activity coefficients.

AIOMFAC is a thermodynamic group-contribution model designed for the calculation of mole fraction-based activity coefficients of different chemical species in each phase of inorganic-organic mixtures developed by (Zuend et al., 2008; Zuend





et al., 2011). The AIOMFAC model calculates activity coefficients for mixtures containing water, organic and inorganic components (electrolytes treated as partially or completely dissociated into ions). In this model, the organic compounds are represented in terms of sets of functional groups mapped to their chemical structures. The abundance and types of functional groups of organic compounds are critical in determining whether, and the extent to which, an aqueous organic mixture will

undergo liquid–liquid phase separation at equilibrium conditions (Zuend et al., 2010; Zuend and Seinfeld, 2013). Liquid–liquid phase separation is often induced in the case of mixtures containing relatively low-polarity organics in the presence of dissolved inorganic ions (Song et al., 2012; You et al., 2014; Zuend and Seinfeld, 2013), while it is expected to be either absent or less distinctive in the case of isoprene-derived SOA, which tends to contain relatively polar, hygroscopic organic compounds (typically of O:C ratios above 0.7).

Depending on temperature and chemical composition, in the case of LLPS or SLE a combination of co-existing liquid and/or solid phases may be present, with up to two distinct liquid phases considered by the AIOMFAC-based equilibrium model used here. Knowing the degree of non-ideality in the liquid phases and the pure component properties, the mass concentrations of individual organic components in the gas phase, $C_i^g$, typically given in units of µg m$^{-3}$ air, is accounted for by modified Raoult's law, valid at gas–particle equilibrium (Zuend and Seinfeld, 2012):

$$C_i^g = C_i^0\, x_i^\alpha\, \gamma_i^{(x),\alpha}. \tag{1}$$

Here, $C_i^0$ is the liquid-state, pure-component gas phase saturation concentration of component $i$ at temperature $T$, e.g. calculated with the EVAPORATION model in the case of organic compounds or by an adequate parameterization in the case of water (Murphy and Koop, 2005). $x_i^\alpha$ is the mole fraction and $\gamma_i^{(x),\alpha}$ is the activity coefficient of $i$ in liquid phase α, defined on mole fraction basis indicated by superscript $(x)$. Here, the activity coefficients are computed by the AIOMFAC model for each phase

composition and temperature. The resulting mole fraction and the absolute molar amount of a component in a certain phase is a result of a non-linear partitioning calculation constrained by mass conservation conditions for the system components and the equilibrium conditions given by Eq. (1). The fraction $r_i^{PM}$ of component $i$ present in the overall particulate matter (PM) phase/phases with respect to the total molar or mass amount in the gas plus PM phases is described in terms of mass-based concentrations by

$$r_i^{PM} = \frac{C_i^{PM}}{C_i^{PM}+C_i^g}. \tag{2}$$

Here, $C_i^{PM}$ is the mass concentration of component $i$, collectively present in the PM. In our gas–particle partitioning method, an initial guess for the total PM mass concentration is used to iteratively solve a system of non-linear equations for the set of $r_i^{PM}$ values that fulfill gas–particle equilibrium (Zuend et al., 2010; Zuend and Seinfeld, 2012). Compounds with a resulting $r_i^{PM}$ value between 0.01 and 0.99 are typically considered to be part of the semi-volatile category under the given conditions

(Donahue et al., 2006; 2012). The components with $r_i^{PM}$ values close to 0.5 are the ones most sensitive in their dynamic gas-



particle partitioning with respect to small perturbations in the environmental conditions, such as gas phase dilution by "clean" air or a change in RH.

### 2.2.1 Pure component vapour pressures

The EVAPORATION model developed by Compernolle et al. (2011) is used in this study for the calculation of the temperature-dependent pure component vapour pressures of organic compounds in the liquid state. The pure component vapour pressure is important in determining the order-of-magnitude extent to which a component may partition to the particle phases at equilibrium (Barley and McFiggans, 2010; Booth et al., 2010), as a consequence of Eq. (1). EVAPORATION uses a group-contribution approach to represent the effects of different functional groups in organic molecules on the pure compound. This

approach is similar to other estimation methods, such as SIMPOL.1 (Pankow and Asher, 2008), but in addition, EVAPORATION also includes second-order chemical structure-describing parameters to take into account the effects of certain intermolecular group–group interactions and to correct limitations in the applied first-order functional group additivity assumption. The predicted pure-component liquid-state vapour pressure, $p_i^0(T)$, which is a function of temperature only, is used to calculate the saturation concentration $C_i^0$ via

$$C_i^0 = \frac{p_i^0 M_i}{RT}. \tag{3}$$

Here, $M_i$ is the molar mass of the compound, $R$ the universal gas constant, and $T$ the absolute temperature (all expressed in SI units or with appropriate conversion applied). The level of accuracy obtained for $p_i^0(T)$ is often important for the calculated total PM mass concentration and related model uncertainties when the component is in the semi-volatile range (O'Meara et al., 2014).

A parameterization for the temperature dependence of $p_i^0$ applied in EVAPORATION is given by Compernolle et al. (2011):

$$\log_{10}\left(\frac{p_i^0}{[1\,\text{atm}]}\right) = A_i + \frac{B_i}{T^\kappa}. \tag{4}$$

An optimal value of $\kappa = 1.5$ has been determined by Compernolle et al. (2011) to account for the vapour pressure of hydrocarbons with or without hetero-atoms over a wide temperature range; atm denotes the unit pressure scale here (1 atm = 101325 Pa). For applications in this study, Eq. (4) is solved for the values of $A_i$ and $B_i$ using $p_i^0$ calculated by the online

EVAPORATION model (http://tropo.aeronomie.be/index.php/models/evaporation/15-tropospheric/44-evaporation-run) at two sufficiently different temperatures (e.g. at 0 °C and 60 °C) for each component $i$. The component-specific coefficients $A_i$ and $B_i$ can then be used to parameterize $p_i^0$ for the system components in the temperature range of interest.

### 2.3 Simplified isoprene system

### 2.3.1 Oxidation product information



The selection of an SOA-forming set of characteristic chemical components is a key input to our modelling framework. The choice of surrogate components representing isoprene oxidation products is described in the following. Whether a component resides in the gas phase, the condensed phase or partitions significantly between both is, in principle, irrelevant at input, yet can be exploited to select mainly low-volatility and semi-volatile components that substantially contribute to SOA mass under

given environmental conditions. A state-of-the-art chemical mechanism for gas phase reactions, the Master Chemical Mechanism (MCM), version 3.3.1 (Jenkin et al., 2015), available online: http://mcm.leeds.ac.uk/MCM, was used to account for the reactions of isoprene with ozone as well as OH radicals formed during the reaction process. MCM is useful to predict the formation of stable, volatile early-generation products under given experimental conditions. For conditions mimicking the CLOUD chamber setting, a gas-phase chemistry simulation with the MCM provided the time evolution of the molecular

concentrations of reacted isoprene, ozone, as well as stable early-generation products formed, namely glyoxal, methylglyoxal, methyl vinyl ketone (MVK), hydroxymethacrylate (MACO$_3$H), 2-hydroperoxy-3-hydroxy-2-methylpropanal (MACROOH) and 2-hydroperoxy-3-hydroxy-2-hydroxymethyl propanal (HMACROOH). The major stable products predicted by the MCM for the first/early generations of oxidation are compounds of low molecular mass with relatively low O:C ratios and high vapour pressures (VOC to IVOC class compounds). Hence, these compounds reside predominantly in the gas phase and

organic particulate matter forms only at very low temperatures (low compared to 298 K), at very high relative humidity levels, or for very high (atmospherically irrelevant) concentrations of isoprene reacted. In order to validate the amounts of ozone, OH and "isoprene reacted" predicted by the MCM, conditions and measurements from the CLOUD 10 and CLOUD 9 chamber experiments, reported by Fuchs (2017), were used for a comparison with observations, further discussed in the following.

**2.3.2 Surrogate components**

Additional components of semi-volatile and low-volatility nature formed via multi-generation oxidation, which are not fully considered by MCM, need to be included in the partitioning model as part of the isoprene system. Therefore, we introduce a selection of organic surrogate species, included in the model in the form of frequently observed higher-generation oxidation products, to represent wider classes of isoprene ozonolysis-derived SOA compounds. The selected surrogate components are shown in Fig. 1. Oxalic acid along with compounds such as 2-methyltetrol, 2-hydroxy-dihydroperoxide, 2-methylglyceric acid

and a C$_5$-alkene triol are among the main semi-volatile and low-volatility products that are expected to form after oxidation (by ozone and by OH) of the early generation compounds under low NO$_x$ conditions. The selected C$_{10}$ hemiacetal dimer is a compound representing a whole class of potential dimer and oligomer compounds formed by accretion reactions from the above-mentioned five higher generation products. The five SVOC and LVOC compounds along with the ELVOC dimer form the set of surrogate species included in our model to account for the gas–particle partitioning of isoprene-derived SOA in this

study.



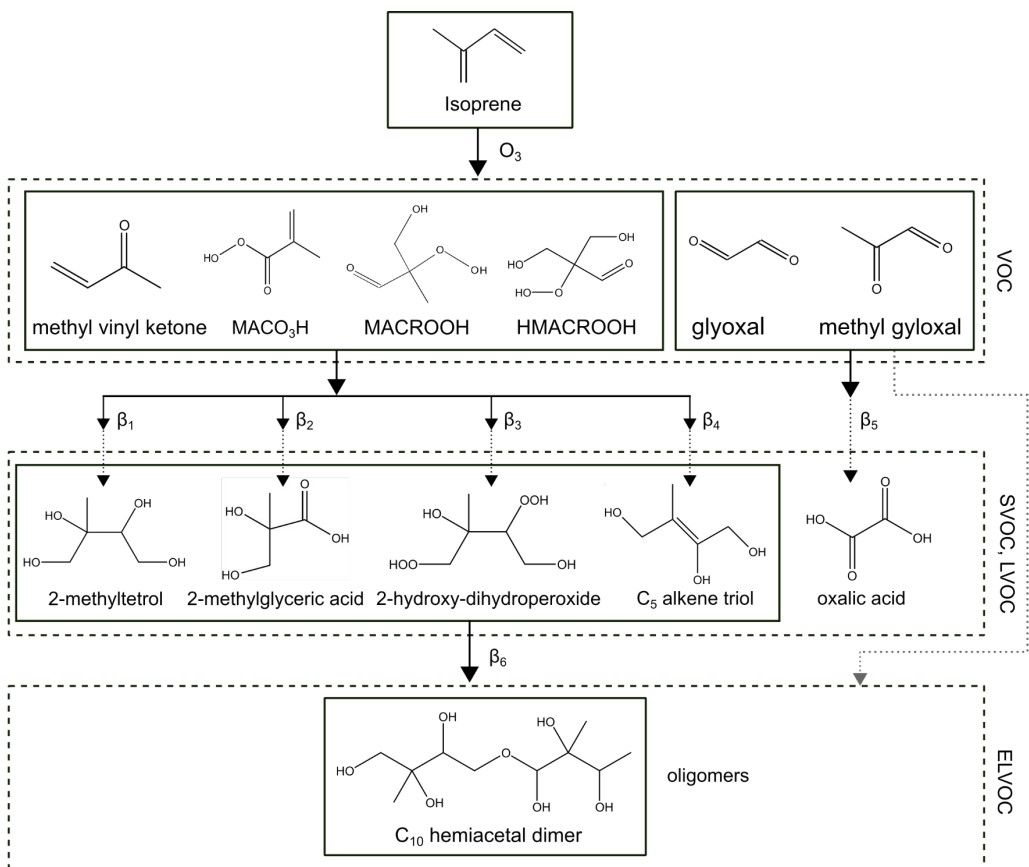

**Figure 1.** The system of selected isoprene-derived SOA surrogate components for use with the gas–particle partitioning model. This relatively simple system consists of early-generation gas-phase oxidation products, with yields that were directly predicted by MCM shown in the figure as VOC compounds, and a set of selected higher-generation oxidation products in the form of SVOC and LVOC surrogate species, including an oligomer species (an ELVOC at 283.15 K). $\beta_1 - \beta_6$ represent scaling parameters used to calculate the pseudo-molar yields of the higher-generation oxidation products from the concentrations of MCM-predicted early-generation species.

In an aqueous solution, it has been observed that glyoxal and methylglyoxal exist predominantly as their corresponding mono- or dihydrates, whereas MCM predicts the non-hydrated forms of glyoxal and methylglyoxal from the gas phase chemistry. To account for these different forms of the species, pure compound vapour pressures of the monohydrates of glyoxal and methylglyoxal have been considered to account for partitioning and the dihydrate forms for liquid-phase interactions in the thermodynamic mixing model. Also, monohydrates can further react in aqueous solutions to form hemiacetals, oxidized oligomers, imidazoles and organosulfate compounds (Ervens and Volkamer, 2010). Moreover, in the presence of aqueous





sulfate, glyoxal-dihydrates form ELVOC by displacing water molecules in the hydration shell of sulfate ions; this pathway is indicated by the dotted grey arrow in Fig. 1. This gives rise to a salting-in effect on glyoxal (Kampf et al., 2013; Waxman et al., 2015). At the molecular level, these ELVOC correspond to the formation of glyoxal-dihydrate-sulfate complexes (Kurtén et al., 2015); indicated in the upper left of Fig. 2 as a dotted grey range in volatility. Such complexes are presently not covered

in the AIOMFAC-based equilibrium partitioning framework. Therefore, in this study, the aqueous chemistry of glyoxal is only partially represented; however, given the relatively low gas-phase yields of glyoxal and methylglyoxal for the reaction conditions of the CLOUD experiments, omitting a more detailed aqueous phase representation of these species is considered acceptable. Moreover, the scaling parameters $\beta_5$ and $\beta_6$ can be considered to account for glyoxal-derived SVOC and ELVOC effects indirectly in the parameterized model (see discussion of scaling parameters in Sect. 2.3.4).

Predicted amounts from a MCM-based continuous flow chamber simulation for the two species $MACO_3$ and MACR (abbreviations are the names assigned by MCM) have been lumped into $MACO_3H$. Similarly, the cumulative amounts of $MACRO_2$ and MACROOH have been lumped into MACROOH because of the species' chemical pathways and the simplifying approach of avoiding chemical species in radical form from the lists of surrogates. We emphasize that the higher-generation surrogate products selected (and shown in Fig. 1) for the model are not formed directly from the (parent) early-generation

oxidation products; rather several unresolved steps of chemical reactions take place to arrive at those SVOC compounds. Here the emphasis is not on detailed chemical pathways; rather, we focus on the characterization of the SOA formation and growth due to the partitioning behaviour of the multigeneration oxidation products. Insights into typical compounds formed from isoprene oxidation gained from laboratory and field experiments were used to assign the set of higher-generation products in the model (Carlton et al., 2009; Couvidat and Seigneur, 2011; Surratt et al., 2010; Kroll et al., 2006). Surratt et al. (2010)

proposed a chemical mechanism via the reactive intermediate epoxydiols (IEPOX) pathway leading to the formation of 2-methyltetrols and $C_5$-alkene triols for the oxidation of isoprene by OH. 2-methyltetrol and $C_5$-alkene triol are part of the current surrogate model system, since OH radicals also form during the ozone-initiated oxidation of isoprene and its products (Zhang et al., 2018).

Multigeneration surrogate products from the MCM predicted yields of early-generation products are used to ensure that no

carbon mass is unaccounted for. Figure 2 shows the pure component volatilities as a function of the O:C ratio for all the surrogate components considered in the gas–particle partitioning model (details described in Section 2.3.4).

### 2.3.3 Steady-state flow chamber conditions

The CLOUD chamber is a continuous flow chamber in which the ozone-initiated isoprene oxidation process is controlled by

reactant inflow, mixing during the chamber mean residence time and continuous outflow of reactor components. At steady-state conditions, the rate of reactant inflow and outflow to/from the chamber is the same and the net variability of the reactant





species in the chamber is approximately zero. Therefore, the steady-state organic mass concentrations formed are associated with the amount of isoprene reacted per residence time period in the chamber. This includes the amount of isoprene reacted responsible for subsequently forming the SOA sampled by the instruments. Zhang et al. (2018), among others, proposed a kinetic modelling scheme and steady-state assumptions to understand the evolution of species concentrations in a continuous

flow chamber for the interpretation of experimental SOA mass concentration and yield data.

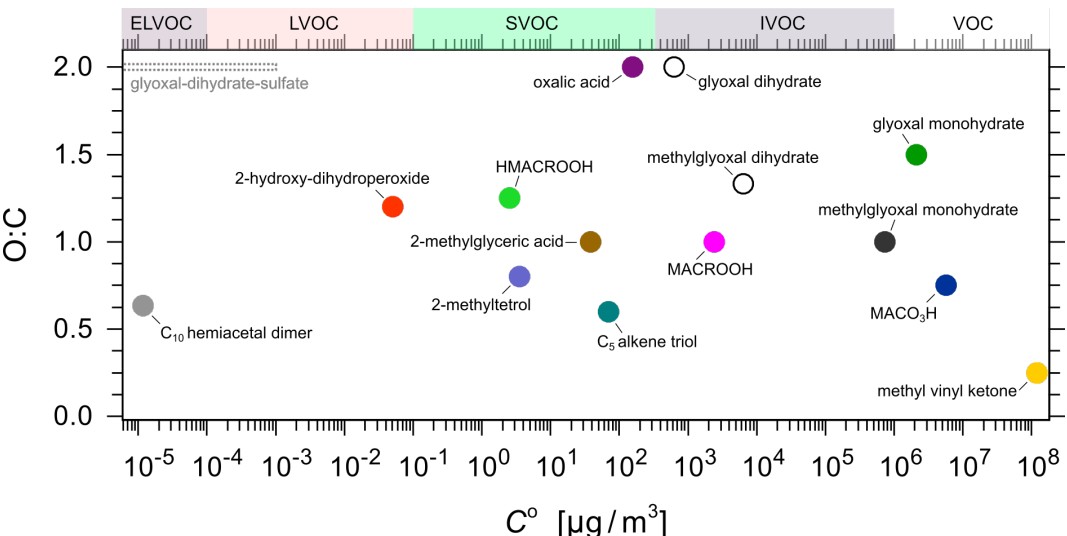

**Figure 2.** O:C ratio vs. pure component saturation mass concentration $C^0$ for the model system components (colored, solid

symbols). Selected other aqueous phase products from glyoxal and methylglyoxal hydration are shown as open symbols (there are many more that are not shown). The shown $C^0$ range for the system components at $T = 283.15$ K indicates that the MCM-predicted species are mostly VOCs and IVOCs whereas the selected higher generation products are SVOCs, LVOCs and ELVOC.

The concentration of isoprene reacted, [ISOPRCT], quantifies the modelled gas-phase concentration of an artificial tracer component added to the MCM simulation. It is the concentration resulting from the cumulative sum of all reactions of isoprene with $O_3$, OH or $NO_3$ including a correction for chamber dilution. In short, [ISOPRCT] is the difference between the isoprene inflow concentration and its steady-state concentration. This tracer is essential for the calculation of molar yields of

system components. We can express [ISOPRCT] by accounting for its reactive formation (*rf*) via isoprene oxidation, its loss due to dilution (*out*), its loss due to partitioning to chamber walls (*wl*) and loss due to condensation to particulate matter *(pm)*. The variability of [ISOPRCT] at steady state in the flow chamber is then given by



$$\frac{d[\text{ISOPRCT}]}{dt} = \left.\frac{\partial[\text{ISOPRCT}]}{\partial t}\right|_{rf} - \left.\frac{\partial[\text{ISOPRCT}]}{\partial t}\right|_{out} - \left.\frac{\partial[\text{ISOPRCT}]}{\partial t}\right|_{wl} - \left.\frac{\partial[\text{ISOPRCT}]}{\partial t}\right|_{pm} = 0. \tag{5}$$

This definition implies that there is no inflow of ISOPRCT into the chamber nor related first generation products of isoprene oxidation. As a simplification, assuming that there were no wall-loss nor loss by condensation to particles, Eq. (5) simplifies to (at steady state)

$$\left.\frac{\partial[\text{ISOPRCT}]}{\partial t}\right|_{rf} = \left.\frac{\partial[\text{ISOPRCT}]}{\partial t}\right|_{out} \tag{6}$$

$$\left.\frac{\partial[\text{ISOPRCT}]}{\partial t}\right|_{out} \approx \frac{[\text{ISOPRCT}]_{ss}}{V_{ch}} \cdot \frac{\Delta V_{ch}}{\Delta t} \tag{7}$$

Here, $V_{ch}$ is the volume of the chamber and $\frac{\Delta V_{ch}}{\Delta t}$ is the exchanged chamber volume portion during the discrete time interval $\Delta t$
due to inflow/outflow. Since $\Delta V_{ch}$ exchanged during residence time $\tau$ is equivalent to $V_{ch}$, we obtain from Eqs. (6, 7)

$$[\text{ISOPRCT}]_{ss} = \left.\frac{\partial[\text{ISOPRCT}]}{\partial t}\right|_{rf} \cdot \tau . \tag{8}$$

To this end, the MCM-predicted molecular concentrations of [ISOPRCT] are averaged over 30-minute time periods to account for quasi-steady-state concentrations in the chamber when flow conditions were maintained for an extended time period of at
least one hour (characteristic of the mean chamber residence time). The averaged amounts of isoprene reacted along with the corresponding steady-state mass concentrations of the various organic products formed are used to calculate pseudo molar yields for the MCM-predicted products. Molar yields of MCM-predicted products used in the model for the CLOUD 10 (seed-free) case were different based on the individual concentration of components present at the time when the chamber stabilised for the seed-free experiment. In the CLOUD 9 (seeded) case, during specific time intervals, cloud droplets were formed by the
process of adiabatic expansion whereby the previously pressurised chamber was depressurised from 220 hPa above ambient pressure back to ambient pressure (approx. 1013 hPa) for a time period of 8 minutes. Stable conditions were maintained for 1.5 hours during which the system relaxed back to subsaturated water vapour conditions. Additional adiabatic expansions were conducted over similar time intervals. Given the pressure, humidity and temperature variations with the cloud formation cycles during the CLOUD 9 experiments, the corresponding molar yields of the MCM-predicted products were based only on the
component concentrations at the times when the chamber had stabilised during subsaturated conditions, i.e. excluding time periods of cloud formation and immediately after.

Details of the MCM box model approach in Fuchs (2017) are as follows: the box model used inputs for the isoprene concentration, ozone concentration, condensation sink (from SMPS data), inorganic seed concentration, rainout rate constant (quantifies the rate of formation of cloud droplets and the associated mass loss in the cloud), dilution rate constant, aerosol
wall loss rate constant, RH, reaction rate constants of isoprene with the OH and $O_3$ oxidants, the hygroscopic growth factors



of inorganic seeds and organics. The virtual tracer ISOPRCT was simulated with oxidation of isoprene as the source and dilution as the only sink (i.e. without wall loss). In each time step, the produced ISOPRCT was instantaneously distributed to a volatility basis set distribution. In their comprehensive version of the box model, Fuchs (2017) simulated wall losses, dilution, condensation/evaporation, and hygroscopic growth assuming internal mixing and using the kinetic partitioning method by

Donahue et al. (2006) as a basis for calculating evaporation and the condensation sink to represent gas–particle partitioning. In this work we use output from a version of the MCM-based box model by Fuchs (2017) that employs a simpler wall-loss correction for generating the evolution of early generation oxidation products. These product concentrations at selected points in time were then used as the input data for our RH-dependent equilibrium partitioning model.

**2.3.4   Matching model predictions and observations: oxidant-specific scaling parameters**

In the absence of a mechanistic, quantitative prediction for the formation of the higher-generation products from the early-generation gas-phase oxidation products, their formation is here constrained by the quasi steady-state molar concentrations of the six MCM-predicted components (the VOCs in Fig. 1). Those components serve as parent compounds of the less volatile surrogate compounds chosen (see Fig. 1). This establishes a direct constraint in our model on the maximum molar (or mass)

amounts of carbon available for the formation of the SOA-relevant SVOC, LVOC and ELVOC compounds. The estimated molar amounts of surrogate species formed from the VOC precursors at a certain time in an experiment are expressed by a set of dimensionless scaling parameters, $\beta_i$, further described below.

The yield calculation for each higher generation compound (SVOC, LVOC) is done by constraining them with molar yields of the MCM-predicted gas phase oxidation products (the early generation products):

$$n_i = \beta_i \times \sum_j n_j^0. \tag{9}$$

Here, $n_i$ is the molar concentration (mol m$^{-3}$ of air) of the SVOC surrogate product "$i$" formed (2-methyltetrol, 2-hydroxy-dihydroperoxide, 2-methylglyceric acid or C$_5$ alkene triol), excluding oxalic acid (see Fig. 1); $n_j^0$ are the MCM-predicted molar concentrations at quasi-steady state of the parent products MVK, MACO$_3$H, MACROOH and HMACROOH, as indicated in the scheme of Fig. 1. Since the concentrations of these parent components decrease in the process of the formation of higher-

generation species, a resulting proportional decrease in each of the compounds consumed in Eq. (9) is accounted for by subtracting a mole-fraction-weighted amount of cumulative higher-generation products formed from each of the parent products considered:

$$n_j = n_j^0 - x_j^0 \times \sum_i n_i , \quad \text{with } x_j^0 = \frac{n_j^0}{\sum_l n_l^0}, \tag{10}$$


where indices $j$, $l$ cover the parent compounds and index $i$ the SVOC surrogate products formed (excluding oxalic acid).
Prior to the calculation of surrogate yields, a check is carried out to ensure that the sum of the assigned branching ratios does
not exceed 1 in order to maintain the carbon mass balance. Equations (9) and (10) describe an empirical scheme for the quasi-
instantaneous conversion of a fraction of the early generation products to higher-generation compounds, the latter being

important for the gas–particle partitioning of SOA. This approach is a substantial simplification of the actual chemical system;
however, this scheme provides a physical constraint on the molar formation budget of surrogate species. The method used in
Eqs. (9) and (10) is also adopted for calculating the amount of oxalic acid formed from the parent products glyoxal and
methylglyoxal and their associated decrease in molar concentration.

The molar concentration $n_k$ of the surrogate oligomeric species $C_{10}$ hemiacetal dimer is calculated in a similar way, yet based
on the cumulative molar amounts of the SVOC surrogate components rather than directly from the early-generation products:

$$n_k = \beta_k \times (0.5 \times \sum_i n_i^0). \tag{11}$$

Here, $\beta_k$ is the parameter constraining the amount of dimer formed and the $n_i^0$ are the (initial) calculated molar concentrations
of its parent compounds, namely 2-methyltetrol, 2-hydroxy-dihydroperoxide, 2-methylglyceric acid and $C_5$ alkene triol. The
sum of the molar amounts of these four species is divided by two for estimating the molar dimer amount ($n_k$), because the
dimer (as a surrogate for a variety of similar $C_{10}$-dimers) is formed as a result of accretion reactions of certain combinations
of parent surrogate products, each of which contains typically half the number of carbon atoms compared to the formed dimer.
The corresponding decrease in each parent surrogate's concentration is accounted for by subtracting a mole-fraction-weighted
amount of dimer formed from each of the potential parents:

$$n_i = n_i^0 - x_i^0 n_k. \tag{12}$$

Our decision making process involved in setting initial scaling parameter values ($\beta_i$ and $\beta_k$) was informed by the O:C vs.
volatility space of all the system components, an approach that is also the basis of the two-dimensional volatility basis set
approach to describe the formation and growth of organic aerosols (Donahue et al., 2011; Donahue et al., 2012). In our simple
model, the scaling parameters (branching ratios) are adjustable and are determined iteratively by model–measurement
comparison. After the gas–particle partitioning model was used to calculate the equilibrium SOA mass concentration formed
at varying levels of reacted isoprene, covering the range observed in the CLOUD chamber experiments, and for high to low
relative humidity, the values of the scaling parameters were gradually adjusted in the model to influence the predicted molar
amounts of the different higher-generation species formed.  In addition, the distinct hygroscopic contributions to aerosol water
content and associated changes in gas–particle partitioning by all organic components also offered some guidance in the





optimization of the scaling parameters. Reasonably good agreement was achieved after a few iterations of this optimization process. The determined scaling parameters for the surrogate species considered by the model are provided in Table 1.

**Table 1.** Values of scaling parameters of the list of surrogate species used for the gas–particle partitioning calculations. For reference, a branching ratio value of $\beta = 6.6 \times 10^{-3}$ for 2-methyltetrol indicates that 0.66 % of the predicted molar yield from the product's cumulative parent species determine the yield of 2-methyltetrol.

| Surrogate component | Chemical structure (SMILES) | $M$ | $p^0\,(T = 283.15\ \text{K})$ | $\beta$ |
|---|---|---|---|---|
| | | (g mol$^{-1}$) | (Pa) | (-) |
| 2-methyltetrol | CC(CO)(C(CO)O)O | 136.146 | $5.92 \times 10^{-5}$ | $6.6 \times 10^{-3}$ |
| 2-methylglyceric acid | CC(CO)(C(=O)O)O | 120.104 | $7.51 \times 10^{-4}$ | $1.1 \times 10^{-2}$ |
| 2-hydroxy-dihydroperoxide | CC(O)(COO)C(CO)OO | 168.15 | $6.92 \times 10^{-5}$ | $3.0 \times 10^{-3}$ |
| $C_5$ alkene triol | CC(=CO)C(CO)O | 118.131 | $1.35 \times 10^{-3}$ | $1.1 \times 10^{-2}$ |
| Oxalic acid | C(=O)(C(=O)O)O | 90.035 | $4.02 \times 10^{-3}$ | $3.3 \times 10^{-2}$ |
| $C_{10}$ hemiacetal dimer | CC(O)C(C)(O)C(O)OCC(O)C(C)(O)CO | 254.28 | $1.00 \times 10^{-10}$ | $1.2 \times 10^{-1}$ |

## 3 Results and Discussion

### 3.1 Modelled isoprene SOA formation: seed-free case

The SOA mass concentrations and mass yields predicted by the partitioning model were adjusted to simultaneously match the measurements of SOA mass obtained at different levels of reacted isoprene concentrations from the CLOUD chamber experiments using the oxidant-specific scaling parameters described in section 2.3.4. In the following, we discuss these comparisons and investigate the predicted effect of equilibrium gas–particle partitioning and aqueous phase mixing at different RH levels beyond the experimentally explored range. The SOA mass yield is defined as the ratio of the total organic PM mass concentration formed relative to the total mass concentration of the precursor compound reacted to form it, in this case isoprene reacted (ISOPRCT). The hygroscopic growth of particulate matter formed as a result of isoprene ozonolysis was modelled for particles without an inorganic seed as well as for cases with ammonium sulfate, acidic ammonium bisulfate, or sulfuric acid seed particles at different organic to inorganic mixing ratios.

### 3.1.1 Impact of varying isoprene loading levels: seed-free case

Measurements from the CLOUD chamber for the seed-free experiments suggest an SOA mass yield of 1 % – 2.4 % at 35 % RH and 1.7 % – 3.7 % at 85 % RH at a temperature of 5 °C. These mass yields reflect the data when a simple, vapour wall-



loss correction is applied to the measurements, which then indirectly affects the fitted model parameters. We acknowledge that accounting for the actual wall-losses is difficult, may change the stated SOA yields substantially, but is beyond the focus of this study since we are mainly interested in understanding the partitioning effects at different RH levels. Such relative effects are likely not susceptible to the specific type of wall-loss treatment used. However, we note that accounting for a higher actual

wall-loss of low- and semivolatile species would result in a higher actual SOA mass yield than stated above. The isoprene and ozone mixing ratios for the seed-free experiments in the CLOUD chamber were 275 ppb$_v$ (ppb$_v$: parts per billion by volume) and 130 ppb$_v$, respectively, for both the low and high RH conditions. A key point in our evaluation of the RH-effect is that the same molar yields have been used for the MCM-predicted gas-phase concentrations of the early-generation products and the hypothetical model tracer species "isoprene reacted", covering the modelling of the SOA mass concentrations at all levels of

RH.

    The SOA mass concentration data used here were reported by Fuchs (2017). The measurements for SOA mass concentration vs. MCM-predicted isoprene reacted concentration at several points in time in the experiments were compared with the model results obtained for the same conditions using periods for which the quasi-steady state assumption was applicable; see Supplementary Information (SI), Table S1.

Based on the predicted early-generation product yields and the set of determined scaling parameters ($\beta$), the effective molar yields of the surrogate components were calculated with the AIOMFAC-based gas–particle and liquid–liquid equilibrium model for a given input level of reacted isoprene. The same scaling parameter values were used at high and low RH. This was done for two reasons: (i) the assumption that the chemistry behind the formation mechanism of the SOA products remains the same at low and high RH, since the absolute water vapour concentration at 35 % RH and 5 °C is still abundant (for gas-phase

chemistry); (ii) using the same parameters, the model predictions allow for a direct quantification of the role of seed hygroscopicity and associated aerosol water content on the gas–particle partitioning of all SOA components at variable RH. Figure 3 shows the modelled SOA mass vs. isoprene reacted in agreement with the experimental data at 35 % RH (panel a) and 85 % (panel c). This confirms the successful tuning of the scaling parameters $\beta$. Figure 3b shows the model–measurement comparison of SOA mass yield vs. SOA mass concentration at 35 % RH. The error bars shown in Fig. 3 represent the range

of uncertainty for measured SOA mass concentrations and yields. These error bars are based on an estimated relative error of 50 % in measured organic aerosol mass concentration (Fuchs, 2017). Figure panels 3c and 3d show the model calculations of SOA mass concentration and mass yield in comparison to the measurements at 85 % RH. The SOA mass varies in a non-linear manner with the isoprene reacted levels, which is in reasonable agreement with the predictions from the model. Nevertheless, the experimental data show some scatter at isoprene reacted concentrations between ~140 to 175 µg m$^{-3}$, which the model is

unable to reproduce when good agreement at lower isoprene reacted levels is achieved. Based on the seed-free SOA formation case shown in Fig. 3, the measurements and model predictions reveal a relative humidity effect on the SOA mass concentration.




**Figure 3.** Comparison of CLOUD chamber measurements and model calculations of SOA formed for varying levels of isoprene reacted. Experimental conditions: +5 °C, at 35% RH and 85% RH without seed particles present in the continuous flow chamber. Model calculations shown as black curves were performed at the same temperature and RH as the experiments. **(a, c)** Model–measurement comparison of SOA mass concentration vs. isoprene reacted at (a) 35 % and (c) 85 % RH. Hollow green triangles show the measured data at MCM-determined levels of isoprene reacted [ISOPRCT] in the chamber (Fuchs, 2017), with error bars indicating 50 % measurement uncertainty for SOA mass concentration. **(b, d)** Comparison of SOA mass yield vs SOA mass concentration for (b) 35 % and (d) 85 % RH. Solid triangles: experimental data coloured based on level of isoprene reacted.



For example, for an isoprene reacted level of 150 µg m$^{-3}$ the predicted SOA mass concentration at gas–particle equilibrium for 35 % RH is about 2 µg m$^{-3}$, resulting in an organic mass yield of 1.33 % while the corresponding SOA mass concentration predicted for 85 % RH is ~3.5 µg m$^{-3}$, a 75 % enhancement, with an SOA mass yield of approximately 2.33 %. For this seed-free system, the model simulation therefore suggests that the measured increase of SOA mass yield at 85% RH compared to 35% RH can be fully explained by the effect of aerosol water uptake on the equilibrium phase partitioning of all system components.

### 3.1.2 Impact of RH on gas–particle partitioning: seed-free case

The effect of equilibrium partitioning with non-ideal mixing in the liquid phase was further studied using the AIOMFAC-based model over an extended range of RH for a fixed isoprene reacted steady-state concentration. The different SOA surrogate species are shown to provide distinct contributions to the organic PM mass as RH varies. Figure 4b shows stacked bar graphs of the predicted PM mass concentrations of the organic components for the range from near-zero up to 99 % water activity (equivalent to bulk equilibrium RH). The water content in the form of particle phase mass fractions is shown in panel (c) of Fig. 4. These model results were obtained by assuming that all PM-bound organic species remain in a liquid (or amorphous semi-solid) state regardless of RH (no crystallization allowed), an assumption that is considered to be valid for complex multicomponent SOA systems (Marcolli et al., 2004). Furthermore, due to the substantial hygroscopicity of isoprene SOA, as evident from Fig. 4c, as well as the semivolatile nature of organic components like the C$_5$ alkene triol and 2-methyl glyceric acid surrogate compounds, a continuous SOA mass and yield enhancement is predicted with increasing RH (compare panels a and b of Fig. 4). The model predicts this RH-dependence because of the coupled absorptive water uptake and enhanced gas-to-particle partitioning of the semivolatile organics in the absence of any potential changes in gas-phase or liquid-phase chemistry with changes in RH. In the case shown in Fig. 4, a SOA mass concentration of ~ 3.3 µg m$^{-3}$ (yield of ~ 2.64 %) is computed for a water activity of 95 %, more than double that obtained at 35 % RH. Figure S9 shows the predicted mole-fraction-based activity coefficients of the different system components as a function of water activity. In this seed-free system, without any inorganic electrolytes, several of the SOA surrogates show activity coefficients ranging between 0.3 and 1, especially in the RH range from 40 % to 96 %. This indicates favourable mixing among the liquid-phase components and, compared to an ideal mixing assumption on mole fraction basis, an enhanced SOA mass concentration in that RH range. In the RH range above 96 %, several SOA surrogate components show an increase in their activity coefficients (except for oxalic acid), indicative of unfavorable mixing in the presence of a large mass fraction of water.



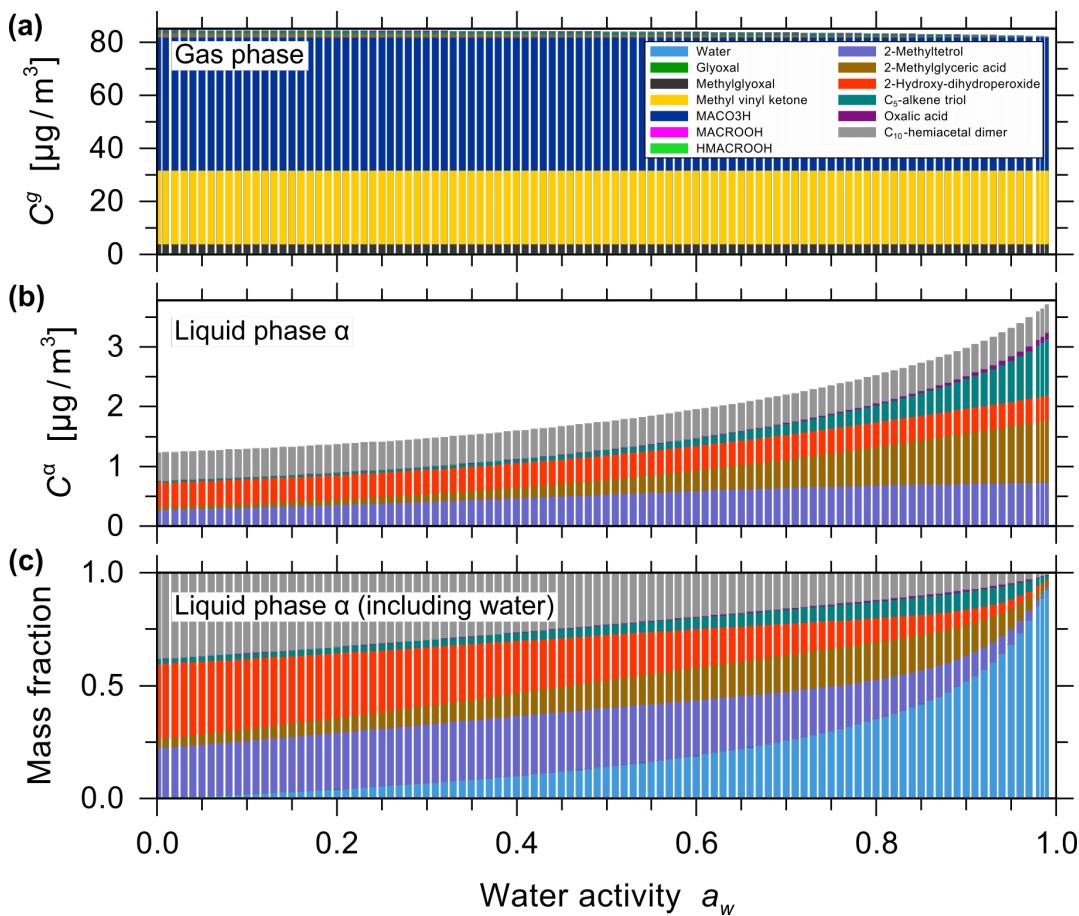

**Figure 4.** Predicted mass concentrations of the components (shown stacked; excluding water) in **(a)** the gas phase and **(b)** the single liquid particle phase present. **(c)** The mass fractions in the liquid phase, including water. All for 0 % to 99 % water activity (equilibrium RH) and a reacted isoprene concentration of 125 µg m$^{-3}$ at 5 °C for the seed-free case. The gas phase is dominated by early-generation species and the single liquid phase is dominated by the higher-generation surrogate species selected to represent SOA.

### 3.1.3 Comparison to other seed-free isoprene oxidation experiments

Since secondary OH radicals form during the ozonolysis of isoprene, quantifying the contribution of OH-reaction-derived SOA mass and the role of radical scavengers is of interest. Sato et al. (2013) discuss the effects of different OH radical scavengers used in sufficient concentrations during smog chamber experiments. They performed experiments using the OH radical scavengers cyclohexane, carbon monoxide, n-hexane, and diethyl ether in a static 6 m$^3$ environmental chamber at 25 °C. For the dark ozonolysis of isoprene without any seed particles and in the absence of an OH scavenger, Sato et al. (2013) obtained SOA yields in the range from 0.29 % to 2.3 % (after applying a particle wall-loss correction) for produced SOA mass



concentrations in the range from 2 to 120 µg m$^{-3}$, i.e. covering a much wider SOA concentration range than that explored in the CLOUD experiments, but only at dry conditions and a higher temperature. The initial isoprene and ozone concentrations in the chamber were in the range 0.5 – 2 ppm$_v$ (approximately 1400 – 5500 µg m$^{-3}$) and 2 – 4 ppm$_v$ at a temperature of 25 °C under very dry conditions (RH < 1 %). Using our equilibrium model along with the same reacted isoprene concentration of

5484 µg m$^{-3}$ and temperature as during the experiments by Sato et al. (2013), an SOA yield of ~ 1.28 % was calculated for dry conditions of 1 % RH (also: ~ 2.62 % yield at 85 % RH), which is within the range of mass yields observed in the experimental study (see Table 2 and Fig. S1). The same pseudo-molar yields for the early-generation products and scaling parameters for the higher generation products as for the CLOUD 10 seed-free case were used for the model calculations. Repeating the model calculation for 5484 µg m$^{-3}$ of isoprene reacted but at 5 °C results in a predicted SOA mass yield of ~ 2.82 % at 1 % RH

(~ 3.07 % yield at 85 % RH). This indicates that the temperature effect (a difference of 20 °C between 5 °C in the CLOUD chamber and 25 °C in the chamber studied by Sato et al. (2013)) increases the pure component volatilities of the surrogate species sufficiently to decrease their partitioning to the aerosol. Since a relatively large fraction of SOA in the model surrogate system is contributed by SVOCs, their gas–particle partitioning is sensitive to changes in temperature.

Experimental studies on the temperature effects of SOA formed from the dark ozonolysis of isoprene conducted by Clark et

al. (2016) detected SOA yields as high as 9 % at 5 °C under dry conditions (RH < 1 %). Their experiments were conducted in a polytetrafluoroethylene (PTFE; Teflon) chamber at initial isoprene and ozone concentrations of 250 ppb$_v$ and 125 ppb$_v$ and an isoprene reacted level of 278 µg m$^{-3}$. Using the equilibrium model to predict isoprene SOA formation for the same conditions as the latter study, an SOA yield of ~ 1.25 % was predicted for dry conditions of ~1 % RH; see Fig. S2. It is observed that even though the experimental conditions in the chamber by Clark et al. (2016) resemble the original CLOUD experiment

conditions, the measured SOA yield by Fuchs (2017) are lower than the SOA yield reported by Clark et al. (2016). The difference in the measured SOA yields is likely explained by differences in the type and sophistication of the particle wall-loss correction procedure used by Clark et al. (2016) compared to the simple vapour wall-loss correction applied to the CLOUD data as referenced in this study. The influence of vapour wall loss in stainless steel chambers has been studied by numerical simulations (Voigtländer et al., 2012) to acquire a better understanding of the vapour wall deposition in the CLOUD chamber

during different types of experiments. Using the SOA mass measured from the CLOUD 10 experiments, it is predicted that pure compound vapour pressures of the semivolatile components constituting the SOA are lowered when the temperature decreases from 25 °C to 5 °C (as is expected). Hence, with this temperature effect, the SOA yields during dark ozonolysis of isoprene at 5 °C become comparable to those reported for isoprene photooxidation at 25 °C (Chhabra et al., 2010; Lambe et al., 2015). The relatively high SOA yield observed in the CLOUD ozonolysis experiments may also partially be due to the

absence of an OH radical scavenger.

Among other model–measurement studies on the photooxidation of isoprene, the MCM and SOA partitioning model results for OH-initiated oxidation of isoprene reported by Chen et al. (2011) are used for comparison with our model predictions for the ozone-initiated oxidation (SI, Sect. 2). For this comparison, a different set of system components, based on components





listed by Chen et al. (2011) for the isoprene oxidation system without an inorganic seed, were used with our AIOMFAC-based model to calculate the gas and particle phase compositions at different levels of water activity. The SOA yields originally predicted by Chen et al. (2011) range from 1.08 % to 2.69 % at isoprene reacted concentrations ranging from 134 to 334 µg m$^{-3}$ at 40 % RH and 25 °C. For the Chen et al. (2011) surrogate mixture, as modified by Rastak et al. (2017) (see Table S3), our

model predicts the SOA mass to exist in a single particle phase at all RH levels, as shown in Fig. 5. For calibration, we constrained our model input composition in terms of total gas plus particle phase molar concentrations such that it predicts the same equilibrium SOA mass concentration as the Chen et al. (2011) model at a reference point: 3.6 µg m$^{-3}$ SOA at 40 % RH, 25 °C and a mean isoprene reacted concentration of 147 µg m$^{-3}$. Therefore, the SOA yields predicted both by Chen et al. (2011) and the AIOMFAC-based model are 2.4 % at 40 % RH. At 85 % RH and $T$ = 25 °C, our model predicts an SOA mass

concentration of ~ 6.2 µg m$^{-3}$ and an SOA yield of ~ 4.2 % for the photooxidation case, since significant amounts of SVOCs partition to the PM at high RH levels (> 80 % RH) similar to the hygroscopic growth and coupled enhanced SVOC partitioning determined for the isoprene ozonolysis case.

The effect of RH on the SOA yield enhancement for the photooxidation of isoprene with OH is observed from the ratio of SOA yield at a temperature of 25 °C for 85 % RH vs. 35 % RH as ~1.97 (= 4.1/2.1). The SOA yield enhancement for the

photooxidation of isoprene at a temperature of 5 °C for 85 % RH vs. 35 % RH is ~ 1.2 (= 7.5/6.4). Thus, SOA yield enhancement with respect to high and low RH levels is here predicted to decrease with decreasing temperature. The SOA yield enhancement for the ozone-initiated oxidation of isoprene at a temperature of 5 °C for 85 % RH and 35 % RH is ~ 1.8. Hence, the SOA yield enhancement for the ozone-initiated oxidation of isoprene is estimated to be higher than the SOA yield enhancement for the photooxidation of isoprene at a temperature of 5 °C. However, we note that the temperature-dependence

of the yield enhancement is also dependent on the choice of surrogate components. Since the SOA mass was only constrained at low RH levels for the photooxidation case, a proper quantification of (chamber-independent) SOA yields at high RH levels are imperative to better constraining and better understanding the role of isoprene SOA as a source of organic particulate matter in global models.

**3.2 Modelled isoprene SOA partitioning in the presence of acidic sulfate seed**

In the following, we discuss the measured SOA mass concentrations from CLOUD chamber experiments at ~85 % RH and low NO$_x$ concentrations with varying levels of seed aerosol concentration in comparison with the predictions from the gas–particle partitioning model calculated at conditions consistent with those experiments. We consider several cases with either a moderately acidic ammonium bisulfate seed aerosol or a highly acidic sulfuric acid seed for experiments conducted at 10 ºC.

The isoprene and ozone mixing ratios for the seeded experiments in the CLOUD chamber were 120 ppb$_v$ and 100 ppb$_v$, respectively. Ideally, the values of chemical branching ratios for determining the amounts of surrogate SOA species formed should remain very similar to those for the non-seeded case, since the gas phase chemistry involved in the formation of isoprene SOA is presumably the same in both cases.

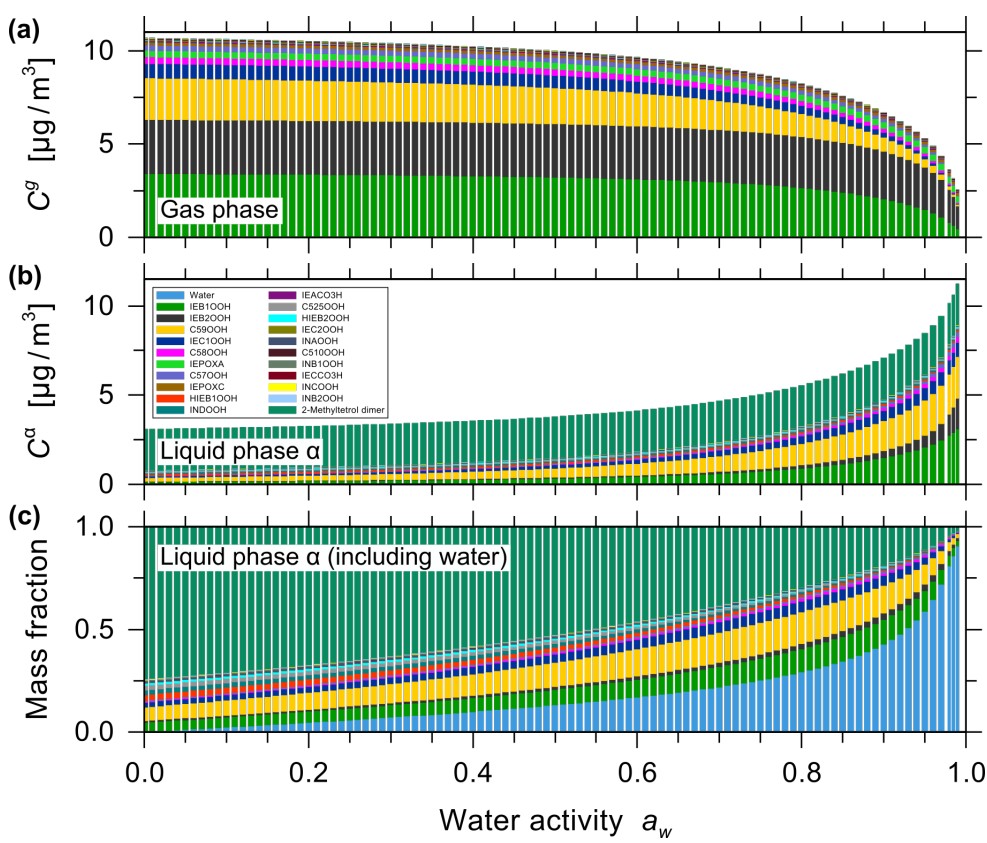

**Figure 5.** Predicted mass concentrations for **(a)** the gas phase and **(b)** the single liquid particle phase of the components (shown

stacked; excluding water) for the isoprene photooxidation case using system components reported by Chen et al. (2011). **(c)**
The mass fractions in the liquid phase including water. All calculations are for 0.1 % to 99 % water activity (equilibrium RH)
and a reacted isoprene concentration of 147 μg m$^{-3}$ at 25 °C for a seed-free case.

However, aqueous phase chemical reactions as well as gas–particle partitioning may be significantly affected by the presence
of inorganic ions, contributed particle water content and liquid phase acidity (Barsanti and Pankow, 2005; Surratt et al., 2007;
Volkamer et al., 2009; Nguyen et al., 2011; Zuend and Seinfeld, 2012; Pye et al., 2020). Notably, our model does not treat
aqueous SOA reactions that form hemiacetals, oligomers, imidazoles, or organic sulfates (Kampf et al., 2013; Waxman et al.,
2015; Kurtén et al., 2015; Sareen et al., 2017). It is unclear to what degree the dissolution of water-soluble species, and their

further processing in either a separate inorganic-rich aqueous phase or a mixed aqueous organic–inorganic phase contributes
aqueous SOA reaction pathways that may be captured (partly) in the empirically determined scaling parameters (Fig. 1). This





topic deserves further investigation. However, in the absence of OH radicals and light in the CLOUD experiments, the rates of aqueous SOA formation pathways are limited to dark processes and are likely only of minor importance.

Since the CLOUD chamber was run under different conditions during the seeded experiments, this leads to different pseudo-molar yields of early-generation products considered in our model (see Table S2). For consistency, the same molar yields were

used for all seeded cases considered since the differences among the experimental conditions during the seeded experiments were relatively minor (yet different compared to seed-free). The experimental data for the seed mass concentrations and steady-state SOA mass concentrations indicate that the organic aerosol mass concentration depends on the mass concentration of the inorganic seed particles at given steady-state concentrations of isoprene reacted. Hence, the gas–particle partitioning model was modified to account for RH-dependent SOA and water partitioning in the presence of an aqueous phase of variable

inorganic seed concentration. We note that the measured steady-state flow chamber aerosol mass concentration is affected by the dynamic interplay between the different vapour and particle loss mechanisms: wall-loss, loss by chamber outflow, and the aerosol condensation sink. The latter depends on the particle size distribution and therefore also on humidity-dependent water uptake by seed particles and organic partitioning.

**3.2.1 Effect of aqueous ammonium bisulfate seed: impact of varying seed mass concentration**

In the following, measurements of SOA mass concentrations in the presence of sulfate- and ammonium-rich aqueous inorganic seed particle populations are compared with model calculations for similar conditions in terms of temperature, pressure, RH, ranges of isoprene reacted and seed concentrations. Since the measured total sulfate to ammonium molar ratio of the selected experiments was approximately 1:1, an inorganic particle composition of the stoichiometry of ammonium bisulfate

($NH_4HSO_4$) was used for the model calculations. This seed type implies partial dissociation of $HSO_4^-$ in aqueous solution into $H^+$ and $SO_4^{2-}$ ions and consequently moderately to strongly acidic conditions. The AIOMFAC-predicted pH ranges from -1.2 to +1.3 for RH ranging from 10 % to 99 % (pH scale using $H^+$ activity on molality basis). Since the experiments and corresponding model predictions were carried out for RH > 75 %, typically ~ 85 % RH, the seed aerosol is assumed to consist of completely deliquesced aqueous solution particles of liquid-like viscosity (Colberg et al., 2004).

When the model calculations were initially run using the same scaling parameters as for the seed-free case, SOA mass concentrations were found to be over-predicted by the model. Possible reasons for the observed model–measurement disagreement will be discussed in the following. During the seeded CLOUD chamber experiments, studying cloud formation from the formed SOA mixed with the seed particles was also of interest. For that purpose, intermittent adiabatic expansions were carried out (chamber pressure was reduced by 220 hPa), during which water vapour supersaturation and (warm) cloud

formation was achieved. Occasionally, the formed clouds led to some precipitation in the chamber, which may have led to subsequent loss of organic aerosol mass and number concentration (reducing the condensation sink). Our interpretation of the importance of these processes are as follows: in the occasional cases where precipitation occurred, the aerosol condensation sink decreased, which in turn may have increased the relative importance of wall losses and may have amplified deviations of





SOA mass concentrations from reaching steady-state conditions. In comparison to the MCM model simulations which neglected precipitation, both of these effects have the potential of decreasing the species concentrations that actually contributed, via gas–particle partitioning, to the observed suspended organic aerosol mass concentration. The corrected isoprene reacted concentration responsible for the measured SOA amount would thus have to be lower than that predicted by

MCM, also lowering the concentrations of the first-generation species modelled by the MCM.

To achieve better model–measurement agreement for SOA mass concentrations formed at varying seed concentrations, a (fitted) scaling parameter of 0.55 was introduced to scale the amounts of semivolatile and low-volatility species predicted by the partitioning model (the one based on branching parameters from the seed-free cases) via scaling of the branching ratios. This scaling factor effectively accounts for the various differences between the seed-free and seeded experiments, such as

differences in the condensation sink, fraction of vapours lost to the wall, transient partitioning effects, etc. An optimal value for the scaling parameter was considered to be one where there is good model–measurement agreement for both ammonium bisulfate and sulfuric acid seeded cases simultaneously. It is noted that the branching ratios could be scaled by any factor between 0.5 and 0.6 to achieve agreement with the experimental data, partially due to the ~ 50 % uncertainty in the AMS measurements that were used to calculate the organic mass, as mentioned in the thesis by Fuchs (2017).

The equilibrium gas–particle partitioning model predictions for SOA mass concentrations resulting at varying ammonium bisulfate mass concentrations are shown in Fig. 6 as black curves with the different line styles corresponding to different isoprene reacted concentrations ranging from 120 to 160 $\mu$g m$^{-3}$. Error bars indicate a 50 % uncertainty in the determined experimental SOA mass concentrations. Figure 6a is of interest for model–measurement comparison since it displays both the effects of variation in isoprene reacted and seed concentration on the SOA mass concentration, while temperature and RH are

kept approximately constant. In Fig. 6b, showing SOA mass yield vs. mass concentration, the ammonium bisulfate concentration dependence is not explicitly visible. The end points of the model curves at the lower SOA mass concentrations in Fig. 6b correspond to 0 $\mu$g m$^{-3}$ seed mass concentration, while the highest values correspond to ~13 $\mu$g m$^{-3}$ ammonium bisulfate. The experimental data in Fig. 6a were obtained over a longer time span of approximately 24 hours since the start of the experiments, with intermittent cloud formation periods during which no data were taken for the comparison here, but

potentially with an effect on the later SOA mass concentration data due to changes in the steady-state dynamics of the chamber (loss of aerosol particles and related condensation sink). This results in potential deviations between actual experimental conditions and those assumed by the model. Measured SOA mass concentrations range between 1.2 and 2.5 $\mu$g m$^{-3}$ for ammonium bisulfate concentrations varying from near zero up to 12 $\mu$g m$^{-3}$. In Fig. 6a, for an isoprene reacted concentration of 130 $\mu$g m$^{-3}$, the predicted SOA mass concentration varies from 1.2 to 1.7 $\mu$g m$^{-3}$ over the range of 0 to 13 $\mu$g m$^{-3}$ of

ammonium bisulfate concentrations shown. The modelled SOA mass concentrations are thus within the uncertainty of the majority of the experimental data, keeping in mind that there remains an additional potential source of error due to vapour wall-loss, which is not included in the error bars of Fig. 6. This is also true for the model calculations at other isoprene reacted levels and their comparison to pertinent experimental data. SOA yield values determined based on the measurements, without





a sophisticated wall-loss and chamber dynamics correction accounted for, vary from 0.4 % to 1.9 % for these conditions with isoprene reacted concentrations in the range from 110 to 165 µg m$^{-3}$ at ~85 % RH and 283 K. The predicted SOA yield values are in the range of 0.8 % to 1.9 % at similar isoprene reacted concentrations.

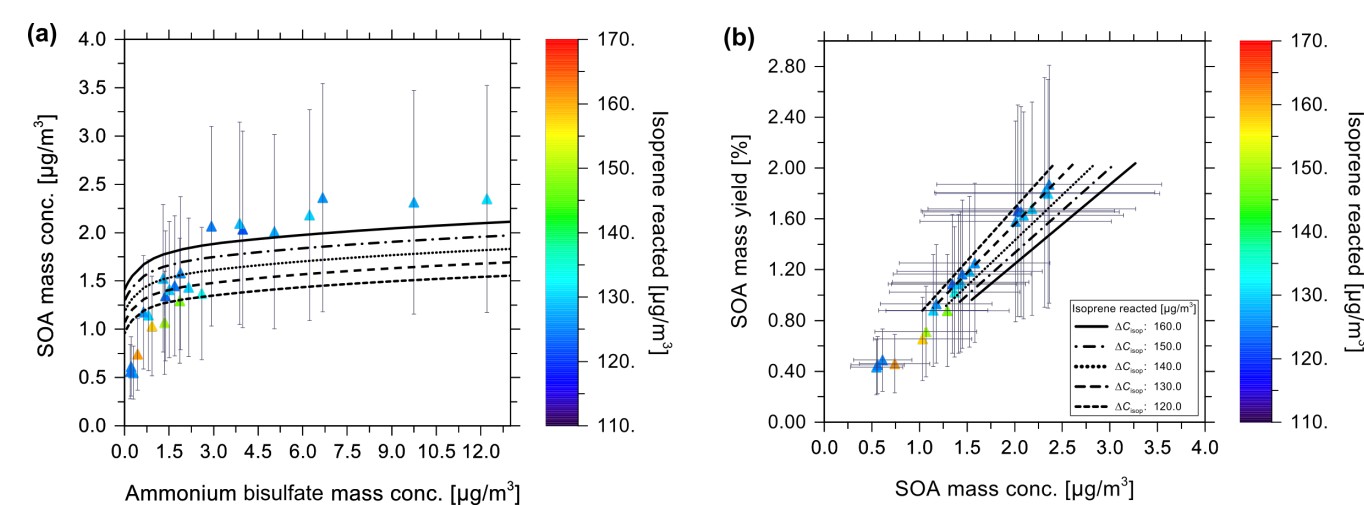

**Figure 6.** Comparison of CLOUD chamber data and gas–particle partitioning model calculations for isoprene SOA at varying inorganic seed concentrations. Experimental conditions: +10 °C, 85 % RH with ammonium bisulfate seed particles. **(a)**
10 Measured and calculated SOA mass concentration vs ammonium bisulfate concentration. Solid triangles: continuous flow data coloured according to the levels of [ISOPRCT] in the chamber (Fuchs, 2017). Black curves: model predictions at different levels of isoprene reacted indicated in the figure key in (b). **(b)** Measured and predicted SOA mass yield vs. SOA mass concentrations for the same range of ammonium bisulfate concentrations as in (a).

### 3.2.2 Effect of aqueous ammonium bisulfate seed: impact of varying water content on predicted composition
Model calculations for gas phase and aqueous solution phase compositions were performed at fixed amounts of total air parcel compositions, except for a variable RH and therefore water content. This affected primarily the aerosol water content, but also the organic aerosol amounts via non-linear gas–particle partitioning feedbacks, while the ammonium bisulfate mass
20 concentration remains constant in the particle phase. The computed equilibrium compositions of the gas phase and the predicted liquid particle phases are shown in Fig. 7, panels (a, b, c). In the stacked bar graphs shown, the emphasis is on the mass concentrations of the organic compounds and the inorganic ions, excluding the water content (although some water is present in the phases). Mass fractions of all components in the liquid phases (including water) are shown in Fig. 7d,e. The



model predicts LLPS to occur between 0 and 83 % RH; here, the two particle phases are a hydrophilic bisulfate-rich, organic-poor phase (phase α) and an organic-rich phase β. Both liquid phases contain substantial inorganic ion amounts; therefore, no complete organic/inorganic phase separation is predicted by the model. We add that the extension of LLPS to low RH is here also a consequence of ignoring any potential crystalline inorganic phase in these computations. The inorganic-rich particle

phase (Fig. 7d) is holding higher water contents compared to the mixed organic–inorganic particle phase (Fig. 7e) at the same RH. Above ~ 84 % RH, both particle phases merge to form a single liquid phase. The gas phase consists of MCM-predicted first generation products while the particle phase mainly consists of surrogate components and the inorganic seed. At 10 °C and an isoprene loading of 130 $\mu g\ m^{-3}$, the model predicts an SOA mass concentration of ~ 0.99 $\mu g\ m^{-3}$, SOA yield of ~ 0.76 % at low RH (~ 1 % to 20 % RH) and therefore low PM water content. At ~ 85 % RH, the aerosol water content is increased, as

is evident from panels (d) and (e) of Fig. 7, affecting the partitioning of all species and resulting in a predicted equilibrium SOA mass concentration of 1.45 $\mu g\ m^{-3}$ and SOA mass yield of ~1.12 %. For an ammonium bisulfate seed concentration of 1.3 $\mu g\ m^{-3}$, water-free organic/inorganic mass ratios of ~ 0.75 at low RH (5 % to 30 % RH) and 1.12 at high RH (~ 85 %) are predicted.

Observations by Kleindienst et al. (2007), using an ammonium sulfate seed (0.05 $\mu g\ m^{-3}$) at 22 °C and an isoprene loading of

1270 $\mu g\ m^{-3}$ in a PTFE chamber, indicated that dark ozonolysis of isoprene produces an estimated SOA yield of 1.0 % at approximately 30 % RH. Using our equilibrium gas–particle partitioning approach to model the SOA formed at varying RH for the case of such an approximately neutral (in terms of pH) ammonium sulfate seed under the same temperature and isoprene loading conditions as used by Kleindienst et al. (2007), we determine SOA mass concentrations of ~ 7 $\mu g\ m^{-3}$ at 35 % RH and ~ 13 $\mu g\ m^{-3}$ at 85 % RH (see Fig. S3). The predicted SOA yields are ~ 0.52 % at 30 % RH, ~ 0.54 % at 35 % RH and ~ 1.03 %

at 85 % RH. For comparison, we modelled the SOA formation for the same conditions, but with ammonium bisulfate instead of ammonium sulfate as seed aerosol. For the Kleindienst et al. (2007) SOA formation conditions, the equilibrium partitioning model predicts the same SOA yields as for the neutral ammonium sulfate seeded case at low and high RH (see Table 3 and Fig. S4). Thus, the gas–particle partitioning mechanism applied to the same surrogate SOA system, in the absence of potential changes to aqueous phase SOA chemistry, leads to very similar SOA mass formed at comparable levels of RH for both a

neutral ammonium sulfate and an acidic ammonium bisulfate seed. For this case with a comparably low seed mass concentration, this finding was perhaps to be expected. Modelled aqueous solution compositions using the neutral ammonium sulfate seed indicate that the amount of the inorganic ions dissolved in the liquid phase can depend on solid–liquid equilibrium (SLE) in the case of ammonium sulfate when model calculations are performed accounting for crystallization in such a case (see Fig. S3).

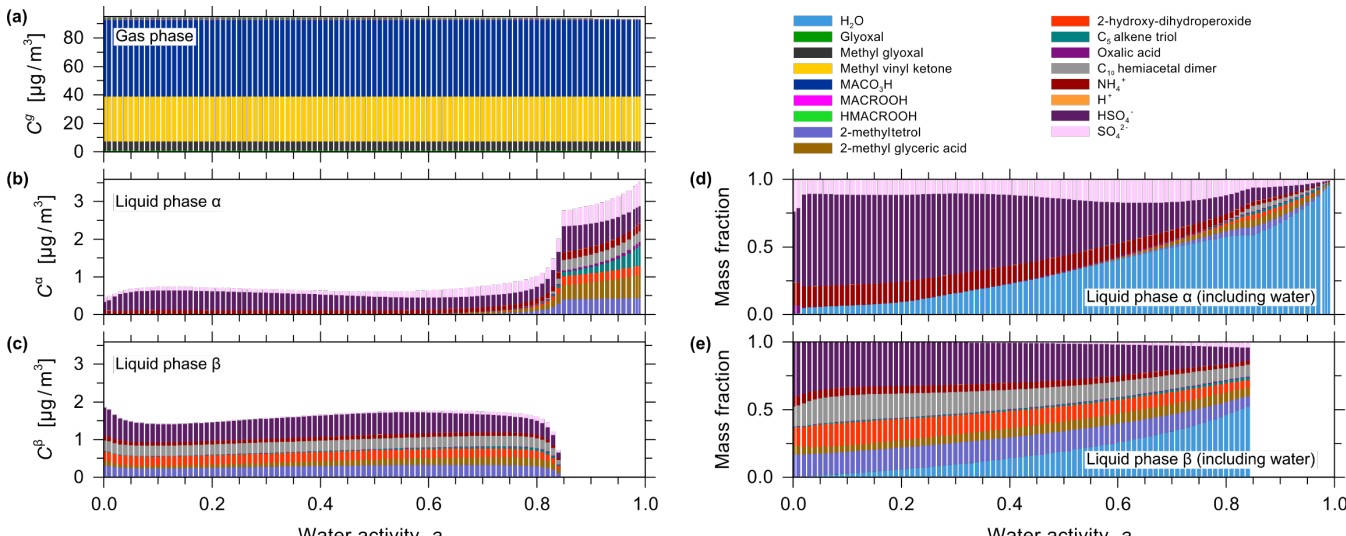

**Figure 7.** Compositions of **(a)** the gas phase and **(b, c)** the liquid aerosol phases (exclusive of present water content) predicted by the bulk equilibrium gas–particle partitioning model. The calculations were carried out for 10 °C, variable water activity (0 % to 99 %), a reacted isoprene concentration of 130 µg m⁻³, and 1.3 µg m⁻³ ammonium bisulfate. The model predicts LLPS to occur between 0 and 84 % RH. **(d, e)** Mass fractions of all species (including water) in the liquid phases.

### 3.2.3 Effect of sulfuric acid seed: impact of varying seed mass concentration

A model–measurement comparison of SOA mass concentration vs. sulfuric acid seed mass concentration is shown in Fig. 8. Experiments with either ammonium bisulfate or sulfuric acid seeds at 10 °C were conducted under similar conditions. Therefore, the branching ratios used to calculate the amounts of higher-generation products in our equilibrium model were scaled by the same correction factor of 0.55 as for the ammonium bisulfate seeded case. At a temperature of 10 °C, 85 % RH and an isoprene reacted amount of ~ 130 µg m⁻³, measured SOA mass concentrations range between 0.5 and 2.4 µg m⁻³ (with uncertainty considered), while the model predicts 1.2 – 2.2 µg m⁻³ for sulfuric acid mass concentrations ranging from 0 up to 5 µg m⁻³ (Fig. 8a, dashed black curve). SOA yield values observed in the chamber vary from 0.3 % to 1.4 % (uncorrected), while the model predicted values are in the range of 0.8 % to 1.4 % for the same SOA mass concentration range (Fig. 8b).





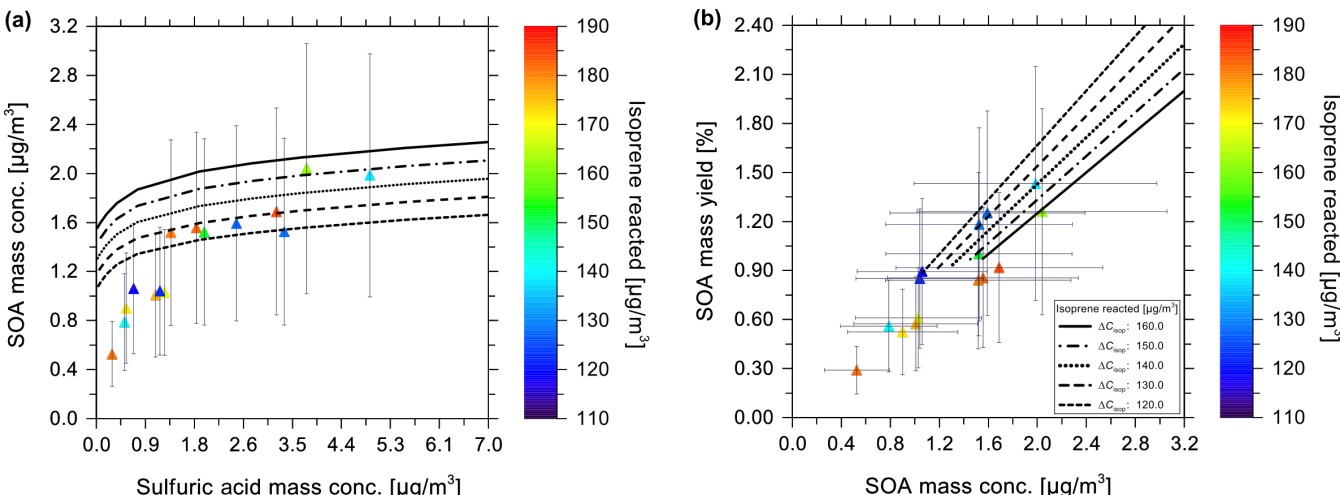

**Figure 8.** Comparison of chamber data and model calculations for isoprene SOA formed at different sulfuric acid seed mass concentrations. Experimental conditions: 10 °C, ~85 % RH with sulfuric acid seed particles; model calculations were carried out for the same conditions and a range of isoprene and seed mass concentrations. **(a)** Measured and calculated SOA mass concentration vs. sulfuric acid mass concentration. Solid triangles: continuous flow chamber data (partially corrected) (Fuchs, 2017) at different ISOPRCT levels (colour scale). Black curves: model predictions for distinct levels of ISOPRCT indicated in (b). **(b)** Measured and predicted SOA mass yields vs. SOA mass concentration.

### 3.2.4 Impact of varying water activity on gas and particle compositions with sulfuric acid seed

Figure 9 shows the predicted equilibrium phase compositions as a function of water activity for the case with an aqueous sulfuric acid seed of 3.7 µg m$^{-3}$. For this amount of inorganic seed, the model predicts LLPS between ~22 % and 82 % water activity, characterized by mixed organic–inorganic phases with both phases containing significant water content. In these calculations, sulfuric acid is assumed to be non-volatile, which is a good assumption at this temperature, except perhaps at the very low RH limit. The AIOMFAC predicted pH ranges from -3.2 to 0.7 for RH ranging from 10 % to 99 %. This indicates that the aqueous particle phase is more acidic than with a bisulfate seed for a comparable RH range, as expected. As shown in Fig. 9, within the LLPS range, the liquid phase $\beta$ (panel c) is a minor phase in terms of absolute mass concentration and it contains organic compounds, like the $C_{10}$ hemiacetal dimer, as well as $H^+$ and $HSO_4^-$ ions, i.e. both liquid phases are predicted to be organic–inorganic mixtures. For an isoprene loading of 130 µg m$^{-3}$, SOA yields in the range of ~ 1.12 % at 60 % RH to 1.76 % at 99 % RH are predicted by the model. SOA mass concentrations are predicted to be in the range from ~2.85 µg m$^{-3}$ at 1 % RH to a minimum of 1.45 µg m$^{-3}$ at 60 % RH followed by a monotonic increase to 2.29 µg m$^{-3}$ at 99 % RH. The highest SOA mass concentrations for the sulfuric acid seeded case are predicted to be found at low RH levels (1 % to 10 % RH) in the



absence of crystalline inorganic phases, because one of the major early generation products, methylglyoxal (comprising ~0.8 % of mass concentration of gas phase isoprene oxidation products), partitions significantly to the particle phase under these conditions. However, the validity of the AIOMFAC-based model predictions at such low water activities and high ionic strengths are rather uncertain. Such results should therefore be interpreted with caution; predictions at > 30 % RH are

considered to be much more reliable.

For comparison of the seed-amount-based yield enhancement, model predictions were made using sulfuric acid seed concentrations comparable to those for the ammonium bisulfate seed case (see Fig. S7). For such a case with 1.3 µg m⁻³ of sulfuric acid seed, the model predicts LLPS between ~ 45 % and 80 % water activity. The predicted pH ranges from -3.2 to 0.7 for RH ranging from 10 % to 99 %. Panels (b) – (e) of Fig. S7 show that both liquid phases are predicted to be organic–

inorganic mixtures. At an isoprene reacted concentration of 130 µg m⁻³, the model predicts SOA yields ranging from ~ 1.32 % (at 1 % RH), ~ 1.10 % (at 35% RH) to 1.71 % (at 99 % RH), with SOA mass concentrations ranging from ~ 1.30 µg m⁻³ (at 60 % RH) to 2.22 µg m⁻³ (at 99 % RH). The differences in the predicted SOA yields compared to those for 3.7 µg m⁻³ sulfuric acid seed (the case shown in Fig. 9) are a result of the different inorganic-to-organic mass ratios involved, the increased amount of seed-contributed water in the case with 3.7 µg m⁻³ seed and the non-linear feedbacks from the coupled LLPS and

gas–particle partitioning.

For a direct comparison of SOA yields of the sulfuric acid seeded case with that of a seed-free case with comparable gas phase chemistry and early generation product yields, the same pseudo-molar yields (using the same scaling factors) were used for a seed-free equilibrium partitioning calculation (see Table 3 and Fig. S6). At an isoprene reacted concentration of 130 µg m⁻³, predicted SOA mass concentrations for this seed-free case are ~ 0.56 µg m⁻³ at 1 % RH, ~ 0.62 µg m⁻³ at 35 % RH and ~ 1.12

µg m⁻³ at 85 % RH, with corresponding SOA yields of ~ 0.43 % at 1 % RH, ~ 0.48 % at 35 % RH, and ~ 0.86 % at 85 % RH. Thus, here a net enhancement of SOA mass concentrations results in higher SOA yields because of a gas–liquid partitioning feedback of semivolatile organics caused by added aerosol water. This hygroscopicity effect is especially prominent when comparing seeded and seed-free calculations at intermediate RH conditions. For example, at 60 % RH and ISOPRCT of 130 µg m⁻³, SOA mass yields of ~ 0.57 % (~ 0.75 µg m⁻³) are predicted without seed while ~ 1.12 % (1.45 µg m⁻³) are predicted

with 3.7 µg m⁻³ sulfuric acid seed. This implies a substantial (non-ideal) mixing effect on the partitioning of SOA due to inorganic seed amounts and inorganics-contributed PM water content when the chemistry involved in the formation of oxidation products remains the same. For comparison and quantification of the non-ideality, we show in Fig. S10 the predicted activity coefficients of the organic compounds for both the seeded and seed-free systems. Figure S10 indicates activity coefficients of the SOA surrogates substantially deviating from unity (expected for ideal mixing behavior), with values both

above and below 1. The activity coefficients are also substantially different when comparing the organic-rich phase (in case of phase separation; system with seed) with the sulfuric-acid rich phase, as expected, as well as when compared to the values in the seed-free system. Also, the water-activity-dependence of the non-ideal mixing in the organic-rich phase in the LLPS





case is not simply resembling the mixing in the seed-free aqueous solution; this is because a non-negligible amount of the ions partitions also to the organic-rich phase.

Several other studies have focused on the role of seed acidity on isoprene SOA formation. Laboratory chamber studies by Jang et al. (2002) measured higher SOA mass formed as a result of catalysed heterogeneous reactions in the presence of inorganic

acids, such as sulfuric acid, for biogenic and anthropogenic carbonyl species. Experimental studies on isoprene ozonolysis in the presence of sulfuric acid seed carried out by Limbeck et al. (2003), Jang et al. (2002) and Czoschke et al. (2003) confirm that acid-catalysed oligomerization reactions result in higher SOA mass compared to that produced in a non-acidic seed particle medium, especially at lower RH. Laboratory studies by Czoschke et al. (2003) were carried out at 24 °C under dry (< 10 % RH) conditions to analyse the impact of acidified sulfate aerosols on SOA formation during isoprene ozonolysis in the presence

of an OH scavenger. Although chemical characterization of the aerosol revealed an enhancement of highly oxidized compounds, the reported SOA yields were low, typically in the range of 0.5 % to 0.8 % by volume.

### 3.2.4  Comparison of RH and seed effects on SOA yield

Table 2 lists measured SOA yields from various studies as well as other properties of these laboratory experiments in

comparison to estimated SOA yields predicted with our partitioning model. The predicted SOA mass yield enhancement, defined as the ratio of the yields computed for two different RH levels, are shown in Table 3 for the seed-free cases as well as seeded cases with ammonium bisulfate or sulfuric acid seeds. Our comparison focuses on the RH effect and differences between the two inorganic seeds on the gas–particle partitioning of semivolatile organic species and water. That is, we are not accounting for possible changes in gas-phase or particle-phase chemical reaction pathways due to changes in RH, aqueous

phase ionic strength and/or acidity. For the given scenarios, the model predicts more substantial SOA yield enhancements when comparing yields at 85 % RH to those at 35 % RH, whereas the enhancements from 1 % RH to 35 % RH are modest or in some cases even below 1.0, indicating a relative yield decrease. For example, the model predicts a SOA yield enhancement of ~ 1.52 at 85 % RH relative to 35 % RH for the case with 1.3 μg m$^{-3}$ ammonium bisulfate seed(SOA/seed mass ratio of 1.1 at 85 % RH), while no enhancement is predicted for the same case when comparing yields at 35 % RH and 1 % RH (yield

ration of 0.97). Using the same pseudo-molar yields (in our surrogate component system) as for the seeded cases to model the SOA formed without an inorganic seed for the purpose of consistent comparisons of predictions made using the same assumptions, we determine a SOA yield enhancement of 1.11 at 35 % RH relative to 1 % RH and a higher SOA yield enhancement of 1.80 at 85 % vs. 35 % RH. The latter enhancement is predominantly because of the added aerosol water amount at higher RH contributed by hygroscopic oxidised organic compounds, indirectly enhancing the partitioning of all

semivolatile organics to the particle phase.



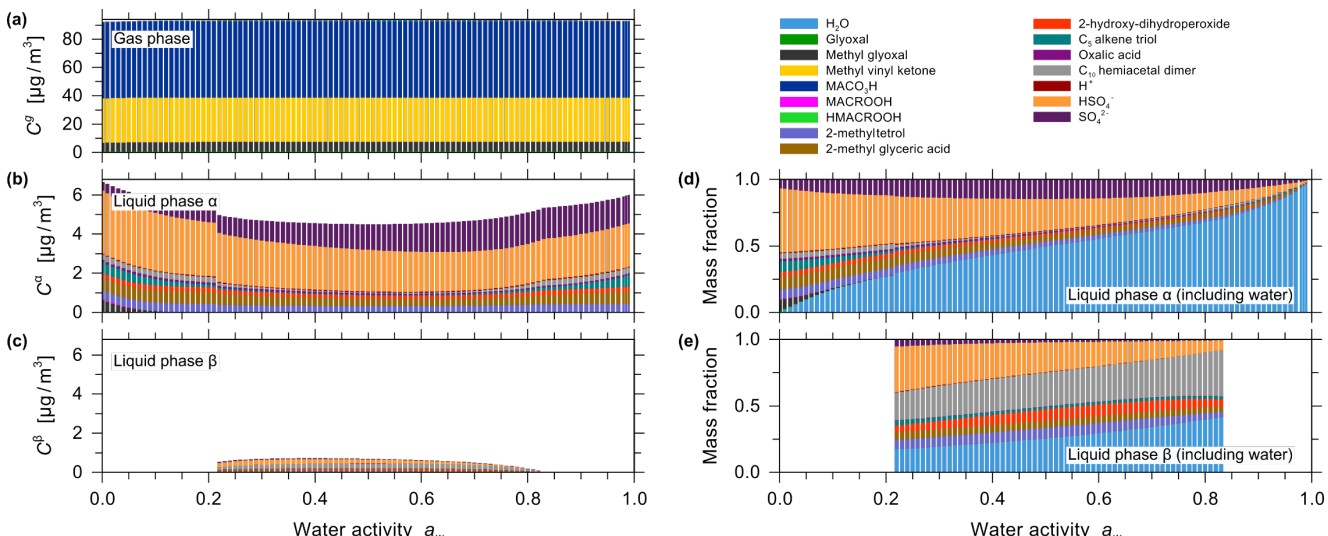

**Figure 9.** Predicted equilibrium gas phase **(a)** and liquid phase **(b, c)** mass concentrations for a reacted isoprene concentration of 130 µg m$^{-3}$ at 10 °C, 3.7 µg m$^{-3}$ sulfuric acid seed and variable water content (1 % to 99 % water activity). The predicted phase compositions shown in (a) to (c) are exclusive of the water content. **(d, e)** Mass fractions in the liquid phases α and β, including water content.

For a sulfuric acid seed concentration of 1.3 µg m$^{-3}$ (see Fig. S7), comparable to the amount of ammonium bisulfate seed in the shown case, slightly higher SOA yield enhancement is predicted in comparison with yield enhancements for a case with 3.7 µg m$^{-3}$ of sulfuric acid seed. The comparison suggests that a sulfuric acid seed, with its effect on the liquid–liquid phase partitioning of water and organics, results in already comparably high values of SOA yield at very low RH levels (1 % RH) and only a small yield enhancement at 85 % RH relative to 35 % RH. The partitioning model further suggests that higher amounts of sulfuric acid seed of about 3.7 µg m$^{-3}$, resulting in an SOA/seed mass ratio of 0.45 at 85 % RH, lead to a slightly lower SOA yield enhancement of 1.06 at 85 % vs. 35 % RH. Comparing the seed-free and the three seeded cases at 10 ºC listed in Table 3, all computed with the same set of pseudo-molar yields and scaling parameters, shows that the highest SOA yields (but not yield enhancements) are obtained for the cases with sulfuric acid seed, while the seed-free case shows the lowest yields. This is mainly due to favourable partitioning of SVOCs in the system into the particle phase as a result of the substantial hygroscopicity of sulfuric acid, particularly so already at 35 % RH. Irrespective of the seed type, presence of a substantial aqueous inorganic seed mass concentration is predicted to enhance the SOA yield (see also Figs. 7a, 8a), especially at high RH, and to influence the RH-dependent SOA yield enhancement.



Comparing the modelled cases for 130 µg m⁻³ of isoprene reacted at a temperature of 25 ºC with their equivalent cases at 10 ºC, a temperature effect on the predicted SOA mass yield enhancement is indicated. For the seeded cases at 25 ºC, the higher enhancement of ~1.39 is found for the ammonium bisulfate seed; however, this enhancement is lower than that of 1.52 for the equivalent case at 10 ºC.

**Table 2.** Summary of the experimental conditions, measurement-based SOA mass yields and predicted yields by the equilibrium gas–particle partitioning model (this study) from previous and current ozone-initiated isoprene oxidation studies at low NO$_x$ conditions. Note: AS – ammonium sulfate, ABS – ammonium bisulfate, SA – sulfuric acid.

| Study, Reference | Oxidant | Seed | Wall loss corrected? | RH (%) | $T$ (°C) | Isoprene reacted (µg m⁻³) | Measured SOA yield (%) | Model-estimated SOA yield [a] (%) |
|---|---|---|---|---|---|---|---|---|
| Sato et al. (2013) | Ozone | seed-free | yes | 0.01 | 25 | 1378 – 5484 [b] | 0.30 – 2.30 | 0.84 – 1.28 |
| Clark et al. (2016) | Ozone | seed-free | yes | < 1 | 5 | 278 [c] | 9.00 | 1.25 |
| Nakayama et al. (2018) | Ozone | seed-free | no | < 1 | 25 | 10578 [b] | 1.99 | 1.63 |
| Fuchs (2017) | Ozone | seed-free | partially [d] | 35 | 5 | 125 [c] | 1.20 | 1.22 |
| Fuchs (2017) | Ozone | seed-free | partially | 85 | 5 | 125 [c] | 2.00 | 2.18 |
| Kleindienst et al. (2007) | Ozone | AS (0.05 µg m⁻³) | yes | 30 | 22 | 1270 [b] | 1.00 | 0.52 |
| Fuchs (2017) | Ozone | ABS (1.3 µg m⁻³) | partially | 85 | 10 | 130 [c] | 1.00 | 1.12 |
| Fuchs (2017) | Ozone | SA (3.7 µg m⁻³) | partially | 85 | 10 | 130 [c] | 1.50 | 1.29 |
| Chen et al. (2011) | OH | seed-free | no | 40 | 25 | 177 [c] | 2.40 | 2.40 |

[a] This study. See details in the text regarding different pseudo-molar yields and surrogate formation scaling parameters ($\beta$) used for seeded vs. seed-free cases.

[b] With OH scavenger

[c] Without OH scavenger

[d] A simple wall-loss correction was applied to the measurement data for use in this study. The correction did not account for dynamic variations in wall loss due to changes in the volatility distribution of gases or changes to the seed concentration and condensation sink (e.g. after precipitation events).

Comparing the SOA yield enhancements for the ammonium bisulfate seeded case with the sulfuric acid seeded cases, the relatively high yield enhancement with an ammonium bisulfate seed is mainly due to the lower SOA mass yield at 35 % RH relative to 85 % RH and relative to the sulfuric acid seeded cases at 35 % RH. At the same time, the SOA yield values at 85 % RH for the case with 1.3 µg m⁻³ ammonium bisulfate seed concentration are comparable with the sulfuric acid seeded case at 1.3 µg m⁻³. A comparison of organic aerosol mass concentrations predicted over a wider range of RH is provided in Fig. S8 for cases with sulfuric acid or ammonium bisulfate seed.



**Table 3.** Predicted SOA mass yields and relative yield enhancements. Abbreviations for inorganic seeds: ABS – ammonium bisulfate, SA – sulfuric acid, AS – ammonium sulfate.

| Case | $T$ (°C) | Isoprene reacted ($\mu g\ m^{-3}$) | at 1 % RH [a] (%) | at 35 % RH [a] (%) | at 85 % RH [a] (%) | (35 / 1) [b] | (85 / 35) [b] |
|---|---|---|---|---|---|---|---|
| Seed-free | 5 | 125 | 0.98 | 1.22 | 2.18 | 1.24 | 1.79 |
| ABS seed (1.3 $\mu g\ m^{-3}$) | 10 | 130 | 0.76 | 0.73 | 1.12 | 0.97 | 1.52 |
| SA seed (3.7 $\mu g\ m^{-3}$) | 10 | 130 | 2.19 | 1.24 | 1.29 | 0.56 | 1.04 |
| SA seed (1.3 $\mu g\ m^{-3}$) [c] | 10 | 130 | 1.32 | 1.10 | 1.17 | 0.84 | 1.06 |
| Seed-free (see Fig. S6) [d] | 10 | 130 | 0.43 | 0.48 | 0.86 | 1.11 | 1.80 |
| AS seed (0.05 $\mu g\ m^{-3}$) [e] | 22 | 1270 | 0.46 | 0.54 | 1.03 | 1.16 | 1.91 |
| ABS seed (0.05 $\mu g\ m^{-3}$) [f] | 22 | 1270 | 0.46 | 0.54 | 1.03 | 1.16 | 1.92 |
| Seed-free | 25 | 130 | 0.56 | 0.61 | 0.84 | 1.08 | 1.39 |
| ABS seed (1.3 $\mu g\ m^{-3}$) | 25 | 130 | 0.44 | 0.42 | 0.59 | 0.95 | 1.39 |
| SA seed (1.3 $\mu g\ m^{-3}$) | 25 | 130 | 1.11 | 0.69 | 0.67 | 0.62 | 0.97 |

[a] SOA mass yield (%) predicted by our AIOMFAC-based gas–particle equilibrium model with inputs from MCM simulations.

[b] SOA mass yield enhancement at indicated higher to lower RH levels.

[c] Using similar amounts of SA seed (1.3 $\mu g\ m^{-3}$) as ABS seed for yield enhancement comparison (see SI, Fig. S7).

[d] Using the same pseudo-molar yields for this seed-free calculation as for the CLOUD 9 seeded cases (ABS seed and SA seed) for comparison. Note that the other seed-free cases (e.g. row 1) use the pseudo-molar yields as determined from the CLOUD 10 seed-free experiments. Therefore, those use different scaling parameters than the case indicated by superscript d.

[e] See SI, section 3.3 and Fig. S3 of the model predictions for comparison with the study by Kleindienst et al. (2007).

[f] See SI, section 3.4 and Fig. S4 of the model predictions for comparison with the study by Kleindienst et al. (2007).

## 4   Conclusions

In the present study, the growth of particulate matter derived from the isoprene ozonolysis system at varying RH levels was modelled to study the effect of thermodynamic equilibrium partitioning on SOA mass concentrations and yields for different

cases: (i) without seed particles, (ii) with ammonium bisulfate and (iii) sulfuric acid seed particles at low $NO_x$ conditions. The gas–particle partitioning model setup is based on a combination of MCM-derived gas phase concentration data, vapour pressures predicted by EVAPORATION and non-ideal thermodynamic mixing by AIOMFAC. The model considers the possibility of liquid–liquid phase separation and, in case of ammonium sulfate seed, also solid–liquid equilibrium. Experimental chamber data of SOA mass concentrations, MCM-predicted isoprene reacted levels and molar-based yields of

MCM-modelled, stable first/early-generation species were used to adjust a set of scaling parameters for the characterization of the isoprene oxidation system in our model. The SOA mass concentration was modelled at varying isoprene loading levels. The effect of thermodynamic equilibrium partitioning at different RH levels on SOA yields and aerosol phase compositions from isoprene ozonolysis have been studied. At equilibrium, the compositions of the aerosol particles are shown to be



dependent on the temperature, amount of isoprene reacted that forms the oxidised species, amount of inorganic seed present and the ambient RH (exerting control on aerosol water content).

For the seed-free case, taken as a reference case, the modelled SOA mass concentrations were adjusted to achieve agreement with the SOA mass measured by dark ozonolysis experiments in the CLOUD chamber at $T = 5\ °C$ for varying isoprene loading levels. The measurements conducted both at moderately low and at high levels of RH in the CLOUD chamber are an improvement over experiments conducted in the past solely under dry conditions. This is certainly useful for this study as they aid in constraining and accurately modelling the hygroscopicity and gas–particle partitioning of largely semivolatile SOA. Effects of temperature, chamber experiment types, and the use of scaling parameters specific to the CLOUD chamber, have been discussed as reasons for variations in SOA yields compared to other studies. As a result of the temperature effect, the model predicts an increase in the SOA yields at low temperatures as the SVOCs partition significantly to the PM phase. No LLPS is predicted by the model at all levels of RH in the absence of an inorganic seed.

Presence of ammonium bisulfate seed particles is shown to influence the concentration and liquid phase compositions at equilibrium since the particles are predicted to undergo LLPS into a hydrophilic bisulfate-rich phase and a less hydrophilic organic-rich phase, with substantial water contents in both liquid phases. The model predicts SLE for the neutral ammonium-sulfate-seeded case at intermediate to low RH (when allowed in the calculations). Thus, the specific seed type has the potential to play a vital role in the gas–particle and liquid–liquid partitioning of the organic surrogate species – aside from the known role of seed acidity on particle phase chemical reactions (which was not considered in this work).

For the sulfuric acid seeded case, the same molar-based yields as for the ammonium bisulfate case were used since the experimental chamber conditions were similar in both seeded cases. The model predicts a LLPS range with a small mixed organic plus bisulfate-rich phase in coexistence with another mixed organic plus sulfuric-acid-rich phase from low to high levels of RH. Model predictions for comparison of the measured yields from the current study and from previous studies from the scientific literature suggest that the extent of LLPS is dependent on the seed amount (wider LLPS range for higher seed amounts) and temperature (lower temperatures favouring phase separation). Therefore, while LLPS is predicted in a certain RH range for the systems containing either ammonium bisulfate or sulfuric acid, the phase compositions are far from a complete organic/inorganic un-mixing (an extreme outcome sometimes observed for other systems), indicating the hygroscopic nature of isoprene SOA and its partial miscibility with aqueous ionic solutions. Comparison with seed-free experiments at similar thermodynamic conditions and pseudo-molar yields for all system components highlights the role of an inorganic seed in enhancing both SOA mass concentrations and yields.

Based on these modelling results, we conclude that SOA from isoprene oxidation is likely to partition into both aqueous and (oxidized) organic-rich aerosol phases, contributing to water uptake and phase polarity – and thereby influencing the gas–particle partitioning of other organic and inorganic species, including ammonia (Pye et al., 2018). The modelled SOA yields and mass concentrations indicate the influence of thermodynamic partitioning on increasing SOA yields with increasing RH.





Atmospheric implications as a result of the RH effect involve an SOA yield enhancement when an inorganic seed is present, regardless of the seed type (ammonium bisulfate, ammonium sulfate or sulfuric acid) in comparison with seed-free conditions at the same RH and isoprene reacted levels. This is because the water content contributed by the seed is diluting the liquid phases and driving the gas–particle partitioning of semi-volatile organic compounds towards particle phase enhancement at

elevated RH.

Since isoprene is one of the major identified precursors for biogenic SOA, accurate representations of SOA in terms of volatility distributions, mass yields and/or sets of molecular surrogate compounds, are imperative to aid in better representing the potential of isoprene to form condensable products. These products contribute to the SOA budget on regional scale, they may modify the CCN properties of different aerosol size modes, and their quantification is essential for accurately predicting

the water uptake by particulate matter as well as related aerosol size distribution changes in air quality and climate models. The CLOUD chamber experiments on isoprene oxidation, with a subset of its data used in this study, also point to the importance of future developments to better account for dynamic vapour and particle wall-losses, condensation sink dynamics, and flow chamber steady-state behaviour. Tools like the AIOMFAC-based phase separation model coupled with chemical kinetics and flow simulations could be used in the future to better constrain the effect of non-ideality and aerosol water content

on the overall chamber dynamics and related interpretation of measurement data.

*Code and data availability*. The AIOMFAC model can be run at https://aiomfac.lab.mcgill.ca for single liquid phases; code is available at https://github.com/andizuend/AIOMFAC. The MCM model is available at http://mcm.leeds.ac.uk/MCM/.

Model system composition data is provided in the supplement. Data underlying the shown figures and related output from the gas–particle partitioning model are available from the following Zenodo online repository: https://doi.org/10.5281/zenodo.4628342.

*Author contributions.* DAA, CRH and AZ conceptualized the project. DAA and AZ performed thermodynamic modelling

and created visualizations. CRH, CH and MGB carried out MCM simulations. CH, CRH, IEH, ME, SMP, AA, JD, SE, VM, UM, MR, YS, RW, JK, AH, NMD, RV and UB prepared and conducted CLOUD chamber experiments at CERN, provided instrument support and calibration, collected and processed measurement data. AZ, MGB, NMD, RV and UB acquired the financial support. DAA, CH, CRH, MGB, IEH, AH, NMD, RV and UB discussed and interpreted scientific findings. DAA, AZ, MGB and CRH co-wrote the manuscript, with contributions by IEH, UB, SMP and ME.

*Competing interests.* The authors declare that they have no conflict of interest.



*Acknowledgements*. We thank CERN for supporting CLOUD with technical and financial resources. This work was supported by the Natural Sciences and Engineering Research Council of Canada (NSERC, grant RGPIN/04315-2014) and by Quebec's FRQNT (grant 2015-NC-181620) as well as the Swiss National Science Foundation (grants 200021_169090, 200020_172602 and 20FI20_159851). This work was also supported by the US National Science Foundation (grants AGS-1452317, AGS-5 1801280, AGS1801574, AGS1801897).

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
