# Peer review of "Modelling the gas-particle partitioning and water uptake of isoprene-derived secondary organic aerosol at high and low relative humidity"

_Atmospheric Chemistry and Physics, 2021_

## Referee Comment (RC1)

Review of the manuscript acp-2021-251 "Modelling the gas–particle partitioning and water uptake of isoprene-derived secondary organic aerosol at high and low relative humidity" by Amaladhasan et al.

**General comment:**

In the presented study, the authors perform model simulations of isoprene oxidation and secondary aerosol formation during a wide range of atmospheric relevant conditions (RH, temperature, seed aerosol type). The model system consist of the master chemical mechanism isoprene gas-phase chemistry and an equilibrium SOA partitioning model based on the AIOMFAC activity coefficient model for mixture non-ideality and the EVAPORATION model for pure compound liquid saturation vapor pressure estimates. The paper is in large parts well written, but it is not straightforward to understand all methods. Especially the SOA yield parameterization and SVOC partitioning will need some clarification and justifications. In addition, the atmospheric relevance of the present work can be highlighted more. After such improvements and careful considerations of my review comments, I think that the paper can be accepted for publication in Atmospheric Chemistry and Physics.

**Specific comments:**

Last sentence in Abstract. Consider to add some conclusions about the results. What are the main findings from the present study?

Page 5, line 9 "The model predictions are compared to isoprene ozonolysis chamber experiments …".
Generally I think it is hard to differentiate between what results that are model fitting to the CLOUD experiment and which model results that actually are independently evaluate against the SOA mass concentration observations from CLOUD.

Page 6, lines 2-5: "The total organic mass concentration, resulting from the gas-phase and aqueous-phase oxidation chemistry under dynamic gas–particle partitioning, was derived from the SMPS measurements of the particle number–size distribution, assuming an average aerosol mass density of 1.3 g cm−3 (Fuchs, 2017). The average aerosol mass density was calculated based on the HR-ToF-AMS-determined mass fractions by using a parameterization by Kuwata et al. (2011), which is based on the elemental oxygen-to-carbon ratio (O:C) and the hydrogen-to-carbon ratio (H:C)."
From this description, it is not clear to me if you used a SOA mass density of 1.3 or densities derived from the SOA elemental composition.

Page 9, lines 1-5: "Whether a component resides in the gas phase, the condensed phase or partitions significantly between both is, in principle, irrelevant at input, yet can be exploited to select mainly low-volatility and semi volatile components that substantially contribute to SOA mass under given environmental conditions."

I don't understand what you want to say with this statement. How can this be irrelevant for SOA formation?

Page 9, lines 5-7 "A state-of-the-art chemical mechanism for gas phase reactions, the Master Chemical Mechanism (MCM), version 3.3.1 (Jenkin et al., 2015), available online: http://mcm.leeds.ac.uk/MCM, was used to account for the reactions of isoprene with ozone as well as OH radicals formed during the reaction process."

I guess one purpose of using MCM is to estimate the fraction of isoprene that is oxidized by OH in the experiments. This will be very relevant for the SOA formation. I would like to see some information in the manuscript about the relative importance of OH vs O3 oxidation of isoprene during the CLOUD experiments. The MCM model simulations should give this information.

Page 9, line 12-14: "The major stable products predicted by the MCM for the first/early generations of oxidation are compounds of low molecular mass with relatively low O:C ratios and high vapour pressures (VOC to IVOC class compounds)."

I suggest that you add numbers on how large fraction of the total modelled stable oxidation products that go via these intermediate early-generation oxidation products. Otherwise you wonder why you only selected these specific compounds.

Page 9, lines 16-18: In order to validate the amounts of ozone, OH and "isoprene reacted" predicted by the MCM, conditions and measurements from the CLOUD 10 and CLOUD 9 chamber experiments, reported by Fuchs (2017), were used for a comparison with observations, further discussed in the following.

Please consider to reformulate this sentence. It is not easy to understand what you mean with "isoprene reacted". Do you mean amount of isoprene oxidized by O3 and OH?

Page 9, lines 20-21: "Additional components of semi-volatile and low-volatility nature formed via multi-generation oxidation, which are not fully considered by MCM, need to be included in the partitioning model as part of the isoprene system."

1. I don't question that MCM do not predict all semi-volatile and low-volatility products that actually contributes to SOA mass, but have you actually tested to simulate the SOA formation from the isoprene oxidation products generated in MCM and compared the results with the observations in CLOUD?
2. I think this should be an easy test to do. I know that the MCM scheme for isoprene reacted with OH generate some semi- to low-volatility products that do contributes to SOA mass and yields comparable with experimental yields (see e.g. Xavier et al., Atmos. Chem. Phys., 19, 13741–13758, 2019).
3. For pure isoprene + O3 oxidation you probably do not form any semi- and low-volatility products that contribute to SOA mass formation but since you have secondary production of OH during the CLOUD experiments and in the model you should also form some low-volatility products from the MCM gas-phase chemistry scheme.  On example is HMACROOH in Fig. 1 and 2.
4. When reading this I also ask the same question again: How large fraction of the reacted isoprene were via isoprene + OH?

Page 9, lines 23-28 "The selected surrogate components are shown in Fig. 1. Oxalic acid along with compounds such as 2-methyltetrol, 2-hydroxy-dihydroperoxide, 2-methylglyceric acid and a C5-alkene triol are among the main semi-volatile and low-volatility products that are expected to form after oxidation (by ozone and by OH) of the early generation compounds under low NOx conditions. The selected C10 hemiacetal dimer is a compound representing a whole class of potential dimer and oligomer compounds formed by accretion reactions from the above-mentioned five higher generation products."

1.  I miss a clear description about why these specific surrogate compounds were selected. At least add some proper references. You want to know how atmospheric relevant the selected SOA composition is.
2.  I also miss information about the modelled and observed SOA elemental composition (O:C and H:C). Clearly these results exists and will provide at least some indication about how well the surrogate SOA species parameterization is able to represent the SOA properties e.g. hygroscopicity during the different CLOUD experiments.

Page 11, lines 19-21: "Surratt et al. (2010) proposed a chemical mechanism via the reactive intermediate epoxydiols (IEPOX) pathway leading to the formation of 2-methyltetrols and C5-alkene triols for the oxidation of isoprene by OH."
The MCM chemistry represent formation of epoxydiols. Why did you not include IEPOX as an early generation precursor to 2-methyltetrols and C5-alkene triols? This would make the modelled SOA mass predictions more explicit.

Page 11, lines 24-25: "Multigeneration surrogate products from the MCM predicted yields of early-generation products are used to ensure that no carbon mass is unaccounted for."
I don't understand this sentence completely. Do you mean that all (100 %) of all isoprene - oxidation pathways will go via the 6 selected early-generation oxidation products in Fig. 1 and hence, no carbon mass is unaccounted for?

Page 13, lines 17-19 "Molar yields of MCM-predicted products used in the model for the CLOUD 10 (seedfree) case were different based on the individual concentration of components present at the time when the chamber stabilized for the seed-free experiment."
This sentence I do not understand. Please clarify what you mean.

Page 13, lines 27-30: "Details of the MCM box model approach in Fuchs (2017) are as follows: the box model used inputs for the isoprene concentration, ozone concentration, condensation sink (from SMPS data), inorganic seed concentration, rainout rate constant (quantifies the rate of formation of cloud droplets and the associated mass loss in the cloud), dilution rate constant, aerosol wall loss rate constant, RH, reaction rate constants of isoprene with the OH and O3 oxidants, the hygroscopic growth factors …"
Is this also the approach you used? With isoprene and ozone concentration, do you mean the steady state concentrations in the chamber or the concentrations in the inflow?

Page 15, lines 23-25: "In our simple model, the scaling parameters (branching ratios) are adjustable and are determined iteratively by model–measurement comparison. After the gas–particle partitioning model was used to calculate the equilibrium SOA mass concentration formed at varying levels of reacted isoprene, covering the range observed in the CLOUD chamber experiments,"

1. Ok, so in a way you produce a 5-species SOA yield parameterization which match the observed SOA mass, or?
2. I don't understand this completely. Was this not the same experiments which you used to fit the branching ratios? I have hard to differentiate between what actually was statistic fitting of the branching ratios and what is standalone modelling without tuning.
3. Which input parameters did you use to constrain the branching ratios and how was it done in practice. I guess you can find many different combinations of the beta parameters that give more or less the same SOA yield curves.

Page 24, line 1: "However, in the absence of OH radicals and light in the CLOUD experiments …"
But OH was formed during the ozonolysis of isoprene.

Page 24, lines 4-6: "For consistency, the same molar yields were used for all seeded cases considered since the differences among the experimental conditions during the seeded experiments were relatively minor (yet different compared to seed-free)."
I don't understand completely. Did you use the same molar yields for the early products as in the seed-free case or different yields? I thought that a novelty with the model approach was that you base the yield calculations of the near-explicit MCM chemistry.
If you do not allow the early generation yields to vary according to the MCM chemistry I don't see the point in calculating the beta parameters based on the specific selected MCM compounds. Can you please clarify this?

Page 24, lines 8-14: "Hence, the gas–particle partitioning model was modified to account for RH-dependent SOA and water partitioning in the presence of an aqueous phase of variable inorganic seed concentration. We note that the measured steady-state flow chamber aerosol mass concentration is affected by the dynamic interplay between the different vapour and particle loss mechanisms: wall-loss, loss by chamber outflow, and the aerosol condensation sink. The latter depends on the particle size distribution and therefore also on humidity-dependent water uptake by seed particles and organic partitioning."

1. It is unclear to me what you actually mean with this. How was the model modified? Did you change the beta values?
2. You note the importance of the dynamics interplay between the vapor on the walls and the condensation onto the particles, but do you consider it somehow in the model?

Page 24, lines 30-21: ". Occasionally, the formed clouds led to some precipitation in the chamber, which may have led to subsequent loss of organic aerosol mass and number concentration (reducing the condensation sink)."
Yes but was not steady state reach in-between the precipitation events? Can you provide some experimental data (e.g. from the SMPS) that justify the suggested decreasing condensation sink? If steady state is not reached, it must be hard to constrain an equilibrium model based on the observations.

Figure 6. Is the increasing SOA mass as a function of the inorganic seed aerosol a result of increased condensation sink, changed chemistry or both? Can you differentiate between these effects?

Page 27, line 14: "Observations by Kleindienst et al. (2007), using an ammonium sulfate seed (0.05 μg m$^{-3}$)" This is a very low concentration of AS seed. Is the value correct?

Page 28, lines 16-17: "SOA yield values observed in the chamber vary from 0.3 % to 1.4 % (uncorrected), while the model predicted values are in the range of 0.8 % to 1.4 % for the same SOA mass concentration range (Fig. 8b)."
Can you explain the difference between the modelled and observed yields at low SA seed concentrations? How would the dynamics i.e. condensation sink affect the results?

Page 29, lines 19-20: "That is, we are not accounting for possible changes in gas-phase or particle-phase chemical reaction pathways due to changes in RH, aqueous phase ionic strength and/or acidity." and maybe even more important the dynamic e.g. condensation vs wall losses of condensable vapors or?

Page 36, lines 13-15: "Tools like the AIOMFAC-based phase separation model coupled with chemical kinetics and flow simulations could be used in the future to better constrain the effect of non-ideality and aerosol water content on the overall chamber dynamics and related interpretation of measurement data."
I fully agree with you.

---

## Author Comment (AC1)

**Responses to referee #1**

We would like to thank the referee for her/his comments and efforts in providing this review. Below, we repeat each comment by this referee (in blue font) followed by our response, and we indicate related changes made to a revised version of the manuscript. Page and line numbers stated are those from the original manuscript.

General comment:

In the presented study, the authors perform model simulations of isoprene oxidation and secondary aerosol formation during a wide range of atmospheric relevant conditions (RH, temperature, seed aerosol type). The model system consist of the master chemical mechanism isoprene gas-phase chemistry and an equilibrium SOA partitioning model based on the AIOMFAC activity coefficient model for mixture non-ideality and the EVAPORATION model for pure compound liquid saturation vapor pressure estimates. The paper is in large parts well written, but it is not straightforward to understand all methods. Especially the SOA yield parameterization and SVOC partitioning will need some clarification and justifications. In addition, the atmospheric relevance of the present work can be highlighted more. After such improvements and careful considerations of my review comments, I think that the paper can be accepted for publication in Atmospheric Chemistry and Physics.

Specific comments:

1) Last sentence in Abstract. Consider to add some conclusions about the results. What are the main findings from the present study?

**Authors' response:** Two additional sentences have been added.

**Changes to manuscript:** "Those studies were conducted at RH levels at or below 40 % with reported SOA mass yields ranging from 0.3 up to 9.0 %, indicating considerable variations. A robust feature of our associated gas–particle partitioning calculations covering the whole RH range, is the predicted enhancement of SOA yield at high RH (> 80 %) compared to low RH (dry) conditions, which is explained by the effect of particle water uptake and its impact on the equilibrium partitioning of all components."

2) Page 5, line 9 "The model predictions are compared to isoprene ozonolysis chamber experiments …".

Generally I think it is hard to differentiate between what results that are model fitting to the CLOUD experiment and which model results that actually are independently evaluate against the SOA mass concentration observations from CLOUD.

**Authors' response:** The above sentence has been rephrased to clarify that some CLOUD data was used to fit model parameters. Other revisions in Sect. 2 also serve better clarification of this model–measurement comparison vs. tuning aspect.

**Changes to manuscript:** page 5, lines 11, rephrased:
"Selected data sets from CLOUD experiments, primarily those for seed-free ozone-initiated oxidation of isoprene, were also used to tune adjustable model parameters to match measurements taken at low and high relative humidity levels; see details described in Sect. 2."

3) Page 6, lines 2-5: "The total organic mass concentration, resulting from the gas-phase and aqueous-phase oxidation chemistry under dynamic gas–particle partitioning, was derived from the SMPS

measurements of the particle number–size distribution, assuming an average aerosol mass density of 1.3 g cm−3 (Fuchs, 2017). The average aerosol mass density was calculated based on the HR-ToF-AMS-determined mass fractions by using a parameterization by Kuwata et al. (2011), which is based on the elemental oxygen-to-carbon ratio (O:C) and the hydrogen-to-carbon ratio (H:C)."

From this description, it is not clear to me if you used a SOA mass density of 1.3 or densities derived from the SOA elemental composition.

**Authors' response**: The above sentences have been rephrased to indicate that an average SOA mass density of 1.3 g cm$^{-3}$ was used. This average SOA density value was estimated from the measured elemental O:C and H:C values.

**Changes to manuscript;** page 6, lines 4 – 8, rephrased:
"The total organic mass concentration, resulting from the gas-phase and aqueous-phase oxidation chemistry under dynamic gas–particle partitioning, was derived from the SMPS measurements of the particle number–size distribution, assuming spherical particles and using an average aerosol mass density of 1.3 g cm$^{-3}$ (Fuchs, 2017). The average organic aerosol mass density was calculated from average composition using a parameterization by Kuwata et al. (2012), which is based on the measured (by HR-ToF-AMS) elemental oxygen-to-carbon (O:C) and hydrogen-to-carbon (H:C) ratios.

4) Page 9, lines 1-5: "Whether a component resides in the gas phase, the condensed phase or partitions significantly between both is, in principle, irrelevant at input, yet can be exploited to select mainly low-volatility and semi volatile components that substantially contribute to SOA mass under given environmental conditions." I don't understand what you want to say with this statement. How can this be irrelevant for SOA formation?

**Authors' response**: We agree, this statement was not clear; we have modified it. The original sentence was meant to indicate that the final phase partitioning of a component is, in principle, irrelevant at the stage of input into the gas–particle partitioning model, since the model will use the total molar concentration in a unit volume of air to determine the equilibrium partitioning of the component – and this notion would work if a nearly complete set of components were used to compute the partitioning. However, because only a selection of surrogate compounds is used, knowing the approximate volatility of candidate components is useful for that initial selection process. We have rephrased the original sentence and expand on the volatility and functional groups (hygroscopicity) properties of importance in our approach.

**Changes to manuscript;** page 9, lines 2 – 5, rephrased:
"The use of a set of surrogate components typically means that the actual system of oxidation products is highly simplified in terms of number and chemical classes of components. Some components will partition mostly to the gas phase and others to a small or large extent to the condensed phase. From the perspective of the gas–particle partitioning physics common to the volatility basis set (VBS) as well as surrogate-based approaches, it is necessary and sufficient to cover different volatility classes by at least one surrogate species. In such a framework, it is then important to approximately match the volatility distribution of the surrogates contributing to the SOA mass, which is achieved by tuning surrogate yields to match observations (as far as the range of measured SOA concentrations allow). As such, knowing the pure-component volatilities of potential surrogate components matters; we exploit this by selecting a set of low-volatility and semi-volatile surrogate components that will likely contribute to the SOA mass under the given environmental conditions. In addition, higher volatility components, directly predicted by a gas-phase chemical mechanism, are part of our system and enable establishing a scalable link

between the yields of those products and the selected lower-volatility surrogate compounds. Aside from surrogate compound volatilities, it is also important to match approximately the distribution of functional groups and molecular sizes, such that the resulting SOA hygroscopicity is captured by the AIOMFAC model. This aspect is considered by tuning the surrogate yields to observations taken at substantially different RH levels."

5) Page 9, lines 5-7 "A state-of-the-art chemical mechanism for gas phase reactions, the Master Chemical Mechanism (MCM), version 3.3.1 (Jenkin et al., 2015), available online: http://mcm.leeds.ac.uk/MCM, was used to account for the reactions of isoprene with ozone as well as OH radicals formed during the reaction process."

I guess one purpose of using MCM is to estimate the fraction of isoprene that is oxidized by OH in the experiments. This will be very relevant for the SOA formation. I would like to see some information in the manuscript about the relative importance of OH vs O3 oxidation of isoprene during the CLOUD experiments. The MCM model simulations should give this information.

**Authors' response**:   The MCM model output suggests that about 30 % of the isoprene oxidation in the dark chamber was due to OH. The remaining approximately 70 % of the isoprene oxidation was due to $O_3$ for low-$NO_x$ conditions.  This is an aspect that was studied by Fuchs (2017), both to confirm that the MCM model is capable of predicting the amounts of isoprene reacted in the continuous flow setup and to determine the main oxidants. Below we include Figure 3.9 from Fuchs (2017) showing modeled oxidant fractions.

[Figure]

Figure 3.9: Modelled fractions of isoprene reacting with $O_3$, OH, or $NO_3$ in dependence of $NO_x$ concentration.  The reaction of isoprene with OH stays overall constant while the reaction with $O_3$ decreases and the reaction with $NO_3$ increases with increasing $NO_x$ concentrations.

Figure 3.9 from the thesis by Claudia Fuchs (Fuchs, 2017).

**Changes to manuscript**; page 9, line 19, sentence added:
"Based on the MCM predictions, the reaction of isoprene by OH radicals is estimated to have accounted for approximately 30% of the isoprene oxidation while the remaining ~ 70% were due to oxidation by ozone (under dark, low-NO$_x$ conditions) (Fuchs, 2017; Section 3.5.3)."

6) Page 9, line 12-14: "The major stable products predicted by the MCM for the first/early generations of oxidation are compounds of low molecular mass with relatively low O:C ratios and high vapour pressures (VOC to IVOC class compounds)."

I suggest that you add numbers on how large fraction of the total modelled stable oxidation products that go via these intermediate early-generation oxidation products. Otherwise you wonder why you only selected these specific compounds.

**Authors' response**:   The fraction of the early generation products that go into forming the intermediate products (later generation products) is indicated by the scaling parameters in Table 1 (Page 16) of the manuscript.

7) Page 9, lines 16-18: In order to validate the amounts of ozone, OH and "isoprene reacted" predicted by the MCM, conditions and measurements from the CLOUD 10 and CLOUD 9 chamber experiments, reported by Fuchs (2017), were used for a comparison with observations, further discussed in the following.

Please consider to reformulate this sentence. It is not easy to understand what you mean with "isoprene reacted". Do you mean amount of isoprene oxidized by O3 and OH?

**Authors' response:**  The above sentenced has been rephrased.

**Changes to manuscript:** page 9, lines 16 – 17, rephrased:
"In order to validate the amounts of ozone, OH and "isoprene reacted" (i.e. the isoprene amount oxidized by O$_3$ or OH radicals) predicted by the MCM, conditions and measurements from …"

8) Page 9, lines 20-21: "Additional components of semi-volatile and low-volatility nature formed via multi-generation oxidation, which are not fully considered by MCM, need to be included in the partitioning model as part of the isoprene system."

I don't question that MCM do not predict all semi-volatile and low-volatility products that actually contributes to SOA mass, but have you actually tested to simulate the SOA formation from the isoprene oxidation products generated in MCM and compared the results with the observations in CLOUD?
9) I think this should be an easy test to do. I know that the MCM scheme for isoprene reacted with OH generate some semi- to low-volatility products that do contributes to SOA mass and yields comparable with experimental yields (see e.g. Xavier et al., Atmos. Chem. Phys., 19, 13741–13758, 2019).

**Authors' response:**  As the early generation isoprene oxidation products simulated by MCM remain predominantly in the gas phase, they are insufficient to predict/replicate the SOA formation in comparison with CLOUD chamber observations. Higher generation products are indeed predicted by multi-generation MCM simulations and using all of those products may allow for a more detailed partitioning simulation. However, in this work, the molecular concentrations for all > 500 compounds that could be output by MCM for isoprene oxidation, have not been evaluated for the partitioning computations. This is due to limitations in the number of components the current AIOMFAC-based

partitioning model can process for coupled gas–particle partitioning and liquid–liquid phase separation. Instead, using a set of surrogate components and associated scaling parameters was considered a more practical and reasonably detailed mode of modeling, especially given the easier tuning of scaling parameters to match experimental data for the purpose of calculations at various RH levels. See also the response to comment 4) above. We agree that improvements are possible in terms of number of simulated oxidation products directly used in partitioning computations. We note that independent work is ongoing to enable the AIOMFAC-based model to run with such more detailed MCM outputs with >100 compounds and to solve limitations in the number of surrogate compounds the AIOMFAC model can work with.

10) For pure isoprene + O3 oxidation you probably do not form any semi- and low-volatility products that contribute to SOA mass formation but since you have secondary production of OH during the CLOUD experiments and in the model, you should also form some low-volatility products from the MCM gas-phase chemistry scheme. On example is HMACROOH in Fig. 1 and 2.

When reading this I also ask the same question again: How large fraction of the reacted isoprene were via isoprene + OH?

**Authors' response:** See the response to comment 5) above.

11) Page 9, lines 23-28 "The selected surrogate components are shown in Fig. 1. Oxalic acid along with compounds such as 2-methyltetrol, 2-hydroxy-dihydroperoxide, 2-methylglyceric acid and a C5-alkene triol are among the main semi-volatile and low-volatility products that are expected to form after oxidation (by ozone and by OH) of the early generation compounds under low NOx conditions. The selected C10 hemiacetal dimer is a compound representing a whole class of potential dimer and oligomer compounds formed by accretion reactions from the above-mentioned five higher generation products."

I miss a clear description about why these specific surrogate compounds were selected. At least add some proper references. You want to know how atmospheric relevant the selected SOA composition is.

**Authors' response:** Several of the selected surrogate compounds were based on SOA-relevant species suggested by Couvidat and Seigneur (2011): 2-methyltetrol, 2-hydroxy-dihydroperoxide, 2-methylglyceric acid; the semi-volatile component oxalic acid from the work by Carlton et al. (2009) and the $C_5$-alkene triol species as well as the $C_{10}$ hemiacetal dimer based on the work by Surratt et al. (2010). The production of the two latter surrogates is based on an aqueous (aerosol) phase reaction pathway following gas–particle partitioning of IEPOX. Associated sentences have been revised.

**Changes to manuscript;** sentences from page 9, lines 23–28 revised to:
"The selected surrogate components are shown in Fig. 1. Oxalic acid along with compounds such as 2-methyltetrol, 2-hydroxy-dihydroperoxide, 2-methylglyceric acid and a $C_5$-alkene triol are among the main semi-volatile and low-volatility products that are expected to form after oxidation (by ozone and by OH) of the early generation compounds under low-$NO_x$ conditions. Oxalic acid formation in the aqueous phase is shown in Carlton et al. (2009); the compounds 2-methyltetrol, 2-hydroxy-dihydroperoxide, 2-methylglyceric acid are suggested as surrogate compounds by Couvidat and Seigneur (2011). The $C_5$-alkene triol and the $C_{10}$ hemiacetal dimer are selected as surrogates based on partitioning of IEPOX to aerosols followed by aqueous-phase reactions described by Surratt et al. (2010). The selected $C_{10}$ hemiacetal dimer is a compound representing a whole class of potential dimer and oligomer

compounds formed by accretion reactions from the above-mentioned five higher generation products."

**12) I also miss information about the modelled and observed SOA elemental composition (O:C and H:C). Clearly these results exists and will provide at least some indication about how well the surrogate SOA species parameterization is able to represent the SOA properties e.g. hygroscopicity during the different CLOUD experiments.**

**Authors' response:** Fuchs (2017) reports average SOA elemental compositions based on measured desorption profiles from a FIGAERO-CIMS instrument. For the seed-free case (and low-$NO_x$ conditions) the determined average O:C for SOA ranged from 0.62 – 0.67 at low RH (~35 % RH) and up to 0.69 at high RH (~85 % RH), i.e. indicating only a small change in average O:C between low and high RH conditions (Fuchs, 2017) (Section 3.6.1, p. 79 in that work). The measured average H:C ratios were about 1.48 to 1.50.  Our modelled SOA is of higher average O:C and H:C ratios. At 278 K, for 140 µg m$^{-3}$ of reacted isoprene (a typical value for comparison with the CLOUD experiments, see Fig. 3), the predicted SOA has the following properties: at 35 % RH, 1.77 µg m$^{-3}$ SOA, O:C of 0.843 and H:C of 2.26; at 85 % RH, 3.14 µg m$^{-3}$ SOA, O:C of 0.844 and H:C of 2.18. The values at RH levels in between are similar. The lower-volatility surrogate species used with the model have individual H:C ≥ 2.0; e.g., the $C_{10}$-dimer has a H:C ratio of 2.2 and O:C ratio of 0.7. While this comparison suggests substantial differences between measured and modelled average elemental organic aerosol composition, it remains unclear how different the associated SOA water uptake behavior is.  We note that we do not have direct hygroscopicity measurements, such as growth factors, to compare to. The O:C ratio of the surrogates are shown in Fig. 2. The above paragraph is added to the revised manuscript.

**Changes to manuscript:** at the end of Sect. 3.1.1 we add the following statements in the revised version: Fuchs (2017) report average SOA elemental compositions for the CLOUD experiments based on measured desorption profiles from a FIGAERO-CIMS instrument. For the seed-free case (and low-$NO_x$ conditions) the determined average O:C for SOA ranged from 0.62 – 0.67 at low RH (~35 % RH) and up to 0.69 at high RH (~85 % RH), i.e., indicating only a small change in average O:C between low and high RH conditions (Fuchs, 2017) (Section 3.6.1 in that work). The measured average H:C ratios were about 1.48 to 1.50.  Our modelled SOA is of higher average O:C and H:C ratios. At 278 K, for 140 µg m$^{-3}$ of reacted isoprene (a typical value for comparison with the CLOUD experiments, see Fig. 3), the predicted SOA has the following properties: at 35 % RH, 1.77 µg m$^{-3}$ SOA, O:C of 0.843 and H:C of 2.26; at 85 % RH, 3.14 µg m$^{-3}$ SOA, O:C of 0.844 and H:C of 2.18. The values at RH levels in between are similar. The lower-volatility surrogate species used with the model have individual H:C ≥ 2.0; e.g., the $C_{10}$-dimer has a H:C ratio of 2.2 and O:C ratio of 0.7.  While this comparison suggests substantial differences between measured and modelled average elemental organic aerosol composition, it remains unclear how different the associated SOA water uptake behavior is.

**14) Page 11, lines 19-21: "Surratt et al. (2010) proposed a chemical mechanism via the reactive intermediate epoxydiols (IEPOX) pathway leading to the formation of 2-methyltetrols and C5-alkene triols for the oxidation of isoprene by OH."**

The MCM chemistry represent formation of epoxydiols. Why did you not include IEPOX as an early generation precursor to 2-methyltetrols and C5-alkene triols? This would make the modelled SOA mass predictions more explicit.

**Authors' response:**  One of the reasons for not including IEPOX as additional early generation precursor is because the AIOMFAC model is limited in the functional groups it can account for and epoxy groups are not covered. Furthermore, the IEPOX uptake to aerosols is not covered by MCM, so there would still

be a gap between predicted IEPOX (by MCM) and formation of the 2-methyltetrols and $C_5$-alkene triol surrogate compounds (requiring a scaling parameter). As stated on page 11, lines 14 – 17, we aimed for a rather simple surrogate system, for the stated reasons, which implies accepting gaps in the chemical pathways leading to the set of selected surrogate compounds.

15) Page 11, lines 24-25: "Multigeneration surrogate products from the MCM predicted yields of early-generation products are used to ensure that no carbon mass is unaccounted for."

I don't understand this sentence completely. Do you mean that all (100 %) of all isoprene -oxidation pathways will go via the 6 selected early-generation oxidation products in Fig. 1 and hence, no carbon mass is unaccounted for?

**Authors' response:** We agree that this sentence was unclear. It refers to how the mass concentrations of higher generation surrogate products were linked to the early generation products directly predicted by MCM. Our approach ensures that the carbon mass present in higher generation surrogate compounds (based on set branching ratios) is accordingly deducted from early generation products that were assumed to act as parent compounds for the higher generation products. Since this is described in more detail in Sect. 2.3.4, we have deleted this unclear sentence in the revised version.

**Changes to manuscript:** sentence deleted.

16) Page 13, lines 17-19 "Molar yields of MCM-predicted products used in the model for the CLOUD 10 (seed-free) case were different based on the individual concentration of components present at the time when the chamber stabilized for the seed-free experiment."

This sentence I do not understand. Please clarify what you mean.

**Authors' response:** We have revised this sentence.

**Changes to manuscript**; page 13, lines 17 – 19, sentence revised to:
"We note that the molar yields of individual MCM-predicted products differed between the simulations of CLOUD 9 and CLOUD 10 cases due to differences in experimental conditions. This is accounted for by means of different sets of input concentration data for the gas–particle partitioning calculations of seeded or seed-free cases when compared to the CLOUD experiments.

17) Page 13, lines 27-30: "Details of the MCM box model approach in Fuchs (2017) are as follows: the box model used inputs for the isoprene concentration, ozone concentration, condensation sink (from SMPS data), inorganic seed concentration, rainout rate constant (quantifies the rate of formation of cloud droplets and the associated mass loss in the cloud), dilution rate constant, aerosol wall loss rate constant, RH, reaction rate constants of isoprene with the OH and O3 oxidants, the hygroscopic growth factors …"

Is this also the approach you used? With isoprene and ozone concentration, do you mean the steady state concentrations in the chamber or the concentrations in the inflow?

**Authors' response:** The approach we used is based on the same MCM simulation setup for the gas phase chemistry, but with a simplified treatment of wall-loss correction. We did not use the volatility basis set box model for this work since the partitioning is computed with our thermodynamic equilibrium model accounting for non-ideal mixing. This is mentioned later in the same paragraph. Regarding the isoprene and ozone concentrations, we mean the concentrations in the chamber inflow.

**Changes to manuscript:** first part of sentence revised to "Details of the MCM box model approach in Fuchs (2017) are as follows: the box model used inputs for the isoprene and ozone inflow concentrations, condensation sink (from SMPS data), inorganic seed concentration, …"

21) Page 15, lines 23-25: "In our simple model, the scaling parameters (branching ratios) are adjustable and are determined iteratively by model–measurement comparison. After the gas–particle partitioning model was used to calculate the equilibrium SOA mass concentration formed at varying levels of reacted isoprene, covering the range observed in the CLOUD chamber experiments,".

Ok, so in a way you produce a 5-species SOA yield parameterization which match the observed SOA mass, or?

**Authors' response:** Yes, it is a form of a yield parameterization. However, as written in the sentences following the cited statement, our parameterization of the branching ratios involves a simultaneous fit to data from both low and high RH conditions in the experiments, which is different from the approach taken in traditional parameterizations, such as with a (1-D) volatility basis set fitted to dry conditions data only.

**Changes to manuscript:** (page 16, line 1) sentence added before "Reasonably good agreement…": "The consideration of observed SOA mass concentrations formed at several levels of reacted isoprene as well as low and high RH levels makes our scaling parameter determination distinct from the approach taken in more traditional SOA yield parameterizations, such as fitting of a (1-D) VBS to dry conditions data only."

22) I don't understand this completely. Was this not the same experiments which you used to fit the branching ratios? I have hard to differentiate between what actually was statistic fitting of the branching ratios and what is standalone modelling without tuning.

**Authors' response:** This section (Sect. 2.3.4) and the specific lines (page 15, lines 20 – 28) describe how the branching ratios were determined based on CLOUD chamber data for seed-free experiments (CLOUD 10) under low-NO$_x$ conditions. Hence this is about the fitting approach not standalone modeling. Results from modeling different conditions (including the whole RH range) based on the determined branching ratios are discussed in section 3. In section 3, we also compare and discuss cases where the branching ratios were scaled (e.g. in case of the seeded CLOUD experiments), for reasons described there (Sect. 3.2.1).

23) Which input parameters did you use to constrain the branching ratios and how was it done in practice. I guess you can find many different combinations of the beta parameters that give more or less the same SOA yield curves.

**Authors' response:** We used a manual model fitting approach, in which the gas–particle partitioning model was run using the MCM-predicted early generation product concentrations and with a guess for the branching ratios to predict the amount of SOA formed in comparison with measured SOA mass concentrations from the seed-free CLOUD experiments. This process was iterated over several times with manual adjustments to the branching ratios to determine a suitable fit of the modelled SOA curve with the measured SOA data at 35 % and 85 % RH simultaneously (described in last paragraph of page 15 and top of page 16) Yes, it is the case that various other combinations of the scaling parameters (i.e. branching ratios) can result in approximately the same SOA yield curves; although attempting to match the observed RH-dependence constrains the

options compared to fitting only to data at one particular RH. Our determined parameter set listed in Table 1 is therefore not unique. We clarify this in the revised text.

**Changes to manuscript:** (page 16, line 2) sentence revised: "We note that our set of determined scaling parameters (i.e. the branching ratios) is not unique; other combinations of scaling parameters may provide similar agreement with the observations. The determined scaling parameters for the surrogate species are provided in Table 1."

24) Page 24, line 1: "However, in the absence of OH radicals and light in the CLOUD experiments …" But OH was formed during the ozonolysis of isoprene.

**Authors' response:** Yes, 30% of isoprene reaction pathway was with OH, so this sentence has been rephrased.

**Changes to manuscript:** Page 24 line 1 "However, in the absence of light in the CLOUD experiments, the rates of aqueous SOA formation pathways are limited to dark processes and are likely only of minor importance."

25) Page 24, lines 4-6: "For consistency, the same molar yields were used for all seeded cases considered since the differences among the experimental conditions during the seeded experiments were relatively minor (yet different compared to seed-free)."
I don't understand completely. Did you use the same molar yields for the early products as in the seed-free case or different yields? I thought that a novelty with the model approach was that you base the yield calculations of the near-explicit MCM chemistry.
If you do not allow the early generation yields to vary according to the MCM chemistry I don't see the point in calculating the beta parameters based on the specific selected MCM compounds.

Can you please clarify this?

**Authors' response:** The phrasing of this sentence was not clear; we have revised it. We used MCM to predict the yields. The point here is about the use of the same molar yields (predicted by MCM) for the early generation products in the case of the various seeded experiments (listed in Table S2). Those yields were different from those for the seed-free simulations.

**Changes to manuscript:** (Page 24, lines 4-6), sentence rephrased to:
"The MCM-predicted molar yields of the early generation products for the conditions of the seeded experiments were different from those for the seed-free ones. In contrast, the predicted molar yields of the various seeded experiments were similar; therefore, for consistency and ease of comparison among different model calculations for seeded cases, a single set of molar yields was used (listed in Table S2)."

26) Page 24, lines 8-14: "Hence, the gas–particle partitioning model was modified to account for RH-dependent SOA and water partitioning in the presence of an aqueous phase of variable inorganic seed concentration. We note that the measured steady-state flow chamber aerosol mass concentration is affected by the dynamic interplay between the different vapour and particle loss mechanisms: wall-loss, loss by chamber outflow, and the aerosol condensation sink. The latter depends on the particle size distribution and therefore also on humidity-dependent water uptake by seed particles and organic partitioning."

It is unclear to me what you actually mean with this. How was the model modified? Did you change the beta values?

**Authors' response:**
Sentence rephrased. "Modified" was perhaps unclear wording. We meant that the model allows for the influence of an aqueous inorganic (seed) phase and partitioning to this phase or an organic-rich phase, where present.

**Changes to manuscript:** (Page 24, line 8) sentence rephrased to:
"Hence, the gas–particle partitioning model was run in a mode which allows for liquid–liquid phase separation as well as non-ideal organic–inorganic mixing in each particle phase, i.e. for RH-dependent partitioning of water, SOA, and inorganic seed either into a single mixed phase or into two distinct particle phases."

27) You note the importance of the dynamics interplay between the vapor on the walls and the condensation onto the particles, but do you consider it somehow in the model?

**Authors' response:** The gas–particle partitioning model considers the impact of this dynamic interplay only indirectly by means of an introduced adjustment of the determined scaling parameters (branching ratios β) by a factor of 0.55 for the seeded cases. This is further described in Sect. 3.2.1 on the following page in the manuscript. We added a sentence to the revised manuscript.

**Changes to manuscript** (Page 24, line 13) sentence added at end of paragraph:
"Such effects are only indirectly accounted for in the equilibrium gas–particle partitioning calculations by means of scaling the branching ratios for the higher generation surrogates (by a factor of 0.55) in the seed-containing cases; see details in Sect. 3.2.1."

28) Page 24, lines 30-21: ". Occasionally, the formed clouds led to some precipitation in the chamber, which may have led to subsequent loss of organic aerosol mass and number concentration (reducing the condensation sink)."

Yes but was not steady state reach in-between the precipitation events? Can you provide some experimental data (e.g. from the SMPS) that justify the suggested decreasing condensation sink?

If steady state is not reached, it must be hard to constrain an equilibrium model based on the observations.

**Authors' response:** For model–measurement comparisons, we only selected time periods when steady state or near-steady conditions were reached in the chamber experiments, i.e. excluding times during and shortly after cloud/precipitation formation in the chamber. Below, we included Fig. 3.2 from the thesis by Fuchs (2017), which shows the SMPS-measured dried aerosol size distribution for a time period spanning several hours, and including three cloud formation events, during the CLOUD 9 experiments. As explained in Fuchs (2017, page 65) the initial decrease in particle number concentration occurs primarily due to dilution in the chamber (seed particles were only added at the beginning). Our statements in this paragraph of the manuscript (page 24, lines 32 to page 25 line 5) provide one aspect of explaining why the MCM-simulated molar yields of early generation species and the resulting SOA mass concentration may deviate from the measurements after such events: increase in the relative importance of wall losses, not considered by the MCM simulations. On the contrary, at least prior to the first cloud formation event, the presence of seed particles would be expected to lower the importance of organic vapour wall losses compared to the seed-free cases (CLOUD 10 experiments). Therefore, while changes in the vapour wall-loss behavior between the two series of CLOUD experiments are likely contributing to the discrepancy, our quantitative understanding remains incomplete.
To this end, we will add a statement to the pertaining paragraph in the revised manuscript.

**Changes to manuscript** (Page 25, line 5) sentence added at end of paragraph:

"On the contrary, the presence of seed particles would be expected to lower the importance of organic vapour wall losses compared to the seed-free cases (CLOUD 10 experiments). However, the application of the equilibrium partitioning model with scaling parameters tuned by CLOUD 10 observations led to over-predicted SOA mass concentrations. Therefore, while changes in the wall-loss behavior between the two series of CLOUD experiments are likely contributing to the resulting discrepancy, our quantitative understanding remains incomplete."

[Figure]

Figure 3.2: Time series of temperature, pressure and dried aerosol size distribution of a typical experiment measured with an SMPS. The drop in pressure and temperature (upper panel) indicates an adiabatic expansion causing a cloud formation (pink areas in the SMPS size distribution, lower panel, demonstrate the presence of a cloud). Grey areas represent sub-saturated conditions (RH ~90 %). The white dots illustrate the mode diameter indicating the growth of the dry aerosol particles. The beginning of each expansion is demonstrated by the vertical lines. The data gaps during the cloud periods are due to flow issues in the SMPS.

Figure 3.2 from the thesis by Claudia Fuchs (Fuchs, 2017), showing (dried) aerosol size distribution data for a time period and conditions during the seeded CLOUD 9 experiments.

29) Figure 6. Is the increasing SOA mass as a function of the inorganic seed aerosol a result of increased condensation sink, changed chemistry or both? Can you differentiate between these effects?

**Authors' response:** In the experiments, the increasing SOA mass concentration is possibly due to a combination of effects, which cannot be easily differentiated from the data alone: hygroscopicity effects leading to enhanced partitioning to the condensed phase due to higher absorbing aerosol mass are likely the dominant effect (considering the related model predictions); however, there may have been changes in aqueous-phase chemistry and some changes in the condensation sink affecting the resulting SOA mass concentration. The gas–particle partitioning model does not include any changes due to chemistry or condensation sink at different seed concentrations. Therefore, the predicted increasing

SOA mass concentration as a function of inorganic seed mass concentration is due to the increased absorbing mass concentration of the mixed organic–inorganic aerosol particles as shown by the model curves in Fig. 6a. The absorbing aerosol mass is increased by a combination of effects: higher seed mass concentration also leads to higher aerosol water content at 85 % RH and this leads to enhanced partitioning of semi-volatile organic (surrogate) compounds to the particles (into a mixed phase), which in turn enhances aerosol water content and absorbing mass. This is the mechanism that the model captures, which is in reasonable agreement with the observed SOA enhancement at constant amounts of reacted isoprene. Such effects are discussed in more detail in section 3.2.4. Note that if the AIOMFAC-based model is run with the inorganic seed and SOA not allowed to mix (forced complete phase separation), there would be no increase in predicted SOA mass concentration with ammonium bisulfate mass concentration. That is, the model curves would be horizontal lines in Fig. 6a at the level predicted for zero seed mass concentration.

30) Page 27, line 14: "Observations by Kleindienst et al. (2007), using an ammonium sulfate seed (0.05 µg m-3)" This is a very low concentration of AS seed. Is the value correct?

**Authors' response:** Yes, this value seems low. We checked again and 0.05 µg m$^{-3}$ of AS seed is indeed the value stated in paragraph [8] of Kleindienst et al. (2007). It may have been so low because their intended use of AS seed was simply to promote aerosol formation.

31) Page 28, lines 16-17: "SOA yield values observed in the chamber vary from 0.3 % to 1.4 % (uncorrected), while the model predicted values are in the range of 0.8 % to 1.4 % for the same SOA mass concentration range (Fig. 8b)."

Can you explain the difference between the modelled and observed yields at low SA seed concentrations? How would the dynamics i.e. condensation sink affect the results?

**Authors' response:** At low sulfuric acid seed concentrations in the experiments, the losses of isoprene oxidation products to the chamber walls may compete substantially with condensation to aerosol particles, which would at least partially explain the observed lower SOA mass concentrations compared to those from our model predictions. At higher seed mass concentrations, such wall-loss effects are expected to be smaller. Figure 8 indicates that the model–measurement agreement for SOA mass concentration and yield improves with increasing seed concentration. This interpretation is further supported by additional unpublished sensitivity analyses using the box model simulations from Fuchs (2017; their Chapter 3) as starting point.

**Changes to manuscript** (Page 28, line 17) sentence added at end of the paragraph:
"At low sulfuric acid seed concentrations in the experiments, the losses of isoprene oxidation products to the chamber walls may compete substantially with condensation to aerosol particles, which would at least partially explain the observed lower SOA mass concentrations compared to those from our model predictions. At higher seed mass concentrations, such wall-loss effects are expected to be smaller. Figure 8 indicates that the model–measurement agreement for SOA mass concentration and yield improves with increasing seed concentration.

32) Page 31, lines 19-20: "That is, we are not accounting for possible changes in gas-phase or particle-phase chemical reaction pathways due to changes in RH, aqueous phase ionic strength and/or acidity." and maybe even more important the dynamic e.g. condensation vs wall losses of condensable vapors or?

**Authors' response:** Yes, in addition we are also not accounting for changes in reaction pathway due to wall-loss of condensable vapors at different RH levels. The wall-loss effects likely explain discrepancies at very low seed concentrations. However, note that the discussion on those lines is concerning the predicted SOA mass yield enhancement effects by the partitioning model in a general sense (not a specific chamber experiment context). Therefore, we think that it would be confusing to mention dynamic condensation and wall loss at that point.

33) Page 36, lines 13-15: "Tools like the AIOMFAC-based phase separation model coupled with chemical kinetics and flow simulations could be used in the future to better constrain the effect of non-ideality and aerosol water content on the overall chamber dynamics and related interpretation of measurement data."

I fully agree with you.

**Authors' response:** Thank you for your comments and queries.

**References**

Carlton, A., Wiedinmyer, C., and Kroll, J.: A review of Secondary Organic Aerosol (SOA) formation from isoprene, Atmospheric Chemistry and Physics, 9, 4987-5005, 2009.

Couvidat, F., and Seigneur, C.: Modeling secondary organic aerosol formation from isoprene oxidation under dry and humid conditions, Atmospheric Chemistry and Physics, 11, 893-909, 2011.

Fuchs, C.: Investigation of the role of $SO_2$ and isoprene in aqueous phase secondary aerosol formation, Dissertation ETH Zurich 2017 No. 24157, 177 pages, 2017.

Jenkin, M. E., Young, J. C., and Rickard, A. R.: The MCM v3.3.1 degradation scheme for isoprene, Atmos. Chem. Phys., 15, 11433-11459, 10.5194/acp-15-11433-2015, 2015.

Kleindienst, T. E., Lewandowski, M., Offenberg, J. H., Jaoui, M., and Edney, E. O.: Ozone-isoprene reaction: Re-examination of the formation of secondary organic aerosol, Geophysical Research Letters, 34, 2007.

Kuwata, M., Zorn, S. R., and Martin, S. T.: Using Elemental Ratios to Predict the Density of Organic Material Composed of Carbon, Hydrogen, and Oxygen, Environmental Science & Technology, 46, 787-794, 10.1021/es202525q, 2012.

Surratt, J. D., Chan, A. W. H., Eddingsaas, N. C., Chan, M. N., Loza, C. L., Kwan, A. J., Hersey, S. P., Flagan, R. C., Wennberg, P. O., and Seinfeld, J. H.: Reactive intermediates revealed in secondary organic aerosol formation from isoprene, PNAS, 107, 6640-6645, 10.1073/pnas.0911114107, 2010.

Xavier, C., Rusanen, A., Zhou, P., Dean, C., Pichelstorfer, L., Roldin, P., and Boy, M.: Aerosol mass yields of selected biogenic volatile organic compounds – a theoretical study with nearly explicit gas-phase chemistry, Atmos. Chem. Phys., 19, 13741-13758, 10.5194/acp-19-13741-2019, 2019.

---

## Author Comment (AC2)

**Responses to referee #2**

We would like to thank the referee for her/his comments and efforts in providing this review. Below, we repeat each comment by this referee (in blue font) followed by our response, and we indicate related changes made to a revised version of the manuscript. Page and line numbers stated refer to those from the original manuscript version.

**General Comments:**

1) In this manuscript, the authors couple a gas-phase mechanism to an equilibrium partitioning model to simulate SOA formation from isoprene ozonolysis in the CLOUD chamber. The main focus of the paper is in understanding the role RH and inorganic seed particles play in determining SOA yield. The model is also used to predict whether liquid/liquid phase separation occurs and under what RH conditions. The topic is of interest to the readers of ACP and fits in well with the journal. The modeling techniques are state of the art. The figures are relatively clear and appropriate. With that said, I had a very hard time evaluating this manuscript for reasons given hereafter and therefore recommend it be revised significantly and potentially re-reviewed before publication.

**Authors' response:** Thank you for your overall assessment. In response to the valuable comments from both reviewers, we have revised several subsections and paragraphs of this manuscript to improve clarity and the discussion of results.

2) First, the manuscript is somewhat long, dense, and tends to wind between several different experimental conditions. I suggest editing for length and clarity; the authors can probably cut 1/3 of the length of the paper without losing much. Many sections read like a thesis with descriptions of the model results, but lack context regarding what the results mean or the real-world implications of the results. Second, the references to the experimental description and data are to a thesis, which is almost 200 pages long. This reviewer didn't read through the 200 page thesis to try to figure out what was done. The experimental results are key to evaluating the modeling. The authors should include a short section describing some aspects of the experiments that a key to understanding the results. Many of these uncertainties are listed below in the specific comments section. There are a lot of small mistakes, contradictions, confusing sections, etc. Any one of these is small on its own, but they add up to make it difficult to understand how robust the results are and frustrating to try to follow the manuscript.

**Authors' response:** In response to the specific comments, we have revised phrasing and contents of several sections to improve clarity. We have also opted to shorten certain sections or to move subsections to the Supplementary Information (SI). However, cutting 1/3 of the lengths of the manuscript is not straightforward and may be a matter of personal preferences. The authors are of the opinion that the discussion of different experimental conditions and comparisons of our model predictions to a number of measurement conditions, including from studies other than those conducted in the CLOUD chamber, are useful. Some of those discussions could be moved to the SI and we have done that in some cases in the revised version of the manuscript (those changes are indicated under specific comments below).
Description of SOA properties and experimental yields from the CLOUD experiments, described on pages 70 – 79 of the PhD thesis by Fuchs (2017), have been briefly summarized on pages 5 and 6 of our manuscript. We will add a new section to the SI, providing further details about the CLOUD chamber experiments, in particular on the use of data from the seed-free experiments listed by Fuchs (2017) for the fit of our gas–particle partitioning model parameters; see also the response to specific comment 13

below.

For further clarification, the goal of our study is not to model the time series of the conditions and concentrations during the continuous flow experiments in the CLOUD chamber. The goal is to model and assess the influence of RH and different inorganic seeds on the gas–particle partitioning of SOA-relevant species. Thus, exploring enhancement effects of water uptake and seeds on SOA mass concentrations under the assumption of unchanged chemistry. The CLOUD chamber experiments serve as valuable source of data to constrain the empirical model parameters introduced related to the surrogate compound yields, such that the SOA concentrations at both high and low RH levels could be reproduced by the model. Hence, the emphasis in this work is on the modeling approach and predictions, and less on the details of specific experiments. We appreciate that those details are also of interest to many readers and refer for full details to the thesis by Fuchs (2017).

3) Most importantly, the authors are trying to probe very small difference in the SOA yields between chamber experiments conducted at different RHs and with different seed. In most cases, the difference in absolute yield is between experiments is around 1%. The authors need to convince the reviewers that his 1% difference is significant. The model is tuned to match results at one particular set of conditions (e.g. 35% RH) and then applied to the other condition. It appears that only one experiment at each condition was conducted. Experiment to experiment variability for chamber experiments is going to be larger than 1% yield. The authors indicate that a simple gas-phase wall loss correction was applied and the same correction appears to be used across the different conditions. In addition, the uncertainty on the measurements themselves is 50%; I don't believe this includes any experiment reproducibility error. These errors will all add up. Finally, the gas-phase mechanism needed to be adjusted between the seeded and unseeded runs. There is little reason to believe that the actual gas-phase chemistry produces a factor of almost 2 difference in the produce yield. So obviously the model is not capturing all the processes that determine yield. In sum, I find it hard to imagine that a 1% difference in yield in a large SOA chamber experiment is significant. The authors haven't show repeated measurements to establish this level of accuracy and reproducibility was achieved. This makes it very difficult to evaluate whether the model results are robust or whether they are essentially modeling experimental noise.

**Authors' response:** It was perhaps not clear that several data points were used to establish the RH effect on SOA yields from the CLOUD experiments; so, we would like to explain why the data is more robust than this referee may have interpreted. While all experimental data were considered to have a 50 % uncertainty in SOA mass concentration, the partitioning model itself was constrained based on several data points for different amounts of reacted isoprene, which we argue, leads to a much lower uncertainty – at least in terms of representing high vs. low RH effects on partitioning, not necessarily with respect to absolute yields. As mentioned above, the scope of the paper is not on explaining differences in SOA yields between different chamber experiments; rather, we focus on investigating the effect of thermodynamic partitioning on SOA yields, with consideration of variations in RH and inorganic seed concentrations, using the AIOMFAC-based equilibrium partitioning model. One research question studied is: does a mixed organic–inorganic particle phase form at moderate to high RH levels, thereby leading to an increase of SOA mass yield with increasing seed concentration due to feedback effects on partitioning? We argue that a difference of an absolute 1 % unit in SOA mass yield at high vs. low RH is substantial in this case and supported by several data points taken at each RH level, those data being shown in Fig. 3. The model parameters (scaling parameters of the higher-generation surrogate component yields) were constrained by the data from the CLOUD experiments for seed-free conditions at 5 °C. Those experiments were conducted at both lower (35 % RH) and relatively high (85 % RH) humidity levels and the measured SOA mass concentrations and yield data collected at near steadystate chamber concentrations refer to different times during those continuous flow experiments. Collectively, the shown data is in support of an RH effect on SOA that is systematic. Furthermore, we also address the dependence of SOA mass concentration on the concentration of inorganic seed present and show that organic–inorganic mixing at high (85 %) RH is important in that case (e.g. Fig. 6).

We note that the seed-free experiments used were run in the same manner over a few days during the CLOUD 10 campaign. Therefore, our claim is that the RH-related change in SOA mass yield is more robust than the absolute numbers of the yield values, which may differ from experiment to experiment, especially when carried out or evaluated in different ways (or using different environmental chambers). For matching the partitioning model predictions of SOA mass concentration to the measurements of the seeded experiments, we had to adjust the yields of the higher-generation surrogate compounds by a single global scaling factor only. Although this would not be necessary in an ideal case, this was done to align predicted and measured SOA mass concentrations offering a more direct comparison between model and measurement with respect to different seeds used. By tuning the partitioning model without wall losses to observations which include wall losses, we implicitly include wall losses in the global scaling factors. The global correction factor of almost 2 in the scaling factor and resulting surrogate component yields can most likely be associated with operating at conditions with different contributions from wall losses. Note that the seeded experiments were conducted in a different campaign (CLOUD 9), using different temperature and pressure conditions in the chamber, such that deviations in the resulting SOA mass concentrations may be due to changes in the way the experiments were conducted, including changes in effective losses to the walls. Again, our goal in this study is to understand and model the direct and indirect effects of RH on SOA, which our model allows us to do even when applying a scaling factor. Clarifications and changes to the manuscript for several of those general comments are further addressed under specific comments.

**Specific Comments:**

4) Through paper: The authors refer to Fuchs 2017 for a description of experimental conditions. This reference is to a thesis, which struck me as somewhat unusual. Was this work never published in a journal? Given that the measurements are central to the modeling, it would be good to expand somewhat more on the conditions.

**Authors' response:** To date, the chamber experiments described in detail in the thesis by Fuchs (2017) have not been published in a journal. In our study, the details on CLOUD chamber experiment setup and instrumentation are referenced to Chapter 3 of the thesis, in particular pages 57 – 60 of Fuchs (2017). In the revised manuscript, we have included more information on the chamber data used (e.g. the table listing values also shown in Fig. 3) and conditions like the split of oxidants ($O_3$ vs. OH); see the following comments and responses.

5) Through paper: What fraction of isoprene reacts with ozone vs OH? Significant OH will be produced given the high concentration and continuous input of isoprene and ozone.

**Authors' response:** The MCM (MCM v3.3.1, Jenkin et al., 2015), model output suggests that about 30 % of the isoprene oxidation in the dark chamber was due to OH. The remaining approximately 70 % of the isoprene oxidation was due to $O_3$ for low-$NO_x$ conditions. This is an aspect that was studied by Fuchs (2017), both to confirm that the MCM model is capable of predicting the amounts of isoprene reacted in the continuous flow setup. Below, we include Figure 3.9 from Fuchs (2017) showing modeled oxidant fractions.

[Figure]

Figure 3.9: Modelled fractions of isoprene reacting with $O_3$, OH, or $NO_3$ in dependence of $NO_x$ concentration. The reaction of isoprene with OH stays overall constant while the reaction with $O_3$ decreases and the reaction with $NO_3$ increases with increasing $NO_x$ concentrations.

Figure 3.9 from the thesis by Claudia Fuchs (Fuchs, 2017).

**Changes to manuscript**; page 9, line 19, sentence added:
"Based on the MCM predictions, the reaction of isoprene by OH radicals is estimated to have accounted for approximately 30 % of the isoprene oxidation while the remaining ~ 70 % were due to oxidation by ozone (under dark, low-$NO_x$ conditions) (Fuchs, 2017; Section 3.5.3)."

6) P2/3 Lines 33-2 : Sentence needs revision, it is unclear.

**Authors' response:** Sentence rephrased.

**Changes to manuscript (P2/3 Lines 33-2)**: "The implementation of advanced thermodynamic aerosol models serve the assessment and operational forecasting of chemical and physical aerosol processes, including how they affect air quality and aerosol–cloud–climate interactions (Johnson et al., 2006; Cappa et al., 2008; Hallquist et al., 2009)."

7) Page 3: Introduction is somewhat difficult to follow because the authors skip and intermingle between ozonolysis and photochemical oxidation. For example, the second paragraph of page 3 is discussing ozonolysis of MVK and MACR, which predominantly form from OH initiated oxidation.

**Authors' response:** We have revised the paragraph and added a sentence on the OH-initiated formation of MVK and MACR.

**Changes to manuscript**; page 3, line 29, sentence added:
"In the gas phase, MVK and MACR are also produced via the OH-initiated oxidation of isoprene, which is the predominant pathway during daytime."

8) Page 4, lines 8 – 11. This statement is somewhat misleading. Regarding the contribution of isoprene ozonolysis SOA, Kleindienst et al 2007 conclude: "Thus, even in light of the increased yield for the ozonolysis of isoprene measured in this study, the ozonolysis reaction probably remains a minor contributor to secondary organic aerosol in PM2.5 from the atmospheric oxidation of isoprene."

**Authors' response:** We have revised the sentence as shown below.

**Changes to manuscript**; page 4, lines 8 – 11:
"Ensuing studies, to re-examine the contributions to SOA by the ozone-initiated oxidation of isoprene, indicate that contributions to aerosol mass by the ozonolysis reaction pathway are likely minor compared to those from the daytime OH reaction pathway (Kleindienst et al., 2007)."

9) Page 5, Section 2.1
This section is confusing in a few respects. First, the authors mention that NOx ratio was varied, but later say experiments were conducted under low-NOx conditions. Second, they discuss operating in the chamber with cloud processing, but it isn't clear how this was taken into account by the model. If cloud droplets were formed wouldn't aerosol either be rained out or undergo a completely different type of processing?

**Authors' response (1):** Regarding the $NO_x$ ratio, there were various experiments conducted during those CLOUD 9 and 10 campaigns, including some on variation of NOx ratios. For this study, we focus only on the low-$NO_x$ cases. We have rephrased the sentence on line 24 to clarify this point.

**Changes to manuscript:** page 5, line 24 – 25, sentences revised:
"Two measurement campaigns were conducted that included experiments with the CLOUD chamber operated under low $NO_x$ conditions. In this work, we focus on the data from those low-$NO_x$ experiments for the purpose of parameterizing our model and comparisons to the CLOUD measurements. (1) During the CLOUD 9 campaign, …"

**Authors' response (2):** Regarding cloud formation in the chamber. Fuchs (2017) report that clouds were formed for typically 5 – 8 minutes followed by much longer periods of lower relative humidity (> 1 hour). For the purpose of model–measurement comparison, only data that were taken prior to the first cloud formation event or at least 60 min after a cloud formation event were considered (when the gas-phase conditions had been stabilized for at least 30 min). Potential effects of prior cloud formation events on aerosol particle concentrations during the seeded experiments are discussed in Section 3.2.1 (e.g. page 25, lines 23 – 30). One key effect of the expansions to form clouds is a reduction of the inorganic seed mass concentration present 60 min later, when steady state is re-established.

10) Page 5, Section 2.1 – Were the aerosol particles dried before SMPS and AMS measurements? Are there any measurements of the liquid water content of the particles under the experimental conditions?

**Authors' response:** The aerosol particles were dried before the SMPS and AMS measurements. There were no measurements of hygroscopic growth factors or liquid water content. The equilibrium partitioning computations with the AIOMFAC model provide predicted water uptake from low to high RH conditions (Figs. 4, 5, 7 and 9).

11) P5 Lines 30 – Were the isoprene measurements used in any way in the model? Why not just input the measured concentrations? Did the modeled isoprene and reacted isoprene agree with the measurements?

**Authors' response:** The MCM-modelled amount of isoprene reacted was confirmed to be consistent with a set of special chamber measurements conducted to test the capability of MCM for this purpose (Fuchs, 2017; Fig. 3.7 there). For that purpose, the chamber was first filled with isoprene (no ozone inflow) and after isoprene concentrations were stable, ozone was injected, and the concentration of isoprene reacted measured. The MCM-predicted isoprene reacted concentrations were about 4 % – 12 % low than the measured ones, which is considered to be a small difference. This suggests that MCM is capable of modeling isoprene reacted also under other conditions, such as when inflows of ozone and isoprene occur simultaneously. For modeling SOA formation under continuous flow conditions, our goal is to link the amount of SOA formed to the amount of isoprene reacted, not the mean isoprene concentration in the camber. Therefore, a tracer "isoprene reacted" was introduced in MCM for our modeling work and the MCM simulation informed by the chamber inflow mixing ratios and other known/measured environmental variables characterizing the chamber experiments.

**Changes to manuscript:** page 5, line 34, sentence added:
"Separate experiments in the CLOUD chamber had confirmed that MCM is capable of modeling the concentration of reacted isoprene (Fuchs, 2017, Section 3.5.3)."

12) P6 Lines 6 – 8. Was the aerosol density assumed to be 1.3 or calculated according to the parameterization?

**Authors' response:** The average organic aerosol density value of 1.3 g cm$^{-3}$ was determined using the parameterization by Kuwata et al (2012). We have rephrased the sentences to clarify this; see our response to comment 3) of referee 1.

13) P6 Lines 9 – 10. Earlier you said the chamber was run in continuous-flow mode. Here yields are calculated as a function of time. Are you using transient concentrations while running in continuous-flow mode or batch mode? The table indicates a single set of conditions. It is confusing where the yield is coming from?

**Authors' response:** SOA mass yields were calculated using measured mass concentrations of SOA at different points in time after/when the concentrations in the chamber reached approximately steady-state conditions, while the chamber was run in continuous-flow mode. The MCM simulation provided the average concentration of isoprene reacted for the (transient) SOA yield calculation. The data for the CLOUD experiments listed in Table 2 refer to typical examples for yields calculated for a specific concentration of reacted isoprene. We have added a table in the revised SI to provide additional data (new Table S4, shown below). Those are the data used for the seed-free case and associated model fit shown in Fig. 3.

**Table S4**. Measured SOA mass concentrations and mass yields at approximately 35 % or 85 % RH for the seed-free experiments (CLOUD 10 data).

| SOA mass ($\mu g\ m^{-3}$) | SOA yield (%) | RH (%) | $T$ (°C) | Isoprene reacted ($\mu g\ m^{-3}$) |
|---|---|---|---|---|
| 3.35 | 2.35 | 35 | 5 | 142.8 |
| 3.47 | 2.26 | 35 | 5 | 153.4 |
| 3.46 | 2.14 | 35 | 5 | 161.8 |
| 3.51 | 2.08 | 35 | 5 | 168.4 |
| 2.76 | 1.59 | 35 | 5 | 173.9 |
| 2.33 | 1.45 | 35 | 5 | 161.0 |
| 2.01 | 1.38 | 35 | 5 | 146.0 |
| 1.90 | 1.36 | 35 | 5 | 139.6 |
| 1.84 | 1.37 | 35 | 5 | 134.1 |
| 1.72 | 1.32 | 35 | 5 | 130.2 |
| 1.68 | 1.41 | 35 | 5 | 119.2 |
| 1.52 | 1.35 | 35 | 5 | 112.7 |
| 1.21 | 1.06 | 35 | 5 | 114.7 |
| 5.49 | 3.67 | 85 | 5 | 149.9 |
| 5.63 | 3.47 | 85 | 5 | 162.0 |
| 5.70 | 3.32 | 85 | 5 | 171.6 |
| 5.70 | 3.19 | 85 | 5 | 178.7 |
| 5.71 | 3.15 | 85 | 5 | 181.5 |
| 4.83 | 2.75 | 85 | 5 | 175.7 |
| 3.76 | 2.41 | 85 | 5 | 156.0 |
| 3.09 | 2.27 | 85 | 5 | 136.1 |
| 2.60 | 2.11 | 85 | 5 | 122.9 |
| 2.37 | 2.08 | 85 | 5 | 113.8 |
| 2.31 | 2.15 | 85 | 5 | 107.6 |
| 2.02 | 1.97 | 85 | 5 | 102.3 |
| 1.78 | 1.85 | 85 | 5 | 96.4 |
| 1.57 | 1.74 | 85 | 5 | 90.0 |

14) P9 Lines 2-5. I don't understand what you are trying to say here.

**Authors' response:** This statement was unclear; we have modified it. Please see our response to referee 1, comment 4).

15) 2.3.2. It is a little hard to understand why the MCM was used or what it was used for. In the end, you tune the model to produce all the generic compounds that partition into the condensed phase? Why go to all the trouble of using the MCM? Why not just react isoprene at the know rate constant to make a few products and then tune that output? Most of the representative products are from OH oxidation, but the experiment was primarily an ozonolysis experiment.

**Authors' response:** The MCM model was run to simulate a continuous flow experiment covering several hours with the aim for providing the data on isoprene reacted and early generation products formed from the various reactions with ozone and OH, including consideration of some variations in inflow concentrations, RH and pressure. This makes using MCM a convenient choice.

16) Page 12 – 13 – This entire section is extremely confusing. Presumably, the authors are using the MCM to calculate reacted isoprene and the amount of first-generation oxidation products. Therefore, it isn't clear why these two pages of text are so complicated and confusing. Analytical solution for time-dependent concentration of a particular species in continuous-flow mode are widely available in the literature and not difficult to solve. Why is there a need to introduce a tracer reacted isoprene concentration when the MCM should already calculate this?

**Authors' response:** A regular MCM simulation will provide the time-dependent isoprene concentration, while the use of the tracer compound "isoprene reacted" serves the purpose of keeping better track of the time-dependent evolution of the reacted molecular concentration of isoprene. This is useful because the inflow mixing ratios, as determined from the measurements were not perfectly steady. While there may be analytical solutions available to approach a simulation in a different way, we consider the use of MCM informed by the experimental conditions and inflow concentrations to be a valid and appropriate approach for this application.

17) P 13, line 13  - Why average over 30 minutes if the chamber is at steady state and you are using model outputs? Shouldn't the average over the 30 minutes be the same as at any time during that interval if the chamber is at steady state? Was the chamber at steady state?

**Authors' response:**  In practice, the chamber was not continuously at perfect steady state due to some fluctuations in the inflow conditions, described as quasi-steady-state, which were considered in the MCM simulation. Averaging further serves to reduce measurement noise. Therefore, we are averaging over periods of 30 minutes.

18) P 13 line 17 – 19. Not clear what you are trying to say here.

**Authors' response:** This sentence was unclear; we have revised it (also in response to a comment by referee 1).

**Changes to manuscript**; page 13, lines 17 – 19, sentence revised to:
"We note that the molar yields of individual MCM-predicted products differed between the simulations of CLOUD 9 and CLOUD 10 cases due to differences in experimental conditions. This is accounted for by means of different sets of input concentration data for the gas–particle partitioning calculations of seeded or seed-free cases when compared to the CLOUD experiments.

19) P13, lines 21 – 22. Conditions weren't stable if the water saturation is changing, particularly given you're investigating the role of RH on yield.

**Authors' response:** Agreed; the inflow conditions were stable (held at subsaturation RH), while the chamber RH relaxed back to the inflow RH. We have revised the statement.

**Changes to manuscript;** page 13, lines 21 – 22, sentence modified to:
**"**After cloud formation, the inflow RH was returned to a subsaturated level and maintained for about 1.5 hours during the first hour of which the system relaxed back to near steady state conditions.**"**

20) P13 – If there was cloud formation during the experiments, wouldn't this impact yield? Wouldn't any cloud-processing reactions alter the SOA composition? How were these taken into account?

**Authors' response:** As stated on page 13, lines 23 – 26, we did not use any measured SOA mass concentrations or simulated product yields for an extended period after a cloud cycle. After one chamber residence time period, the impact on SOA composition was considered to be small. Note that for the seeded cases, the main findings from this study on the seed impact related to changes in RH are directly from the gas–particle partitioning calculations carried out at various RH levels. The data from the seeded CLOUD experiment were only used for determining the need for setting a global scaling parameter of 0.55 (section 3.2.1) and related comparison to the model predictions.

21) P15, lines 24 – 28. How unique are the solutions you arrived at for these scaling parameters? With 6 free parameters, I imagine there is a large solution space and many different solutions may describe the yield equally well or nearly equally well given the measurement uncertainty.

**Authors' response:** Yes, the solution we found is not unique, but still reasonably constrained by the use of data spanning different isoprene concentrations and relative humidities (as well as seed concentrations in the case of the seeded experiments). In this context (see also our response to referee 1, comments 21 – 23). We have added the following clarification to the manuscript:
"We note that our set of determined scaling parameters (i.e. the branching ratios) is not unique; other combinations of scaling parameters may provide similar agreement with the observations. The determined scaling parameters for the surrogate species are provided in Table 1."

22) P16 lines 23-24 and throughout document. Since yield is dependent on SOA mass loading, when listing yield, mass loadings also need to be specified. Similarly, when comparing yield under different RH and T, list the mass loading.

**Authors' response:** We have revised this sentence and other such cases in the manuscript to add the SOA mass concentration information.

**Changes to manuscript:** page 16, lines 22:
Measurements from the CLOUD chamber for the seed-free experiments suggest an SOA mass yield of 1 % – 2.4 % at 35 % RH (SOA mass concentration, $M_{SOA}$, of ~1.5 to 3 µg m$^{-3}$) and 1.7 % – 3.7 % at 85 % RH ($M_{SOA}$ of ~1.5 to 5 µg m$^{-3}$) at a temperature of 5 °C.

23) P17, lines 11-14. Table S1 actually shows the pseudo-molecular yields for the surrogate compounds. I could not find the measured and modeled SOA mass concentration vs reacted isoprene in the document. This is crucial to understanding what was done, so it really needs to be made available. There are only a few entries in Table 2 in the main text that do not appear to correspond to the data in the figures.

**Authors' response:** Table S1 shows the pseudo-molar yields for the MCM-modelled early generation products by Fuchs (2017). Measured and modelled SOA mass concentration versus isoprene reacted concentrations are shown in Fig. 3. We have added a (new) Table S4 to the SI, which lists the measured and modelled SOA mass concentration data shown in Fig. 3.

**Changes to manuscript:** page 17, lines 13 – 14, sentence revised and statement added:
"… obtained for the same conditions using periods for which the quasi-steady state assumption was applicable. Those data are shown in Fig. 3 and listed in Supplementary Information (SI) Table S4. The set

of pseudo-molar yields of the surrogate components used by the partitioning model are provided in Table S1."

24) P17, lines 20-30. The data shown in Figure 3 seem to indicate a bimodal distribution of yield values, with some of the values aligning well with the model data and other measured values much higher than the modeled data. Were these all a single experiment? There is only one experiments for each set of conditions listed in Table 2, so it isn't clear where the multiple datapoints come from. Are they from different points in the equilibrium or different points as the system approached equilibrium or something else? Were the experiments repeated more than once for a given set of conditions? From Table 2, it appears only one experiment was run for each condition.

**Authors' response:** Figure 3 shows a comparison of measured SOA mass concentrations from CLOUD 10 experiments versus computed SOA mass concentrations by the equilibrium partitioning model at varying levels of isoprene loading. P17, lines 27– 29 discuss that the measured SOA mass concentrations show some scatter at isoprene reacted concentrations between ~140 to 175 µg m$^{-3}$, which the model is unable to reproduce while good agreement at lower isoprene reacted levels is achieved (i.e. the model's scaling parameters were tuned based on achieving good agreement with those lower SOA mass concentration data for both 35 % and 85 % RH conditions simultaneously).

The different data points of the SOA mass measurements are coming from a long-duration chamber experiment, during which the isoprene and ozone mixing ratios in the inflow were adjusted from time to time. The data are taken at different times (averaged 30 min periods) when the system was in quasi-equilibrium at varying isoprene (inflow) loading levels and a fixed RH level of either approximately 35 % (Fig. 3a) or 85 % (Fig. 3c). The new Table S4 added to the revised manuscript/SI lists those values (see response to comment 23). Table 2 lists one representative case of the SOA mass concentrations that were measured or modelled at varying RH levels for a fixed isoprene loading level (the case shown in Fig. 4).

25) It isn't necessarily surprising that all the data don't fit a single curve perfectly due to variability in experiments. But what does this mean for your modeling? The "outlier data" have approximately 1.5x the yield of the other data. This is similar or even greater than the differences between the 35 and 85% RH experiments. Given that the model was tuned to the lower yield data for the 35% yield case, how do you know the differences in the model are significant? If you tuned the model to the outlier data in the 35% case how will it fit the 85% RH data?

**Authors' response:** If the model were tuned to fit the outlier data at 35 % RH, the modelled curve would only fit 2 experimental data points well, rather than all the outliers, due to the outlier data comprising similar SOA mass concentrations at varying higher levels of isoprene reacted. The model would similarly fit only 1 experimental data point at 85 % RH. This is an indication that even though the outliers have apparently a 1.5-fold yield compared to the other data, they would represent a poor choice for a fit of most data at 35 % RH as well as at 85 % RH. Therefore, our decision was to focus on achieving better agreement with the other data points, leading to the relatively good alignment with the model calculation as shown in Fig. 3. Importantly, if one were to tune the model to the apparent "outlier" measurement data at both low and high RH, there would still be a clear RH effect on predicted SOA mass concentrations. For example, for about 150 µg m$^{-3}$ of isoprene reacted, the "outlier" experimental data indicate formation of about 3.4 µg m$^{-3}$ of SOA at 35 % RH, yet about 5.4 µg m$^{-3}$ at 85 % RH. Although, the experimental error bars are large in both cases, we argue that the combination of several data points from the measurements at low and high RH used to constrain the surrogate component

scaling parameters (branching ratios) results in a reasonable description of the RH effect on SOA mass concentration. Furthermore, that determined set of model scaling parameters leads to agreement, within error, with all data points shown in Fig. 3.

26) P 19, First paragraph – Yield needs to be compared at similar SOA mass loading, since it is dependent on mass loading at these concentrations. How does yield change at a similar mass loading? Also the absolute difference in yield is only 1%. In general, experiment-to-experiment variability is going to be larger than 1% in absolute yield. How many experiments are evaluated here; is there only one per condition? Given all the uncertainties with wall loss, experimental variability, etc. is a 1% difference in yield significant? Is there reason to believe wall loss isn't changing with RH? Would changing the gas-phase wall loss as a function of RH account for the differences in observe yield?

**Authors' response:** There are different ways by which one can compare SOA yields and SOA mass concentrations. Using a common denominator, here the same amount of isoprene reacted, is arguably showing an impact of RH. In terms of comparison at the same $M_{SOA}$: if we compare the SOA mass yields for about 2 µg m$^{-3}$ of SOA from the model calculation, we get 1.33 % SOA yield at 35 % RH and 2.10 % at 85 % RH (a relative increase of 58 % in yield over this RH range). This is evaluated using the model whose parameter set was tuned based on all the experimental data points shown in Fig. 3.

27) P19, lines 16 – 21. How does the computed particle-phase water content of the SOA compare to measurements of the hygroscopic growth factor? The mass fraction of water at 80% RH appears to be about 25%. This seems much larger than the hygroscopic growth factor typically measured for pure SOA at 80% RH.

**Authors' response:** No hygroscopicity measurements were made during those CLOUD experiments. The AIOMFAC model predicted $\kappa_{org}$ value at 85 % RH is 0.157 for the case shown in Fig. 4 (treating the SOA mass concentration as fixed, i.e. the same at 0 % RH and 85 % RH for the purpose of computing $\kappa_{org}$). This relatively high organic hygroscopicity is (at least in the model system) the result of having organic surrogate compounds of relatively high O:C ratios dominating the SOA composition at the low organic aerosol mass concentration predicted (2.73 µg m$^{-3}$ at 85 % RH). This $\kappa_{org}$ is not unreasonable for isoprene-derived SOA, but certainly higher than typical cases for monoterpene-derived SOA (e.g., Rastak et al., 2017). It is possible that experiments conducted under conditions leading to much higher SOA mass concentrations would have a lower organic aerosol hygroscopicity.

28) P21, lines 23 – 25. What were the conclusions of this study? Can they be used to support your hypothesis that differences in the vapor wall loss are responsible for the differences in yield? It seems out of place to just say studies have been done.

**Authors' response:**  This sentence will be deleted.

29) P21 Lines 26-27  – I'm not sure what is meant here. You didn't measure the temperature dependence of the vapor pressure of individual compounds.

**Authors' response:** We agree, this sentence was unclear. We have replaced it.

**Changes to manuscript:** page 21, lines 26 – 27, sentence rephrased:
"Given that the pure-component saturation vapour pressures of the semi-volatile components decrease

with temperature, the SOA yields are in turn expected to increase when comparing formation at 25 °C vs. at 5 °C."

30) P21 lines 29 – 30. What fraction of isoprene reacted with OH vs ozone in the experiments? This should be reported in the manuscript.

**Authors' response:** About 30 % of isoprene reacted with generated OH; see response to comment 5 above.

31) P21 -22, lines 31 – 34. The discussion here shifts to isoprene photooxidation SOA. This is somewhat confusing. I'm not sure it is particularly relevant to this paper unless a significant fraction of isoprene reacted with OH.

**Authors' response:** We opt for including this discussion because it provides an additional set of model–measurement comparisons using a system of surrogate components that was established by Chen et al. (2011) for the photooxidation of isoprene. This system is completely independent from the simplifications and model parameter tuning we have made based on the CLOUD experiments. As such, it offers an additional, independent case to model the effect of RH on SOA yield enhancement for isoprene-derived SOA.

32) P21 lines 19 – 20. Wouldn't a similar logic apply to the RH dependence as well?  In the unseeded experiments the model is tuned to the 35% RH results.

**Authors' response:** (likely this refers to P22 lines 19 – 20)**.** In principle, yes, but note that in the ozone-initiated case we had measurements at both 35 % and 85 % RH to constrain the model parameters, while for the photooxidation case with the surrogate system by Chen et al. the reference data is only available at 40 % RH.

33) P24 lines 1-2. OH is formed from ozonolysis of isoprene with significant yield.

**Authors' response:** Yes, thanks for noting this. We have rephrased this sentence.

**Changes to manuscript;** page 24 line 1:  "However, in the absence of light in the CLOUD experiments, the rates of aqueous SOA formation pathways are limited to dark processes and are likely only of minor importance."

34) P24 lines 3-6. It isn't clear what you mean when you say the chamber was run under different conditions. The only significant difference I see is the T and the seed identity. Why would a 5 C difference in temperature perturb the gas-phase chemistry such that a different product distribution was necessary? Doesn't the fact that a single gas-phase chemistry product distribution fails to describe these experiments with different seed but similar gas-phase chemistry is similar indicate that something more is going on?

**Authors' response:**  Aside the small temperature difference, a higher total pressure was used. However, comparing Tables S1 and S2 (showing the used pseudo-molar yields) the pseudo-molar yields, as determined from the MCM simulations of early generation species, were different between the seeded and seed-free cases, but not dramatically different. The fact that we needed to apply a correction

(scaling) factor to the surrogate component branching ratios in the seeded case is indeed an indication that something else was different. One possible reason is that the wall losses differed requiring a correction that is implicitly factored into the surrogate yield correction factor of 0.55 in our model. While it is unclear what exactly the main reason for the differences was, possible reasons are discussed on page 25 (section 3.2.1).

Also, the phrasing of the sentence on lines 4 – 6 was not clear; we have revised it.

**Changes to manuscript:** (Page 24, lines 4-6), sentence rephrased to:
"The MCM-predicted molar yields of the early-generation products for the conditions of the seeded experiments were different from those for the seed-free ones. In contrast, the predicted molar yields of the various seeded experiments were similar; therefore, for consistency and ease of comparison among different model calculations for seeded cases, a single set of molar yields was used (listed in Table S2)."

35) P24 Lines 10 – 14. If the vapor wall loss is indirectly impacted by the RH via the condensational sink rate, wouldn't this also impact you results in the seed-free case? Why does it matter in the seed case, but not the seed-free case?

**Authors' response:** Yes, it may impact the seed-free case as well, but likely to a different (smaller) degree, since the particle size distributions and hygroscopicities were different. Also, the seed-free experiments did not include cloud-formation episodes.

36) P25, line 13-14. Here the authors indicate the AMS was used to measure SOA mass. On page 6 the authors indicate the SMPS was used to calculate SOA mass. Please clarify what measurement was used under which conditions. If the SMPS was used in unseeded cases and the AMS in the seeded cases, was a comparison done of the AMS and SMPS derived masses? Are they similar?

**Authors' response:** In the presence of inorganic seed amounts, the SMPS data were used to determine the particle volume and the AMS data were used to determine the mass fractions of inorganic seed and SOA. In combination, one obtains SOA and seed mass concentrations (when accounting for approximate SOA and seed densities).

**Changes to manuscript:** (Page 25, lines 12 – 14), sentence rephrased to:
"It is noted that the branching ratios could be scaled by any factor between 0.5 and 0.6 to achieve agreement with the experimental data, partially due to the ~ 50 % uncertainty in the AMS measurements, which were used alongside with SMPS-derived particle volume data to calculate the organic mass concentrations, as mentioned in the thesis by Fuchs (2017)."

37) P30 Lines 16 – 32. This is confusing. Are you saying you used the downscaled molar yield derived from the seeded experiments to conduct a seed-free model run? If so, I'm not sure what is learned, since the product yields derived from experiments in the seed-free cases were significantly larger than what was used in the comparison. What exactly is learned by comparing a model where the molar yields don't represent what was actually observed in the seed-free case?

**Authors' response:** The purpose of this exercise is to provide a direct model–model comparison to evaluate the RH-dependent SOA yield enhancement for a seed-free case that is otherwise identical (in the model system) to the seeded cases. This eliminates case-specific tuning based on the CLOUD data for seeded vs. seed-free situations, which, as discussed above, may have been caused by differences not directly related to the chemical formation mechanisms. Those differences would likely be absent in

ambient air or chamber experiments with negligible wall loss effects. Hence, as described in this paragraph, we determine whether such a seed-free partitioning calculation would result in a greater or smaller yield enhancement compared to the seeded cases. The tuned partitioning model allows us to conduct such evaluations independent of variability among experiments that may mask the targeted RH effects. Therefore, we argue that those RH-dependent yield enhancement calculations provide relatively robust results of interest to understand the indirect effects of RH on isoprene-derived SOA.

**38) P31 line 11. What is meant by an SOA yield "by volume"?**

**Authors' response:** Czoschke et al. (2003) conducted Teflon (PTFE) bag experiments for which only the starting concentration of the organic gas (precursor of SOA) was known, with the reactions being ozone-limited. They reported yields as $Y$ = (volume conc. of aerosol produced) / (volume of reactive organic gas consumed) rather than using the typical, mass-based definition of organic aerosol yield. Because of estimates made, they mention that they consider the results to be qualitative rather than quantitative. Therefore, we have modified this sentence.

**Changes to manuscript:** page 31, lines 10 – 11, sentence changed to:
"The chemical characterization of the aerosol by Czoschke (2003) revealed an enhancement of highly oxidized compounds in the acidified cases. Qualitatively, comparing pairs of experiments using non-acidic or acidic inorganic seed, in presence or absence of an OH scavenger, the SOA yields were increased by approximately 2 – 3 times in the acidified cases."

**References**

Cappa, C. D., Lovejoy, E. R., and Ravishankara, A. R.: Evidence for liquid-like and nonideal behavior of a mixture of organic aerosol components, PNAS, 105, 18687-18691, 2008.

Czoschke, N. M., Jang, M., and Kamens, R. M.: Effect of acidic seed on biogenic secondary organic aerosol growth, Atmospheric Environment, 37, 4287-4299, 2003.

Fuchs, C.: Investigation of the role of $SO_2$ and isoprene in aqueous phase secondary aerosol formation, Dissertation ETH Zurich 2017 No. 24157, 177 pages, 2017.

Hallquist, M., Wenger, J. C., Baltensperger, U., Rudich, Y., Simpson, D., Claeys, M., Dommen, J., Donahue, N. M., George, C., Goldstein, A. H., Hamilton, J. F., Herrmann, H., Hoffmann, T., Iinuma, Y., Jang, M., Jenkin, M. E., Jimenez, J. L., Kiendler-Scharr, A., Maenhaut, W., McFiggans, G., Mentel, T., Monod, A., Prevot, A. S. H., Seinfeld, J. H., Surratt, J. D., Szmigielski, R., and Wildt, J.: The formation, properties and impact of secondary organic aerosol: current and emerging issues, Atmos. Chem. Phys., 9, 5155-5236, 2009.

Jenkin, M. E., Young, J. C., and Rickard, A. R.: The MCM v3.3.1 degradation scheme for isoprene, Atmos. Chem. Phys., 15, 11433-11459, 10.5194/acp-15-11433-2015, 2015.

Johnson, D., Utembe, S., Jenkin, M., Derwent, R., Hayman, G., Alfarra, M., Coe, H., and McFiggans, G.: Simulating regional scale secondary organic aerosol formation during the TORCH 2003 campaign in the southern UK, Atmospheric Chemistry and Physics, 6, 403-418, 2006.

Kleindienst, T. E., Lewandowski, M., Offenberg, J. H., Jaoui, M., and Edney, E. O.: Ozone-isoprene reaction: Re-examination of the formation of secondary organic aerosol, Geophysical Research Letters, 34, 2007.

Kuwata, M., Zorn, S. R., and Martin, S. T.: Using Elemental Ratios to Predict the Density of Organic Material Composed of Carbon, Hydrogen, and Oxygen, Environmental Science & Technology, 46, 787-794, 10.1021/es202525q, 2012.

Rastak, N., Pajunoja, A., Acosta Navarro, J. C., Ma, J., Song, M., Partridge, D. G., Kirkevåg, A., Leong, Y., Hu, W. W., Taylor, N. F., Lambe, A., Cerully, K., Bougiatioti, A., Liu, P., Krejci, R., Petäjä, T., Percival, C., Davidovits, P., Worsnop, D. R., Ekman, A. M. L., Nenes, A., Martin, S., Jimenez, J. L., Collins, D. R., Topping, D. O., Bertram, A. K., Zuend, A., Virtanen, A., and Riipinen, I.: Microphysical explanation of the RH-dependent water affinity of biogenic organic aerosol and its importance for climate, Geophys. Res. Lett., 44, 5167-5177, 10.1002/2017GL073056, 2017.